# Density-driven Regularization for Out-of-distribution Detection

**Wenjian Huang**[*1], **Hao Wang**[*1], **Jiahao Xia**[2], **Chengyan Wang**[3], **Jianguo Zhang**[4,5†]

[1]Dept. of Computer Science and Engineering, Southern University of Science and Technology
[2]Faculty of Engineering and IT, University of Technology Sydney
[3]Human Phenome Institute, Fudan University
[4]Research Institute of Trustworthy Autonomous Systems and
Dept. of Computer Science and Engineering, Southern University of Science and Technology
[5]Peng Cheng Lab, Shenzhen, China
huangwj@sustech.edu.cn, 11812301@mail.sustech.edu.cn
jiahao.xia@student.uts.edu.au, wangcy@fudan.edu.cn, zhangjg@sustech.edu.cn

## Abstract

Detecting out-of-distribution (OOD) samples is essential for reliably deploying deep learning classifiers in open-world applications. However, existing detectors relying on discriminative probability suffer from the overconfident posterior estimate for OOD data. Other reported approaches either impose strong unproven parametric assumptions to estimate OOD sample density or develop empirical detectors lacking clear theoretical motivations. To address these issues, we propose a theoretical probabilistic framework for OOD detection in deep classification networks, in which two regularization constraints are constructed to reliably calibrate and estimate sample density to identify OOD. Specifically, the density consistency regularization enforces the agreement between analytical and empirical densities of observable low-dimensional categorical labels. The contrastive distribution regularization separates the densities between in distribution (ID) and distribution-deviated samples. A simple and robust implementation algorithm is also provided, which can be used for any pre-trained neural network classifiers. To the best of our knowledge, we have conducted the most extensive evaluations and comparisons on computer vision benchmarks. The results show that our method significantly outperforms state-of-the-art detectors, and even achieves comparable or better performance than methods utilizing additional large-scale outlier exposure datasets.

## 1 Introduction

Deep networks have achieved impressive accuracy on many classification tasks when evaluated in a static and closed environment, where the test samples are sampled from the same distribution as the training data. However, when deployed in real-world applications, deep learning (DL) models often encounter anomalous samples different from the training data. Detecting such unexplored samples is referred to as out-of-distribution (OOD) detection [1–13], which is crucial for reliable decision making, especially in safety-critical applications such as autonomous driving, medical diagnosis and secure authentication[1, 3, 8, 9]. This topic is particularly important for deep classifiers because it has been well documented that it can be easily fooled by OOD samples to produce high confidence predictions [14, 15, 7, 8, 11, 9, 16, 4]. In recent years, most reported and favored strategies for DL-based OOD detection are to design the detectors by inspecting the distribution of high-level features extracted by the relevant target task network. The studies are briefly reviewed below.

---

[*]Equal contribution.
[†]Corresponding author.

36th Conference on Neural Information Processing Systems (NeurIPS 2022).

As a pilot study, Hendrycks et al. proposed to detect OOD samples based on maximal softmax probability (Msp) [16], i.e., $\max_y p(y|x)$, which has become a common baseline for OOD detection [10, 4–7, 12, 9, 17, 13, 18]. Subsequently, Liang et al. proposed the Odin method, which provided temperature scaling and input perturbation strategies, to push the Msp score further apart for ID and OOD data [17]. Hsu et al. generalized the Odin method to avoid access to OOD samples for parameter tuning [5]. Another similar technique, which uses softmax probability for OOD detection, is pNML [3]. In pNML, Bibas et al. detected OOD samples using their proposed regret score $\log \sum_y \overline{p}(y|x)$, where $\overline{p}(y|x)$ denotes online fine-tuned softmax probability by adding the test sample $x$ to the training set with an arbitrary label $y$. However, these methods relying on softmax probability score $p(y|x)$ can suffer from overconfidence problem since many studies have shown that neural networks can produce arbitrarily high softmax probability for OOD inputs [4, 19, 1, 10, 8, 12, 20].

Another line of research utilizes the distribution of extracted high-level features $f(x)$ to estimate ID data density $p(x)$ for OOD detection as modeling $p(x)$ directly with high-dimensional input $x$ is prohibitive in most cases. For example, Lee et al. [9] proposed using the Mahalanobis distance between the test sample's feature representations and the class-conditional gaussian distributions in the intermediate layers to train a logistic regression OOD detector. Essentially, this Mah method constructed the OOD score using $\max_y p(f(x)|y)$, in which $p(f(x)|y)$ was assumed to obey Gaussian distribution for the ID training data. Afterward, Sehwag et al. [21] and Zhou et al. [19] further extended the Mah model for unsupervised learning (no training labels available) and semi-supervised learning settings (small set of training labels available), respectively. However, the strong parametric assumption for these studies that class-conditional feature representation follows Gaussian distribution often fails in practical applications. Liu et al. [4] interpreted the softmax function from an energy perspective which assumed that $\log(p(x)) \propto \log(\sum_y \exp(f^y(x)/T))$ where $T$ was temperature coefficient and $f^y(x)$ denoted the classification logit for input $x$ and class $y$. The energy method proposed to use Helmholtz free energy $-T \log(\sum_y \exp(f^y(x)/T))$ as the OOD score. Nevertheless, we will explain in the method section that the underlying assumption of the energy model does not necessarily hold, which can lead to poor estimation of $p(x)$ for OOD detection.

Some other studies reported different empirical methods for OOD detection. For example, Sastry et al. [8] detected OOD samples by defining the deviation of test samples from training data based on a layer-wise Gram matrix that encoded pairwise feature correlations between channels. Inspired by the contrastive learning strategy, Tack et al. [22] contrasted different types of augmented training samples and proposed an empirical OOD score by measuring the distance between the test sample and nearest training samples in contrastive representation. Additionally, Sun et al. [1] observed that OOD data tend to trigger unit activation with larger variations and positive skewness. In light of this, the proposed ReAct model [1] directly truncated high activations in the penultimate layer, which was experimentally confirmed to be helpful in improving the performance of common OOD scores.

In summary, the existing methods have one or more of the following limitations: **(1)** Due to lack of clear probabilistic definition of OOD detection, some studies identify OOD samples using estimated discriminative probability $p(y|x)$ instead of data distribution $p(x)$, which can be susceptible to the influence of overconfident posterior; **(2)** Some researchers estimate $p(x)$ by imposing strong explicit parametric or implicit assumptions of high-level extracted features $f(x)$, which however are not necessarily justified and can impair the OOD detection performance; **(3)** Several empirical methods lack a clear theoretical motivation.

To overcome these limitations, we will explore how to reliably estimate ID data distribution $p(x)$ for OOD detection using the high-level representation features. First, we consider it is inappropriate to estimate the sample distribution $p(x)$ directly from the distribution of latent feature representation learned by the discriminative network since it only concerns the conditional probability $p(y|x)$ without any constraints to guarantee the reliability of the estimated $p(x)$. To address this issue, we propose a novel density-consistency regularization strategy: given that the distribution of low dimensional variable $Y$ can be estimated with high accuracy, this study enforces the consistency between empirical density $\widehat{p}(y)$ and the estimated density $p(y)=\int p(x,y)dx=\int p(x)p(y|x)dx$ derived from latent feature representation in deep layers, e.g., the last logit layer, on a batch-by-batch basis. Theoretically, with sufficient training samples (batch numbers), combining this consistency regularization with common supervised loss can reliably calibrate and estimate both $p(y|x)$ and $p(x)$, thus enabling effective OOD detection. Secondly, to ensure OOD samples highly deviated from the ID distribution can also be identified, we further propose contrastive distribution regularization to produce low-density estimates

for the distribution deviated samples. These two regularizations are properly combined to further improve the detection accuracy.

The following methods section will investigate how to achieve the above density regularizations. It includes building the consistency criterion with the law of large numbers, analyzing asymptotic properties for the deviation from consistency with the central limit theorem, developing asymptotic-test based regularization for the consistency hypothesis, and finally incorporating density regularization with contrastive distributions. Our contribution can be summarized as follows: **(1)** We propose a unified probabilistic framework for OOD detection in discriminative networks, which reveals that direct estimation of ID data distribution $p(x)$ via latent feature representations can be biased and introduces density-consistent and contrastive criteria to address this issue. Unlike previous studies, our method does not rely on any unproven assumptions. **(2)** We derive analytical regularization terms by analyzing the asymptotic properties of the density-consistent and contrastive criteria, and provide a simple yet surprisingly effective computational algorithm for OOD detection applicable to any deep networks with a softmax classifier. **(3)** Extensive experiments on various standard OOD detection benchmarks demonstrate that the proposed density-driven OOD detector achieves state-of-the-art performance compared with many other existing methods.

## 2    Related work

In general, OOD detectors can be broadly categorized into generative and discriminative approaches.

**OOD detection with generative models**    Deep generative models have been explored for OOD detection [6, 7, 23, 11, 24, 2, 25–28] as an intuitive strategy since they enable modeling likelihoods $p_\theta(x)$ for high dimensional input. One contributing aspect of this approach is that it directly models the input distribution without using labeled data[11]. However, it requires additional generative networks for classification tasks, which can be prohibitively challenging to optimize [4, 29, 1]. Moreover, significant issues have been highlighted that deep generative models can frequently assign spuriously higher likelihoods to OOD than ID samples [2, 7, 4, 25, 11, 24, 30, 10, 23, 6, 5, 1, 8, 31, 27], and the performance can often lag behind the discriminative approaches [7, 1, 29]. Recent studies revealed that it might be because deep generative models can be confounded by background statistics [11], or because of the discordance between the model's typical set and high-density region [23].

**Discriminative model-based OOD detection**    The discriminative model-based approach, which derives OOD score by using representations from a predefined classification task, has advantages over generative-based approaches by circumventing the difficult optimization process and obtaining better detection performance [1, 9, 4, 6], especially for complex datasets [22]. This can be because supervised learning effectively learns discriminative representations useful in identifying in-distribution samples by exploiting label information [8, 6, 19]. A brief overview of the baseline Msp model [16] and state-of-the-art discriminative OOD detection methods, including Odin [17], Mah [9], Gram [8], Energy [4], pNML [3] and ReAct [1], is presented in the introduction section.

**Discriminative model-based OOD detection with outlier exposure**    Some researchers have explored using auxiliary datasets of outliers, which allow the model to be explicitly regularized through fine-tuning, to increase the margin between the OOD scores of the ID and OOD data. Such an outlier exposure strategy has been reported by Oe [10], Oecc [32] and Energy [4] to be helpful in OOD detection. However, OOD samples are different and infinite. To avoid specific exposure OODs inducing biases, researchers typically used a large-scale diverse dataset as the outlier exposure dataset [10, 32, 4], e.g., 80 Million Tiny Images. Yet it is often not possible to have such a large amount of external data in many real-world tasks, especially those with large ID datasets.

## 3    Methods

### 3.1    Probabilistic framework for OOD detection

The goal of a machine learning classifier is to learn the mapping from $X$ to $Y$, where input $X$ is usually a high-dimensional random vector and $Y$ is a discrete random variable. Let $g(x, y)$ be the probability density function (p.d.f.) of $(X, Y)$ with respect to measure $\nu_X \times \pi_Y$, where $\nu_X$ is Lebesgue measure

in Euclidean space $R^m$ and $\pi_Y$ is discrete measure with $\pi_Y (Y = y) = 1, 1 \leq y \leq L$. $L$ is the number of classification categories. The classification probabilities for discriminant model $P(Y|X)$ can be represented

$$P(Y = y \mid X) = \frac{g(X, y)}{\int g(X, Y) d\pi_Y}. \tag{1}$$

For deep learning models, researchers generally considered the following softmax values as $P(Y|X)$.

$$P(Y = y \mid X) = \frac{\exp(h^y(X))}{\sum_{1 \leq y' \leq L} \exp(h^{y'}(X))} \tag{2}$$

where $h^y(X)$ is the classification logits produced by the deep neural networks. By connecting Eq.(1) and Eq.(2), $\exp(h^y(x))$ can be considered as the Boltzmann-Gibbs form for $g(x, y)$ from an energy-based perspective [33, 34], i.e.,

$$g(x, y) \propto \exp(h^y(x)). \tag{3}$$

Alternatively, $g(x, y) = \frac{1}{Z} \exp(h^y(x))$ and denominator $Z \triangleq \sum_{1 \leq y \leq L} \int \exp(h^y(X)) d\nu_X$.

For OOD detection, we first argue that OOD score can be more appropriately constructed from $P(X)$ rather than $P(Y|X)$, where the latter is commonly used in previous studies [16, 17, 5, 3], since the estimated $P(Y = y|X = x)$ can create over-confidence probability [9, 4] even when $P(X = x)$ is relatively small. In terms of $P(X)$, we can obtain the following estimated p.d.f. $k(x)$ based on Eq.(3):

$$k(x) = \int g(x, Y) d\pi_Y = \frac{1}{Z} \sum_{1 \leq y \leq L} \exp(h^y(x)). \tag{4}$$

$k(x)$ appears to be a reasonable OOD detection score. Similarly, Liu et al. proposed to use Helmholtz free energy $-T \log(\sum_{1 \leq y \leq L} \exp(h^y(x)/T))$ as OOD score with temperature $T$ [4], which is similar to $k(x)$ from a computational perspective, and demonstrated its advantage over OOD scores based on discriminative probability $P(Y|X)$ [16, 17].

However, directly calculating $k(x)$ as the OOD score can lead to poor density estimation since the prerequisite Eq.(3) is not clearly satisfied, which has been overlooked in previous studies. Particularly, classification models generally focus on improving the accuracy of $P(Y|X)$ by classification loss $L_{cls}$ without explicitly enforcing the validity of Eq.(3). Therefore, this study firstly aims to develop an explicit regularization for density calibration to satisfy the constraint in Eq.(3) based on the empirical density estimation of the ID training data. Specifically, since the empirical density function (EDF) of a low-dimensional random variable (r.v.) can be estimated with high accuracy, we start with estimating the EDF of discrete r.v. $Y$ and enforce the consistency between its EDF and its analytical parametric p.d.f. of the following form based on Eq.(3).

$$v(y) = \int g(X, y) d\nu_X = \frac{1}{Z} \int \exp(h^y(X)) d\nu_X \tag{5}$$

In the following sections, we will construct the analytical consistency regularization and implementation algorithm for ensuring the constraints of Eq.(3), thereby improving OOD detection performance, using the law of large numbers (LLN), central limit theorem (CLT) and asymptotic test statistics.

### 3.2 Consistency criterion for low-dimensional density estimation

As noted above, density function $k(x)$, estimated by high layer features, can be a useful OOD score if the underlying assumption Eq.(3) holds. To realize this assumption, we now enforce the consistency between analytical p.d.f $v(y)$ derived by Eq.(3) and the empirical density function (EDF) for low-dimensional variable $Y$, which can be reliably estimated by $\widehat{v}(y) = n_y/N$. $n_y = \sum_{1 \leq i \leq N} 1_{\{Y_i = y\}}$ represents the number of samples in class $y$ and $N = \sum_{y=1}^{L} n_y$ is the sample size. Namely, with Eq.(3), we derive the following property:

$$1 \approx \frac{v(y)}{\widehat{v}(y)} = \frac{1}{Z} \frac{\int \exp(h^y(X)) d\nu_X}{\widehat{v}(y)} = \frac{1}{Z} \int \frac{\exp(h^y(X))}{\widehat{v}(y)} d\nu_X; \forall 1 \leq y \leq L \tag{6}$$

By the law of large numbers, $\widehat{v}(y)$ converges to $v(y)$ almost surely (a.s.) and in probability as $N \to \infty$. Thus, the pivotal variable $\int \frac{\exp(h^y(X)))}{\widehat{v}(y)} d\nu_X$ converges to $Z$ a.s. and in probability. By the Monte Carlo method, $Z$ can be approximated by:

$$Z \approx \int \frac{\exp(h^y(X)))}{\widehat{v}(y)} d\nu_X \approx \frac{\sum_{1 \le i \le N} \exp(h^y(X_i))/N}{n_y/N}; \forall 1 \le y \le L \tag{7}$$

Obviously, the right-hand side (RHS) of Eq.(7) also converges to $Z$ a.s. and in probability as $N \to \infty$. Thus, we arrive at the proposition that for each class $1 \le y \le L$, the exponential of classification logit has a mean value proportional to the class ratio.

Since deep learning models are trained in a batch mode, in which random batch sampling with replacement allows bootstrap estimates, we now proceed to construct batch-wise statistics and analytical regularization for Eq.(7). We first give the following lemma. The proof is in Appendix A.1 of the supplementary material.

**Lemma 1.** *If Eq.(3) holds, then*

$$\frac{\sum_{1 \le i \le N} \exp(h^y(X_i))}{\sum_{1 \le i \le N} \sum_{1 \le l \le L} \exp(h^l(X_i))} - \frac{n_y}{N} \to_{a.s.} 0; \forall 1 \le y \le L \tag{8}$$

Lemma 1 gives a theoretical batch-wise constraint for the classification logit. However, the convergence rate is not specified here, making it challenging to design an adaptive regularization subjected to varying class sample $n_y$ and batch size $N$. To address this issue, in the next section, we will specify the convergence rate of $O(1/\sqrt{N})$ for Eq.(8) using the central limit theorem and accordingly propose the adaptive consistency regularization by our asymptotic test statistics.

### 3.3 Adaptive density-consistency regularization by asymptotic test

To construct the density-consistency regularization adaptive to different class size $n_y$ and batch size $N$, we now consider the convergence rate with respect to sample size for Eq.(8) by the following lemma 2. The proof is provided in Appendix A.2.

**Lemma 2.** *If Eq.(3) holds, then*

$$\Upsilon_N^y = \sqrt{N} \left( \frac{\sum_{1 \le i \le N} \exp(h^y(X_i))}{\sum_{1 \le i \le N} \sum_{1 \le l \le L} \exp(h^l(X_i))} - \frac{n_y}{N} \right) \to_d \mathcal{N}(0, \Delta^y) \tag{9}$$

*where $\Delta^y = [\nabla g(\mu^y)]^\top \Sigma^y [\nabla g(\mu^y)]$ and $g([x,y,z]^\top) = y/x - z$. $\mu^y$ and $\Sigma^y$ represent the mean and covariance matrix for random vector $u^y = \left[ \sum_{1 \le l \le L} \exp(h^l(X)), \exp(h^y(X)), 1_{\{Y=y\}} \right]^\top$.*

Lemma 1 and lemma 2, in fact, show that if implicit assumption Eq.(3) holds, i.e., $\frac{1}{Z} \exp(h^y(x))$ can accurately approximate density $g(x,y)$, we can then construct batch-wise statistics $\Upsilon_N^y$, in which $\frac{1}{\sqrt{N}} \Upsilon_N^y$ converges to zero at the rate of $O(N^{-1/2})$ by analyzing the distribution of low-dimensional $Y$. More specifically, $\Upsilon_N^y$ converges to a particular Gaussian distribution whose parameter can be reliably estimated.

Thus to satisfy assumption Eq.(3), which is deemed to provide a more accurate estimation of $P(x)$ for OOD detection, we propose to construct the asymptotic-test statistics using $\Upsilon_N^y$ from the perspective of hypothesis testing, and creatively transform the hypothesis reject criteria into a penalty term to support the assumption Eq.(3) can be accepted at any specific level of significance.

Specifically, we consider the following hypotheses:

$$H_0 : \theta^y = 0 \quad \text{versus} \quad H_1 : \theta^y \ne 0 \tag{10}$$

where

$$\theta^y = \frac{E[\exp(h^y(X))]}{E\left[\sum_{1 \le l \le L} \exp(h^l(X))\right]} - E1_{\{Y=y\}} \tag{11}$$

First, assumption Eq.(3) implies null hypothesis $H_0$, and accordingly leads to the following convergence in distribution by lemma 2:

$$[\Delta_N^y]^{-1/2} \Upsilon_N^y \to_d \mathcal{N}(0,1) \quad \text{and} \quad [\Delta_N^y]^{-1} [\Upsilon_N^y]^2 \to_d \chi^2(1) \tag{12}$$

where $\Delta_N^y \triangleq [\nabla g(\mu_N^y)]^\top \Sigma_N^y [\nabla g(\mu_N^y)]$ approximates and converges to $\Delta^y$ in probability. $\mu_N^y$ and $\Sigma_N^y$ denote sample mean and sample covariance for $\mu^y$ and $\Sigma^y$, respectively. From the perspective of asymptotic test theory [35], we can construct a $\chi^2$-test with the following reject region which has asymptotic significance level $\alpha$.

$$[\Delta_N^y]^{-1} [\Upsilon_N^y]^2 > \chi_{1,\alpha}^2 \tag{13}$$

where $\chi_{1,\alpha}^2$ is the $(1-\alpha)$th quantile of chi-squared distribution $\chi^2(1)$. When Eq.(13) is observed, the asymptotic test rejects $H_0$, as well as assumption Eq.(3). Hence, to ensure the assumption Eq.(3) is rightly accepted, we propose to transform the hypothesis test rejection criteria into the following regularization term:

$$\lambda \triangleq \gamma \frac{1}{L} \sum_{1 \le y \le L} \max \left( \log \left( [\Delta_N^y]^{-1} [\Upsilon_N^y]^2 / \chi_{1,\alpha}^2 \right), 0 \right). \tag{14}$$

The regularization term $\lambda$ will be added to the classification loss $L_{cls}$, where hyperparameter $\alpha \in [0,1]$ controls the significance level and $\gamma$ is the loss weight balancing the regularization term $\lambda$ and classification loss $L_{cls}$.

### 3.4 Density regularization with contrastive distribution

The above density-consistency regularization (DCR) can calibrate classification logit based on observable low-dimensional density $P(Y)$, which offers a more reliable $P(X,Y)$ estimate for ID-like samples. Nevertheless, the real test OOD samples can come from distribution deviating far from the ID distribution. To ensure these samples with large distribution deviations can also be accurately identified, we further propose the density-contrasting scheme, which enforces that random distribution-deviated (DD) samples produce low-density estimates.

We first define the distribution-deviating operator $\mathcal{S}$, which can be applied to ID data $X$ to generate DD samples $\widetilde{X} = \mathcal{S}(X)$. An ideal density estimator should yield the likelihood ratio $\frac{P(\widetilde{X})}{P(X)} \ll 1$ and $\left[ \frac{P(\widetilde{X})}{P(X)} \right]^r \approx 0$ for large constant $r$, i.e.,

$$\psi_{\mathcal{S}}(X) \triangleq \log \left[ \frac{P(\widetilde{X})}{P(X)} \right]^r = r \cdot \log \frac{\sum_{1 \le l \le L} \exp\left(h^l(\mathcal{S}(X))\right)}{\sum_{1 \le l \le L} \exp\left(h^l(X)\right)} \to -\infty. \tag{15}$$

We incorporate this constraint into the above DCR by defining $\widetilde{\Upsilon}_N^y$:

$$\widetilde{\Upsilon}_N^y \triangleq |\Upsilon_N^y| + \exp \left( \frac{1}{N|\mathbb{S}|} \sum_{\mathcal{S} \in \mathbb{S}, 1 \le i \le N} \psi_{\mathcal{S}}(X_i) + q \right) \tag{16}$$

where $\mathbb{S}$ denotes the distribution-deviating operator set. Here a constant $q$ is added to ensure numerical stability. Since the second term in RHS is considered close to zero, we can obtain that $[\Delta_N^y]^{-1}[\widetilde{\Upsilon}_N^y]^2$ also converges to $\chi^2(1)$ approximately. Thus, our final generalized density regularization with contrastive distribution can be defined:

$$\widetilde{\lambda} \triangleq \gamma \frac{1}{L} \sum_{1 \le y \le L} \max \left( \log \left( [\Delta_N^y]^{-1}[\widetilde{\Upsilon}_N^y]^2 / \chi_{1,\alpha}^2 \right), 0 \right) \tag{17}$$

Figure 1 shows the overview of the proposed density driven regularization (DDR) for OOD detection and Algorithm 1 presents the pseudo-code for the training of our DDR method[3]. It is worth noting that to increase numerical stability, we normalize the classification logit by subtracting the maximal logit before calculating $\widetilde{\lambda}$ in Algorithm 1. This normalization procedure is also widely used to avoid overflow when calculating softmax probability [36]. We will show, in Appendix A.3, that the normalized logit essentially yields the same results as the original logit. Finally, after fine-tuning, a monotonic transformation $\log(Z \cdot k(x)) = \log(\sum_{1 \le y \le L} \exp(h^y(x)))$ of the sample density $k(x)$ is used as the OOD score in this study for any testing sample $x$, which is theoretically equivalent to directly adopting $k(x)$ as the OOD score.

---

[3]Code for the proposed DDR is available at http://WenjianHuang93.github.io/files/OOD_DDR.zip

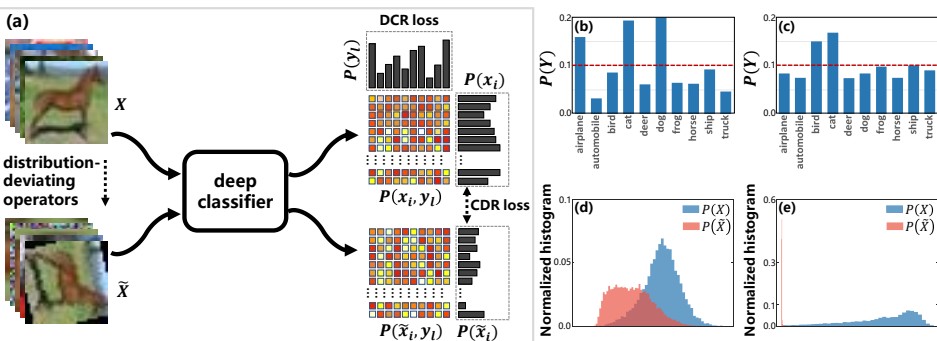

Figure 1: **(a)** Illustration of the proposed method. **(b~c)** The analytical class distributions $p(y_l)$ computed from classification logits for the overall test set (CIFAR-10, WideResNet) w/o and w/ density consistency regularization (DCR) loss, respectively. The summarized mean absolute error of the analytical class distribution w.r.t the actual empirical class distribution (red dotted lines) is reduced from 0.50 to 0.24 by DCR loss. **(d~e)** The normalized histograms for the sample density (test set of CIFAR-10, WideResNet) w/o and w/ contrastive distribution regularization (CDR) loss, respectively. The distributions of in-distribution (ID) sample density and distribution-deviated sample density can be well separated by the CDR loss.

## 4 Experiments

We demonstrate the effectiveness of the proposed method using diverse deep convolutional neural network architectures on multiple OOD detection benchmark datasets, and compare it against various state-of-the-art methods.

**Datasets and experimental setup**     Following existing literature [1, 3, 19, 32, 4, 8, 10, 9, 17, 16], we train OOD models on the training set of CIFAR-10 [37], CIFAR-100 [37] and ImageNet [38], respectively, and evaluate OOD detectors in identifying OOD test samples from other datasets, including iSUN [39], iNaturalist [40], LSUN(R) [17], LSUN(C) [17], Gaussian [17], Places365 [41], Textures [42] and SVHN [43]. ImageNet(R) [3] and Imagenet(C) [3] are also used as OOD sets for the CIFAR experiment. For the ImageNet experiment, CIFAR datasets are considered as OOD sets. These datasets are described and provided in previous studies [17, 4, 1].

For fair comparison and to aid reproducibility, we use the pretrained model for standard ResNet-34 [44], WideResNet-40 [45] and DenseNet-BC-100 [46] as the backbone network for the CIFAR experiment since they are widely used in previous OOD detection studies [1, 3, 32, 4, 8, 10, 9, 17]. Following [1], we adopt the lightweight MobileNet-v2 [47] for the ImageNet experiment. It is noteworthy that since some of the following compared models, e.g., Gram [8] and Mah [9], require prohibitively large memory for large networks with high input dimensions, we only conduct experiments on lightweight MobileNet-v2 for the ImageNet dataset. The pretrained weights for the above ResNet, WideResNet, DenseNet and MobileNet are provided by [17, 9, 1]. When training our OOD detector, we use the same normalization, augmentation and training setting as in [10, 4], where the learning rate was adjusted by cosine decay schedule [48] and the fine-tuning epoch is 10. The initial learning rates are set $10^{-3}$ and $10^{-4}$ for CIFAR and ImageNet experiments. Other hyperparameters are set as follows: batch size of 256, regularization weight $\gamma=10^{-2}$, constan $r=10$, and fixed statistical significance level $\alpha=0.05$. Constant $q$ is a detached tensor dynamically determined to linearly map the exponent from $[-200r, 0]$ to $[-5, 5]$ for ensuring numerical stability. It is worth noting that unlike many previous studies [4, 8, 10, 17, 9, 32], which adjusted hyperparameters for different datasets or networks, our method can achieve state-of-the-art performance across various experiments with fixed values of introduced hyperparameters. Moreover, we will also demonstrate the robustness of our approach to hyperparameter changes in our ablation studies. Our distribution-deviating operators $\mathbb{S}$ include random resized crop and random affine transforms realized by TorchVision library [49], and uniform noise generator (see Appendix A.4 for details).

The proposed approach is compared to the state-of-the-art methods, including Msp [16], Odin [17], Mah [9], Energy [4], Gram [8], ReAct [1] and pNML [3] for OOD detection without outlier exposure,

**Algorithm 1:** Training density-driven regularization (DDR) model for OOD detection

---

**Input:** ID training set $D_{in}$, batch size $N$, initialized/pre-trained network $f_{\theta_0}$, significance level $\alpha$, loss weight $\gamma$, constant $r$, deviating operator set $\mathbb{S}$, learning rate $\eta$ and training/fine-tuning iteration $T$.

**Output:** classification network $f_{\theta_T}$

**for** $t = 0, 1, ..., T-1$ **do**

    sample a batch of $N$ training samples $\{(X_i, Y_i)\}_{i=1}^N$ from $D_{in}$

    generate distribution-deviated samples $\widetilde{X}_i^{\mathcal{S}} = \mathcal{S}(X_i)$ for $\mathcal{S} \in \mathbb{S}$

    initialize regularized loss $\widetilde{\lambda} = 0$

    calculate logit vector $h(X_i) := f_{\theta_t}(X_i)$ and $h(\widetilde{X}_i^{\mathcal{S}}) := f_{\theta_t}(\widetilde{X}_i^{\mathcal{S}})$

    $h(X_i) \leftarrow h(X_i) - C$ and $h(\widetilde{X}_i^{\mathcal{S}}) \leftarrow h(\widetilde{X}_i^{\mathcal{S}}) - C$ where $C$ is the maximal value in all logit vectors `// ensuring numerical stability`

    define $\Omega = \{y \mid \sum_{1 \le i \le N} 1_{\{Y_i = y\}} > 0, 1 \le y \le L\}$

    **for** $y \in \Omega$ **do**

        define $n_y = \sum_{1 \le i \le N} 1_{\{Y_i = y\}}$ and $u_i^y = [\sum_{1 \le l \le L} \exp(h^l(X_i)), \exp(h^y(X_i)), 1_{\{Y_i=y\}}]^\top$

        define $\bar{\psi} := \frac{r}{N|\mathbb{S}|} \sum_{\mathcal{S} \in \mathbb{S}, 1 \le i \le N} \log \frac{\sum_{1 \le l \le L} \exp(h^l(\widetilde{X}_i^{\mathcal{S}}))}{\sum_{1 \le l \le L} \exp(h^l(X_i))}$

        calculate detached tensor $q$ by which $\bar{\psi} + q$ linearly map the original exponent value $\bar{\psi}$ in $[-200r, 0]$ to $[-5, 5]$ `// ensuring numerical stability`

        calculate $\widetilde{\Upsilon}_N^y = \sqrt{N} \left| \frac{\sum_{1 \le i \le N} \exp(h^y(X_i))}{\sum_{1 \le i \le N, 1 \le l \le L} \exp(h^l(X_i))} - \frac{n_y}{N} \right| + \exp(\bar{\psi} + q)$ `// Eq.(16)`

        calculate $\Delta_N^y = [\nabla g(\mu_N^y)]^\top \Sigma_N^y [\nabla g(\mu_N^y)]$ `// `$\mu_N^y$` and `$\Sigma_N^y$` are the sample mean and covariance matrix for `$\mu^y$` and `$\Sigma^y$` computed by `$u_i^y$` (lemma 2)`

        $\widetilde{\lambda} \leftarrow \widetilde{\lambda} + \gamma \frac{1}{|\Omega|} \max\left(\log\left([\Delta_N^y]^{-1}[\widetilde{\Upsilon}_N^y]^2/\chi_{1,\alpha}^2\right), 0\right)$ `// asymptotic test based regularization by Eq.(17)`

    calculate supervised classification loss $L_{cls}$ using $\{(\text{softmax}(f_{\theta_t}(X_i)), Y_i)\}_{i=1}^N$

    combine loss value $L_{cls} + \widetilde{\lambda}$

    update network parameter $\theta_{t+1} \leftarrow \theta_t - \eta \nabla_{\theta_t}(L_{cls} + \widetilde{\lambda})$ by SGD based optimizer

**return** $f_{\theta_T}$

---

as well as Energy [4] and Oecc [32] for OOD detection with outlier exposure. 300K Random Images[10] is used as the outlier exposure dataset to reproduce the Energy method. All compared methods are evaluated with their open-source codes and trained weights. Following existing studies [1, 3, 32, 4, 8, 10, 9, 17, 16], commonly used metrics including AUROC, AURP and FPR (at 95% TPR) are used to evaluate different OOD detectors. The evaluation code is adopted from [4]. All experiments run on the PyTorch framework [50] with Nvidia A100 and GeForce RTX 3090 GPUs, with a maximum RAM of 512GB.

**Experimental results** The experimental results on CIFAR-100 in terms of FPR95% are shown in Table 1. See Appendix A.5 for the variance of the results over ten runs with different subsets of OOD data. Compared to other state-of-the-art detectors without outlier exposure, our method achieves the best detection accuracy for diverse OOD datasets under different network architectures. The averaged FPR95% of the proposed method is significantly lower than that of other detectors without outlier exposure, and even lower than the four state-of-the-art detectors that use an external large-scale outlier exposure dataset with 80 million images. Other evaluation results in terms of AUROC and AUPR also reveal a consistent performance trend, which can be found in Appendix A.5. It is worth noting that the proposed method only utilizes classification logits for OOD detection. In contrast, the best-performed existing method, i.e., Gram, uses features across all layers to detect OODs and requires calculating huge gram matrices dynamically using all latent features of training samples, preventing end-to-end real-time prediction and becoming intractable for deep networks

Table 1: Performance comparison (in terms of FPR95%) of various methods in CIFAR-100 (ID dataset) under three commonly-used networks. The best and second-best models without accessing external outlier datasets are coloured red and blue, respectively.

| | OOD dataset | w/o outlier exposure (FPR95% ↓) Msp/Odin/Mah/Energy/Gram/Energy+ReAct/pNML/Odin+pNML/Ours | w/ outlier exposure (FPR95% ↓) Oecc/Energy/Oecc+pNML |
|---|---|---|---|
| **CIFAR-100 (ResNet34)** | iSUN | 83.10 / 55.65 / 67.20 / 80.91 / 5.01 / 80.00 / 73.70 / 55.10 / 4.93 | 2.70 / 54.76 / 2.14 |
| | LSUN(R) | 80.75 / 53.15 / 68.15 / 77.44 / 2.92 / 77.75 / 70.75 / 54.90 / 3.19 | 1.72 / 54.07 / 1.01 |
| | LSUN(C) | 81.30 / 55.35 / 47.21 / 82.93 / 33.76 / 80.25 / 70.60 / 47.75 / 7.99 | 19.46 / 37.20 / 10.36 |
| | CIFAR-10 | 80.95 / 85.65 / 81.57 / 81.50 / 88.12 / 83.05 / 80.00 / 81.15 / 85.08 | 85.06 / 83.31 / 79.05 |
| | ImageNet(R) | 79.00 / 52.55 / 63.96 / 75.70 / 5.17 / 75.95 / 67.95 / 53.75 / 4.47 | 4.60 / 62.30 / 4.12 |
| | ImageNet(C) | 77.40 / 57.90 / 58.55 / 73.43 / 11.68 / 72.85 / 67.00 / 53.15 / 4.07 | 9.32 / 59.27 / 8.53 |
| | Gaussian | 100.0 / 95.35 / 27.90 / 100.0 / 0.00 / 100.0 / 85.25 / 32.50 / 0.00 | 0.00 / 0.01 / 0.00 |
| | Places365 | 84.55 / 84.10 / 59.75 / 85.68 / 46.89 / 85.00 / 71.25 / 61.80 / 21.12 | 31.76 / 41.05 / 19.74 |
| | Textures | 80.80 / 80.15 / 56.85 / 79.96 / 33.43 / 78.50 / 61.35 / 53.80 / 18.50 | 24.66 / 43.64 / 16.38 |
| | iNaturalist | 73.00 / 71.50 / 58.36 / 73.34 / 44.68 / 71.75 / 57.85 / 57.85 / 31.66 | 46.22 / 68.57 / 47.49 |
| | SVHN | 79.70 / 38.60 / 31.42 / 82.13 / 19.99 / 78.75 / 48.40 / 26.30 / 1.61 | 13.09 / 40.57 / 10.23 |
| | **average** | 81.87 / 66.36 / 56.45 / 81.18 / 26.51 / 80.35 / 68.55 / 52.55 / **16.60** | 21.69 / 49.52 / 18.10 |
| **CIFAR-100 (WideResNet)** | iSUN | 86.75 / 63.60 / 36.73 / 78.80 / 5.23 / 82.70 / 73.45 / 73.45 / 2.34 | 7.93 / 55.97 / 3.01 |
| | LSUN(R) | 85.75 / 60.50 / 34.66 / 79.40 / 2.97 / 79.25 / 77.55 / 77.55 / 1.08 | 5.41 / 53.28 / 1.90 |
| | LSUN(C) | 80.85 / 52.10 / 45.19 / 71.85 / 37.29 / 57.85 / 86.95 / 61.85 / 5.67 | 18.20 / 29.25 / 6.70 |
| | CIFAR-10 | 81.85 / 85.90 / 90.45 / 77.85 / 90.41 / 87.90 / 98.45 / 97.60 / 88.49 | 88.52 / 84.75 / 90.36 |
| | ImageNet(R) | 84.90 / 63.95 / 34.06 / 78.95 / 6.01 / 83.60 / 71.55 / 71.55 / 3.34 | 12.04 / 66.33 / 7.72 |
| | ImageNet(C) | 82.55 / 65.30 / 46.77 / 71.05 / 12.98 / 76.80 / 79.40 / 79.90 / 3.48 | 15.84 / 56.36 / 7.81 |
| | Gaussian | 99.60 / 0.05 / 0.00 / 67.25 / 0.00 / 47.55 / 0.00 / 0.00 / 0.00 | 0.00 / 0.00 / 0.00 |
| | Places365 | 83.80 / 78.80 / 52.73 / 81.05 / 46.27 / 72.50 / 82.45 / 62.60 / 11.27 | 23.26 / 36.49 / 9.68 |
| | Textures | 81.60 / 74.05 / 39.05 / 78.45 / 35.98 / 66.00 / 75.40 / 58.75 / 12.40 | 19.62 / 38.03 / 8.10 |
| | iNaturalist | 76.15 / 70.35 / 59.72 / 71.40 / 47.86 / 70.75 / 85.40 / 73.40 / 17.10 | 38.38 / 62.70 / 24.92 |
| | SVHN | 77.30 / 42.45 / 33.27 / 85.40 / 19.26 / 76.00 / 74.85 / 57.05 / 1.68 | 9.43 / 37.50 / 3.64 |
| | **average** | 83.74 / 59.73 / 42.97 / 76.50 / 27.66 / 73.15 / 73.22 / 64.88 / **13.35** | 21.69 / 47.33 / 14.89 |
| **CIFAR-100 (DenseNet)** | iSUN | 83.85 / 62.10 / 16.48 / 82.45 / 3.99 / 40.35 / 18.40 / 16.20 / 5.66 | 3.21 / 76.09 / 0.92 |
| | LSUN(R) | 82.75 / 58.55 / 13.48 / 77.45 / 2.38 / 34.45 / 16.40 / 14.40 / 4.20 | 1.98 / 73.39 / 0.57 |
| | LSUN(C) | 73.40 / 42.25 / 74.92 / 47.05 / 34.70 / 45.80 / 35.10 / 35.10 / 8.35 | 26.35 / 33.94 / 16.89 |
| | CIFAR-10 | 80.90 / 84.20 / 98.86 / 82.05 / 87.70 / 81.90 / 97.70 / 97.60 / 92.46 | 89.58 / 86.30 / 91.82 |
| | ImageNet(R) | 81.90 / 56.95 / 16.56 / 76.85 / 4.15 / 39.70 / 13.55 / 11.95 / 2.83 | 3.85 / 75.50 / 1.19 |
| | ImageNet(C) | 75.45 / 46.75 / 35.92 / 58.65 / 10.92 / 33.30 / 22.75 / 21.90 / 3.53 | 8.09 / 60.05 / 3.40 |
| | Gaussian | 100.0 / 100.0 / 0.00 / 100.0 / 0.00 / 100.0 / 0.00 / 0.00 / 0.00 | 0.00 / 100.0 / 0.00 |
| | Places365 | 76.15 / 51.80 / 81.42 / 58.20 / 45.56 / 53.00 / 36.35 / 36.35 / 15.18 | 41.32 / 46.77 / 32.74 |
| | Textures | 75.15 / 56.50 / 59.34 / 58.85 / 36.38 / 46.38 / 25.90 / 25.90 / 14.07 | 31.87 / 48.95 / 23.20 |
| | iNaturalist | 70.30 / 51.80 / 79.96 / 48.90 / 48.61 / 43.05 / 45.25 / 44.85 / 22.50 | 48.27 / 64.23 / 44.94 |
| | SVHN | 73.75 / 44.30 / 49.71 / 65.95 / 10.52 / 48.55 / 20.50 / 20.50 / 3.16 | 11.63 / 69.06 / 9.72 |
| | **average** | 79.42 / 59.56 / 47.88 / 68.76 / 25.90 / 51.58 / 30.17 / 29.52 / **15.63** | 24.20 / 66.75 / 20.49 |

trained on large datasets (See Appendix A.10 for computational cost comparision with our approach). Due to the page limit, the reader is referred to Appendix A.6 for detailed results of the CIFAR-10 experiment, in which our method also achieves the best detection accuracy in both FPR95%, AUROC and AUPR compared to other existing methods without outlier exposure. The performance of the proposed method is also comparable to the state-of-the-art OOD detectors that require large-scale additional outlier exposure datasets. Tables in Appendix A.7 also provide the experimental results of FPR95%, AUROC and AUPR values on ImageNet. Due to the lack of large-scale high-resolution outlier exposure datasets, which should theoretically be much larger than the ID dataset, none of the outlier-exposure OOD detectors had been evaluated on the ImageNet benchmark in literature. Therefore, only OOD detectors without requiring outlier exposure datasets are evaluated. It can be observed that the proposed method also achieves the highest averaged detection accuracy on the ImageNet benchmark. It is noteworthy that our method performs relatively poorly on the iNaturalists and Places365 compared to other OOD datasets, as is also the case for other traditional methods. It can be because the ImageNet includes semantically overlapping objects as the iNaturalists and Places365, as is shown in Appendix A.9, even though we have used the conceptually disjoint subsets of Textures and iNaturalists provided by the previous study [1] for model evaluation.

**Ablation studies** We conduct ablation experiments on CIFAR-100 using the WideResNet network. Table 2 verifies the effectiveness of each component, i.e., density consistency regularization and contrastive distribution regularization. We remark that the baseline model without regularization

Table 2: Contribution of the proposed regularization terms to OOD detection performance. DCR: density consistency regularization. CDR: contrastive distribution regularization. The result is the average value across all OOD datasets.

| in-dist (model) | DCR | CDR | mean AUROC (↑) | mean AUPR (↑) | mean FPR95% (↓) |
|---|---|---|---|---|---|
| CIFAR-100 (WideResNet) | - | - | 80.05 | 95.11 | 77.04 |
| | ✓ | - | 90.84 | 97.88 | 41.59 |
| | ✓ | ✓ | 96.09 | 98.94 | 13.35 |

Table 3: Ablation studies on CIFAR-100 (ID dataset) with WideResNet demonstrating the proposed method's robustness to hyperparameters. The result is the average value across all OOD datasets.

| ablation studies | parameters | mean AUROC (↑) | mean AUPR (↑) | mean FPR95% (↓) |
|---|---|---|---|---|
| batch size | 128 | 96.14 | 98.94 | 13.43 |
| | 256 | 96.09 | 98.94 | 13.35 |
| | 512 | 96.08 | 98.93 | 13.63 |
| regularization weight $\gamma$ | 0.005 | 96.02 | 98.94 | 14.08 |
| | 0.01 | 96.09 | 98.94 | 13.35 |
| | 0.02 | 96.15 | 98.91 | 12.92 |
| constant $r$ | 5 | 96.02 | 98.94 | 14.03 |
| | 10 | 96.09 | 98.94 | 13.29 |
| | 20 | 96.12 | 98.91 | 12.87 |

corresponds theoretically to the Energy model with no outlier exposure and temperature coefficient of $T = 1$. Table 3 verifies the robustness of the proposed method to hyperparameters, including batch size, regularization weight $\gamma$ and constant $r$. The model performance is quite consistent when increasing or decreasing the default values by 100% and 50%, respectively. More detailed results on different OOD datasets and more hyperparamter choices can be found in Appendix A.8. Please note that unlike many previous approaches involving hyperparameter tuning [4, 8, 10, 17, 9, 32], our method uses fixed hyperparameters for OOD detection across different ID datasets, OOD datasets and networks. It further confirms the robustness of the proposed method.

## 5    Conclusion

In this paper, we propose a theoretical probabilistic framework for OOD detection, which derives analytical density regularization by investigating the inherent consistent and contrastive characteristics of the ID sample density to calibrate and build an accurate sample density estimator. Unlike many previous studies, our method neither relies on unproven assumptions about the latent feature distribution nor requires additional outlier exposure datasets to calibrate ID density estimates or tune hyperparameters. A simple yet effective computational algorithm is also developed, which can be used for any pretrained classification network. Extensive evaluations on benchmark datasets show that the proposed approach significantly outperforms existing state-of-the-art OOD detectors. Furthermore, our method even achieves comparable or better performance than the approaches requiring external large-scale outlier exposure datasets. A limitation of our approach is that the fixed hyperparameters may not be optimal for a specific task, although it enables our fairly robust model to achieve state-of-the-art performances across benchmarks. We leave it for future work to investigate adaptively setting hyperparameters with the ID dataset. Additionally, since our approach requires fine-tuning, this can lead to the problem that there exists a discrepancy in classification probabilities between the original network and the fine-tuned network. This problem can be avoided using the original and the fine-tuned networks for the classification and OOD detection tasks, respectively. However, it can double the inference time and computation load. Finally, we believe our accurate sample density estimator can also guide future studies in related machine learning fields, such as active learning, open-world classification, semi-supervised learning, etc.

## Acknowledgments and Disclosure of Funding

This work is supported in part by National Key Research and Development Program of China (2021YFF1200800)

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
