# Supplementary Material
# Density-driven Regularization for Out-of-distribution Detection

## A.1 Proof of lemma 1

**Lemma 1.** *If Eq.(3) holds, then*

$$\frac{\sum_{1 \le i \le N} \exp\left(h^y\left(X_i\right)\right)}{\sum_{1 \le i \le N} \sum_{1 \le l \le L} \exp\left(h^l\left(X_i\right)\right)} - \frac{n_y}{N} \to_{a.s.} 0; \forall 1 \le y \le L$$

*Proof.* First, according to Eq.(7), the left-hand side of the following equation converges to $Z$ almost surely (a.s.) and in probability as $N \to \infty$.

$$\frac{N}{n_y} \frac{\sum_{1 \le i \le N} \exp\left(h^y\left(X_i\right)\right)}{N} \approx \frac{\sum_{1 \le i \le N} \sum_{1 \le l \le L} \exp\left(h^l\left(X_i\right)\right)}{N}; \forall 1 \le y \le L$$

Next, according to the law of large number, the right-hand side also converges to $Z$ a.s. and in probability as $N \to \infty$ by Eq.(3). Specifically,

$$\frac{\sum\limits_{1 \le i \le N} \sum\limits_{1 \le l \le L} \exp\left(h^l\left(X_i\right)\right)}{N} \to_{a.s.} \int \sum_{1 \le l \le L} \exp\left(h^l\left(X\right)\right) d\nu_X = \iint \exp\left(h^y\left(X\right)\right) d\pi_Y d\nu_X = Z$$

By simple transformation of the first equation, we obtain the lemma. $\qquad\square$

## A.2 Proof of lemma 2

**Lemma 2.** *If Eq.(3) holds, then*

$$\Upsilon_N^y = \sqrt{N} \left( \frac{\sum_{1 \le i \le N} \exp\left(h^y\left(X_i\right)\right)}{\sum_{1 \le i \le N} \sum_{1 \le l \le L} \exp\left(h^l\left(X_i\right)\right)} - \frac{n_y}{N} \right) \to_d \mathcal{N}\left(0, \Delta^y\right)$$

*where $\Delta^y \triangleq \left[\nabla g\left(\mu^y\right)\right]^\top \Sigma^y \left[\nabla g\left(\mu^y\right)\right]$ and $g\left([x, y, z]^\top\right) = y/x - z$. $\mu^y$ and $\Sigma^y$ represent the mean and covariance matrix for random vector $u^y = \left[\sum_{1 \le l \le L} \exp\left(h^l\left(X\right)\right), \exp\left(h^y\left(X\right)\right), 1_{\{Y=y\}}\right]^\top$.*

*Proof.* First consider the random vector

$$u_i^y = \begin{bmatrix} \sum\limits_{1 \le l \le L} \exp\left(h^l\left(X_i\right)\right) \\ \exp\left(h^y\left(X_i\right)\right) \\ 1_{\{Y_i=y\}} \end{bmatrix} \quad \text{and} \quad U_N^y = \sum_{1 \le i \le N} u_i^y.$$

36th Conference on Neural Information Processing Systems (NeurIPS 2022).

For independent and identically distributed (i.i.d.) random vector $u_i^y$, the multidimensional central limit theorem (CLT) gives

$$\sqrt{N}\left(\frac{U_N^y}{N} - \mu^y\right) = \frac{1}{\sqrt{N}}\left(U_N^y - N\mu^y\right) \to_d \mathcal{N}\left(0, \Sigma^y\right)$$

where $\mu^y$ and $\Sigma^y$ are the mean and covariance matrix for $u_i^y$, respectively. Let

$$g\left([x, y, z]\right) = y/x - z.$$

Then, by using delta-method [1], we obtain

$$\sqrt{N}\left(g\left(\frac{U_N^y}{N}\right) - g\left(\mu^y\right)\right) \to_d \mathcal{N}\left(0, \left[\nabla g\left(\mu^y\right)\right]^\top \Sigma^y \left[\nabla g\left(\mu^y\right)\right]\right)$$

where

$$g\left(\frac{U_N^y}{N}\right) = \frac{\sum_{1 \le i \le N} \exp\left(h^y\left(X_i\right)\right)}{\sum_{1 \le i \le N, 1 \le l \le L} \exp\left(h^l\left(X_i\right)\right)} - \frac{\sum_{1 \le i \le N} 1_{\{Y_i = y\}}}{N}$$

and $g\left(\mu^y\right) = 0$ based on Eq.(3). Hence, we have

$$\sqrt{N}\left(\frac{\sum_{1 \le i \le N} \exp\left(h^y\left(X_i\right)\right)}{\sum_{1 \le i \le N, 1 \le l \le L} \exp\left(h^l\left(X_i\right)\right)} - \frac{n_y}{N}\right) \to_d \mathcal{N}\left(0, \left[\nabla g\left(\mu^y\right)\right]^\top \Sigma^y \left[\nabla g\left(\mu^y\right)\right]\right).$$

$\square$

## A.3 Equivalent regularization with normalized logit

To increase numerical stability, we normalize the classification logit by subtracting the maximal logit before calculating the regularization term $\widetilde{\lambda}$ in Algorithm 1. Here, in the following proposition, we prove that our normalized logit will theoretically produce the same results as the original logit.

**Proposition 1.** *subtracting a fixed constant from the classification logits leads to the same consistency regularization term.*

*Proof.* For $\forall C \in R$, subtracting $-C$ from the original logit $h^y\left(X_i\right)$ yields the following shifted logits $^*h^y\left(X_i\right)$:

$$^*h^y\left(X_i\right) = h^y\left(X_i\right) + C$$

The corresponding $^*\Upsilon_N^y$ and $^*\Delta_N^y$ for $^*h^y$ are calculated as below. First,

$$
\begin{aligned}
^*\Upsilon_N^y &= \sqrt{N}\left(\frac{\sum_{1 \le i \le N} e^{^*h^y(X_i)}}{\sum_{1 \le i \le N, 1 \le l \le L} e^{^*h^l(X_i)}} - \frac{n_y}{N}\right) \\
&= \sqrt{N}\left(\frac{\sum_{1 \le i \le N} e^{h^y(X_i)+C}}{\sum_{1 \le i \le N, 1 \le l \le L} e^{h^l(X_i)+C}} - \frac{n_y}{N}\right) = \Upsilon_N^y
\end{aligned}
$$

and, similarly, it can be shown that $^*\widetilde{\Upsilon}_N^y = \widetilde{\Upsilon}_N^y$. Then, by the definition of $u_i^y$, we have

$$
^*u_i^y = \begin{bmatrix} \sum_{1 \le l \le L} e^{^*h^l(X_i)} \\ e^{^*h^y(X_i)} \\ 1_{\{Y_i = y\}} \end{bmatrix} = \begin{bmatrix} e^C \cdot \sum_{1 \le l \le L} e^{h^l(X_i)} \\ e^C \cdot e^{h^y(X_i)} \\ 1_{\{Y_i = y\}} \end{bmatrix} = \begin{bmatrix} e^C (u_i^y)_1 \\ e^C (u_i^y)_2 \\ (u_i^y)_3 \end{bmatrix}
$$

where subscripts $1, 2$ and $3$ denote the corresponding element in $u_i^y$. Next, by the definition of sample mean ${}^*\mu_N^y$ and sample variance ${}^*\Sigma_N^y$ for ${}^*u_i^y$, we derive:

$$
\begin{aligned}
{}^*\Delta_N^y &= \left[\nabla g\left({}^*\mu^y\right)\right]^\top {}^*\Sigma^y \left[\nabla g\left({}^*\mu^y\right)\right] \\
&= \begin{bmatrix} -\frac{\left({}^*\mu_N^y\right)_2}{\left({}^*\mu_N^y\right)_1^2} & \frac{1}{\left({}^*\mu_N^y\right)_1} & -1 \end{bmatrix}
\begin{bmatrix} \left({}^*\Sigma_N^y\right)_{11} & \left({}^*\Sigma_N^y\right)_{12} & \left({}^*\Sigma_N^y\right)_{13} \\ \left({}^*\Sigma_N^y\right)_{12} & \left({}^*\Sigma_N^y\right)_{22} & \left({}^*\Sigma_N^y\right)_{23} \\ \left({}^*\Sigma_N^y\right)_{13} & \left({}^*\Sigma_N^y\right)_{23} & \left({}^*\Sigma_N^y\right)_{33} \end{bmatrix}
\begin{bmatrix} -\frac{\left({}^*\mu_N^y\right)_2}{\left({}^*\mu_N^y\right)_1^2} \\ \frac{1}{\left({}^*\mu_N^y\right)_1} \\ -1 \end{bmatrix} \\
&= \begin{bmatrix} -\frac{1}{e^C}\frac{\left(\mu_N^y\right)_2}{\left(\mu_N^y\right)_1^2} & \frac{1}{e^C}\frac{1}{\left(\mu_N^y\right)_1} & -1 \end{bmatrix}
\begin{bmatrix} e^{2C}\left(\Sigma_N^y\right)_{11} & e^{2C}\left(\Sigma_N^y\right)_{12} & e^{C}\left(\Sigma_N^y\right)_{13} \\ e^{2C}\left(\Sigma_N^y\right)_{12} & e^{2C}\left(\Sigma_N^y\right)_{22} & e^{C}\left(\Sigma_N^y\right)_{23} \\ e^{C}\left(\Sigma_N^y\right)_{13} & e^{C}\left(\Sigma_N^y\right)_{23} & \left(\Sigma_N^y\right)_{33} \end{bmatrix}
\begin{bmatrix} -\frac{1}{e^C}\frac{\left(\mu_N^y\right)_2}{\left(\mu_N^y\right)_1^2} \\ \frac{1}{e^C}\frac{1}{\left(\mu_N^y\right)_1} \\ -1 \end{bmatrix} \\
&= \begin{bmatrix} -\frac{\left(\mu_N^y\right)_2}{\left(\mu_N^y\right)_1^2} & \frac{1}{\left(\mu_N^y\right)_1} & -1 \end{bmatrix}
\begin{bmatrix} \left(\Sigma_N^y\right)_{11} & \left(\Sigma_N^y\right)_{12} & \left(\Sigma_N^y\right)_{13} \\ \left(\Sigma_N^y\right)_{12} & \left(\Sigma_N^y\right)_{22} & \left(\Sigma_N^y\right)_{23} \\ \left(\Sigma_N^y\right)_{13} & \left(\Sigma_N^y\right)_{23} & \left(\Sigma_N^y\right)_{33} \end{bmatrix}
\begin{bmatrix} -\frac{\left(\mu_N^y\right)_2}{\left(\mu_N^y\right)_1^2} \\ \frac{1}{\left(\mu_N^y\right)_1} \\ -1 \end{bmatrix} \\
&= \left[\nabla g\left(\mu^y\right)\right]^\top \Sigma^y \left[\nabla g\left(\mu^y\right)\right] = \Delta_N^y.
\end{aligned}
$$

Finally, we obtain

$$
\left[{}^*\widetilde{\Upsilon}_N^y\right]^\top [{}^*\Delta_N^y]^{-1} \left[{}^*\widetilde{\Upsilon}_N^y\right] = [\Upsilon_N^y]^\top [\Delta_N^y]^{-1} [\Upsilon_N^y]
$$

It shows that any shifted logit theoretically derives the same density regularization term. $\qquad\square$

## A.4   Distribution-deviating operator

The random affine and resized crop transforms from the TorchVision library [2] are used as the distribution-deviating (DD) operators, including four specific transformations: $\mathcal{S}_1$, random affine with only rotation (angle of $60°$~$300°$ avoiding similarity); $\mathcal{S}_2$, random affine with both rotation and scaling (same rotation angle with a scaling factor of 1/5~5); $\mathcal{S}_3$, random resized crop (area ratio of 0~0.6 and aspect ratio of 1/9~9); and $\mathcal{S}_4$, the composed transformation of $\mathcal{S}_2$ and $\mathcal{S}_3$ (allowing cropping rotated images). DD set $\mathbb{S}$ also contains a random uniform generator $\mathcal{S}_5$ that generates uniform noises of range 0~255 as the random input image.

## A.5   FPR95%, AUROC and AUPR for CIFAR-100 experiment

Tables 1~3, 4~6 and 7 ~9 below show the model comparisons in terms of AUROC and AUPR for the CIFAR-100 experiments, respectively.

Table 1: Performance comparison (mean±std, in terms of FPR95%) of various methods in CIFAR-100 (ID dataset) under ResNet34. The best and second-best models without accessing external outlier datasets are coloured red and blue, respectively.

| OOD dataset | w/o outlier exposure (FPR95% ↓) | | | | | | | | | w/ outlier exposure (FPR95% ↓) | | |
|---|---|---|---|---|---|---|---|---|---|---|---|---|
| | Msp | Odin | Mah | Energy | Gram | Energy+ReAct | pNML | Odin+pNML | Ours | Oecc | Energy | Oecc+pNML |
| **CIFAR-100 (ResNet34)** | | | | | | | | | | | | |
| iSUN | 83.10 ±0.52 | 55.65 ±1.09 | 67.20 ±0.66 | 80.91 ±0.85 | 5.01 ±0.38 | 80.00 ±0.81 | 73.70 ±0.74 | 55.10 ±0.79 | 4.93 ±0.41 | 2.70 ±0.23 | 54.76 ±0.60 | 2.14 ±0.27 |
| LSUN(R) | 80.75 ±0.73 | 53.15 ±0.65 | 68.15 ±1.15 | 77.44 ±0.72 | 2.92 ±0.32 | 77.75 ±0.78 | 70.75 ±0.56 | 54.90 ±0.70 | 3.19 ±0.29 | 1.72 ±0.24 | 54.07 ±0.78 | 1.01 ±0.21 |
| LSUN(C) | 81.30 ±0.67 | 55.35 ±0.65 | 47.21 ±0.67 | 82.93 ±0.75 | 33.76 ±0.84 | 80.25 ±0.82 | 70.60 ±0.71 | 47.75 ±0.75 | 7.99 ±0.44 | 19.46 ±0.75 | 37.20 ±0.98 | 10.36 ±0.63 |
| CIFAR-10 | 80.95 ±0.98 | 85.65 ±0.65 | 81.57 ±0.58 | 81.50 ±0.68 | 88.12 ±0.35 | 83.05 ±0.73 | 80.00 ±0.57 | 81.15 ±0.73 | 85.08 ±0.41 | 85.06 ±0.60 | 83.31 ±0.75 | 79.05 ±0.74 |
| ImageNet(R) | 79.00 ±0.87 | 52.55 ±1.00 | 63.96 ±1.05 | 75.70 ±0.75 | 5.17 ±0.44 | 75.95 ±0.68 | 67.95 ±0.77 | 53.75 ±0.74 | 4.47 ±0.37 | 4.60 ±0.36 | 62.30 ±1.00 | 4.12 ±0.35 |
| ImageNet(C) | 77.40 ±0.91 | 57.90 ±0.76 | 58.55 ±0.63 | 73.43 ±0.66 | 11.68 ±0.50 | 72.85 ±0.73 | 67.00 ±0.37 | 53.15 ±0.48 | 4.07 ±0.36 | 9.32 ±0.65 | 59.27 ±1.00 | 8.53 ±0.52 |
| Gaussian | 100.0 ±0.00 | 95.35 ±0.33 | 27.90 ±0.86 | 100.0 ±0.00 | 0.00 ±0.00 | 100.0 ±0.00 | 85.25 ±0.67 | 32.50 ±1.02 | 0.00 ±0.00 | 0.00 ±0.00 | 0.01 ±0.02 | 0.00 ±0.00 |
| Places365 | 84.55 ±0.66 | 84.10 ±0.66 | 59.75 ±0.95 | 85.68 ±0.79 | 46.89 ±0.91 | 85.00 ±0.72 | 71.25 ±0.69 | 61.80 ±0.60 | 21.12 ±0.98 | 31.76 ±0.87 | 41.05 ±0.57 | 19.74 ±0.91 |
| Textures | 80.80 ±0.77 | 80.15 ±0.64 | 56.85 ±0.64 | 79.96 ±0.57 | 33.43 ±0.70 | 78.50 ±0.66 | 61.35 ±0.68 | 53.80 ±0.81 | 18.50 ±0.57 | 24.66 ±0.51 | 43.64 ±0.60 | 16.38 ±0.57 |
| iNaturalist | 73.00 ±0.76 | 71.50 ±0.73 | 58.36 ±0.84 | 73.34 ±1.05 | 44.68 ±0.61 | 71.75 ±0.95 | 57.85 ±0.66 | 57.85 ±1.29 | 31.66 ±0.92 | 46.22 ±0.62 | 68.57 ±0.81 | 47.49 ±1.04 |
| SVHN | 79.70 ±0.58 | 38.60 ±1.00 | 31.42 ±1.14 | 82.13 ±0.99 | 19.99 ±0.87 | 78.75 ±0.95 | 48.40 ±1.25 | 26.30 ±1.45 | 1.61 ±0.29 | 13.09 ±0.74 | 40.57 ±1.33 | 10.23 ±0.77 |
| **average** | 81.87 | 66.36 | 56.45 | 81.18 | 26.51 | 80.35 | 68.55 | 52.55 | 16.60 | 21.69 | 49.52 | 18.10 |

Table 2: Performance comparison (mean±std, in terms of FPR95%) of various methods in CIFAR-100 (ID dataset) under WideResNet. The best and second-best models without accessing external outlier datasets are coloured red and blue, respectively.

| OOD dataset | w/o outlier exposure (FPR95% ↓) | | | | | | | | | w/ outlier exposure (FPR95% ↓) | | |
|---|---|---|---|---|---|---|---|---|---|---|---|---|
| | Msp | Odin | Mah | Energy | Gram | Energy+ReAct | pNML | Odin+pNML | Ours | Oecc | Energy | Oecc+pNML |
| **CIFAR-100 (WideResNet)** | | | | | | | | | | | | |
| iSUN | 86.75 ±0.43 | 63.60 ±0.64 | 36.73 ±0.82 | 78.80 ±0.66 | 5.23 ±0.28 | 82.70 ±0.57 | 73.45 ±0.77 | 73.45 ±0.59 | 2.34 ±0.32 | 7.93 ±0.64 | 55.97 ±1.37 | 3.01 ±0.33 |
| LSUN(R) | 85.75 ±0.64 | 60.50 ±1.10 | 34.66 ±0.78 | 79.40 ±0.72 | 2.97 ±0.29 | 82.95 ±0.87 | 77.55 ±0.85 | 77.55 ±0.59 | 1.08 ±0.21 | 5.41 ±0.29 | 53.28 ±0.78 | 1.90 ±0.27 |
| LSUN(C) | 80.85 ±0.49 | 52.10 ±1.12 | 45.19 ±0.87 | 71.85 ±0.90 | 37.29 ±1.17 | 57.85 ±0.60 | 86.95 ±0.81 | 61.85 ±0.40 | 5.67 ±0.62 | 18.20 ±0.86 | 29.25 ±0.76 | 6.70 ±0.67 |
| CIFAR-10 | 81.85 ±0.62 | 85.90 ±0.53 | 90.45 ±0.96 | 77.85 ±0.75 | 90.41 ±0.56 | 87.90 ±0.68 | 98.45 ±0.29 | 97.60 ±0.30 | 88.49 ±0.62 | 88.52 ±0.56 | 84.75 ±0.34 | 90.36 ±0.70 |
| ImageNet(R) | 84.90 ±0.97 | 63.95 ±0.92 | 34.06 ±0.69 | 78.95 ±0.58 | 6.01 ±0.28 | 83.60 ±0.61 | 71.55 ±0.85 | 71.55 ±0.78 | 3.34 ±0.24 | 12.04 ±0.57 | 66.33 ±0.73 | 7.72 ±0.58 |
| ImageNet(C) | 82.55 ±0.80 | 65.30 ±0.86 | 46.77 ±0.70 | 71.05 ±0.73 | 12.98 ±0.27 | 76.80 ±0.71 | 79.40 ±0.45 | 79.90 ±0.74 | 3.48 ±0.32 | 15.84 ±0.65 | 56.36 ±1.20 | 7.81 ±0.38 |
| Gaussian | 99.60 ±0.09 | 0.05 ±0.00 | 0.00 ±0.00 | 67.25 ±0.63 | 0.00 ±0.00 | 47.55 ±0.68 | 0.00 ±0.00 | 0.00 ±0.00 | 0.00 ±0.00 | 0.00 ±0.00 | 0.00 ±0.00 | 0.00 ±0.00 |
| Places365 | 83.80 ±0.61 | 78.80 ±0.99 | 52.73 ±0.94 | 81.05 ±1.01 | 46.27 ±0.67 | 72.50 ±0.59 | 82.45 ±0.22 | 62.60 ±1.00 | 11.27 ±0.61 | 23.26 ±0.72 | 36.49 ±1.14 | 9.68 ±0.60 |
| Textures | 81.60 ±0.46 | 74.05 ±1.08 | 39.05 ±0.76 | 78.45 ±0.88 | 35.98 ±0.68 | 66.00 ±0.71 | 75.40 ±0.91 | 58.75 ±0.81 | 12.40 ±0.70 | 19.62 ±0.68 | 38.03 ±1.36 | 8.10 ±0.63 |
| iNaturalist | 76.15 ±0.75 | 70.35 ±1.02 | 59.72 ±1.00 | 71.40 ±0.84 | 47.86 ±0.90 | 70.75 ±0.87 | 85.40 ±0.75 | 73.40 ±1.25 | 17.10 ±0.57 | 38.38 ±1.18 | 62.70 ±0.93 | 24.92 ±0.99 |
| SVHN | 77.30 ±0.96 | 42.45 ±0.88 | 33.27 ±1.00 | 85.40 ±0.62 | 19.26 ±0.69 | 76.00 ±0.96 | 74.85 ±0.76 | 57.05 ±0.57 | 1.68 ±0.31 | 9.43 ±0.75 | 37.50 ±0.94 | 3.64 ±0.41 |
| **average** | 83.74 | 59.73 | 42.97 | 76.50 | 27.66 | 73.15 | 73.22 | 64.88 | 13.35 | 21.69 | 47.33 | 14.89 |

Table 3: Performance comparison (mean±std, in terms of FPR95%) of various methods in CIFAR-100 (ID dataset) under DenseNet. The best and second-best models without accessing external outlier datasets are coloured red and blue, respectively.

| OOD dataset | w/o outlier exposure (FPR95% ↓) | | | | | | | | | w/ outlier exposure (FPR95% ↓) | | |
| --- | --- | --- | --- | --- | --- | --- | --- | --- | --- | --- | --- | --- |
| | Msp/Odin/Mah/Energy/Gram/ Energy+ReAct/pNML/Odin+pNML/Ours | | | | | | | | | Oecc/Energy/ Oecc+pNML | | |
| iSUN | 83.85 | 62.10 | 16.48 | 82.45 | 3.99 | 40.35 | 18.40 | 16.20 | 5.66 | 3.21 | 76.09 | 0.92 |
| | ±0.64 | ±0.86 | ±0.55 | ±0.68 | ±0.23 | ±0.80 | ±0.68 | ±0.80 | ±0.48 | ±0.25 | ±0.77 | ±0.17 |
| LSUN(R) | 82.75 | 58.55 | 13.48 | 77.45 | 2.38 | 34.45 | 16.40 | 14.40 | 4.20 | 1.98 | 73.39 | 0.57 |
| | ±0.71 | ±0.45 | ±0.78 | ±0.60 | ±0.19 | ±1.02 | ±0.64 | ±0.66 | ±0.53 | ±0.26 | ±0.69 | ±0.18 |
| LSUN(C) | 73.40 | 42.25 | 74.92 | 47.05 | 34.70 | 45.80 | 35.10 | 35.10 | 8.35 | 26.35 | 33.94 | 16.89 |
| | ±1.04 | ±0.90 | ±0.63 | ±0.74 | ±1.06 | ±1.02 | ±0.88 | ±0.78 | ±0.59 | ±0.96 | ±0.94 | ±0.84 |
| CIFAR-10 | 80.90 | 84.20 | 98.86 | 82.05 | 87.70 | 81.90 | 97.70 | 97.60 | 92.46 | 89.58 | 86.30 | 91.82 |
| | ±0.71 | ±0.48 | ±0.25 | ±0.42 | ±0.65 | ±0.48 | ±0.30 | ±0.18 | ±0.48 | ±0.43 | ±0.22 | ±0.50 |
| ImageNet(R) | 81.90 | 56.95 | 16.56 | 76.85 | 4.15 | 39.70 | 13.55 | 11.95 | 2.83 | 3.85 | 75.50 | 1.19 |
| | ±0.78 | ±1.23 | ±0.62 | ±0.64 | ±0.12 | ±0.51 | ±0.71 | ±0.52 | ±0.31 | ±0.22 | ±0.52 | ±0.17 |
| ImageNet(C) | 75.45 | 46.75 | 35.92 | 58.65 | 10.92 | 33.30 | 22.75 | 21.90 | 3.53 | 8.09 | 60.05 | 3.40 |
| | ±0.77 | ±0.96 | ±1.14 | ±0.63 | ±0.44 | ±1.17 | ±0.50 | ±0.63 | ±0.33 | ±0.36 | ±0.64 | ±0.36 |
| Gaussian | 100.0 | 100.0 | 0.00 | 100.0 | 0.00 | 100.0 | 0.00 | 0.00 | 0.00 | 0.00 | 100.0 | 0.00 |
| | ±0.00 | ±0.08 | ±0.00 | ±0.00 | ±0.00 | ±0.00 | ±0.00 | ±0.00 | ±0.00 | ±0.00 | ±0.00 | ±0.00 |
| Places365 | 76.15 | 51.80 | 81.42 | 58.20 | 45.56 | 53.00 | 36.35 | 36.35 | 15.18 | 41.32 | 46.77 | 32.74 |
| | ±1.08 | ±0.78 | ±0.62 | ±0.80 | ±0.84 | ±1.11 | ±0.78 | ±1.06 | ±0.44 | ±0.82 | ±1.20 | ±0.81 |
| Textures | 75.15 | 56.50 | 59.34 | 58.85 | 36.38 | 47.25 | 25.90 | 25.90 | 14.07 | 31.87 | 48.95 | 23.20 |
| | ±0.66 | ±0.71 | ±0.36 | ±1.13 | ±0.85 | ±0.88 | ±0.71 | ±0.47 | ±0.54 | ±0.98 | ±0.82 | ±0.90 |
| iNaturalist | 70.30 | 51.80 | 79.96 | 48.90 | 48.61 | 43.05 | 45.25 | 44.85 | 22.50 | 48.27 | 64.23 | 44.94 |
| | ±0.70 | ±0.71 | ±0.68 | ±0.72 | ±1.20 | ±0.83 | ±1.24 | ±0.28 | ±0.73 | ±0.86 | ±0.68 | ±1.11 |
| SVHN | 73.75 | 44.30 | 49.71 | 65.95 | 10.52 | 48.55 | 20.50 | 20.50 | 3.16 | 11.63 | 69.06 | 9.72 |
| | ±1.04 | ±0.62 | ±1.38 | ±1.02 | ±0.44 | ±1.46 | ±1.10 | ±0.99 | ±0.37 | ±0.53 | ±0.72 | ±0.58 |
| **average** | 79.42 | 59.56 | 47.88 | 68.76 | 25.90 | 51.58 | 30.17 | 29.52 | 15.63 | 24.20 | 66.75 | 20.49 |

(ID row label at left: CIFAR-100 (DenseNet))

Table 4: Performance comparison (in terms of AUROC) of various methods in CIFAR-100 (ID dataset) under ResNet34. The best and second-best models without accessing external outlier datasets are coloured red and blue, respectively.

| OOD dataset | w/o outlier exposure (FPR95% ↓) | | | | | | | | | w/ outlier exposure (FPR95% ↓) | | |
| --- | --- | --- | --- | --- | --- | --- | --- | --- | --- | --- | --- | --- |
| | Msp/Odin/Mah/Energy/Gram/ Energy+ReAct/pNML/Odin+pNML/Ours | | | | | | | | | Oecc/Energy/ Oecc+pNML | | |
| iSUN | 75.94 | 85.67 | 82.53 | 78.16 | 98.96 | 78.29 | 83.37 | 87.92 | 98.98 | 99.04 | 89.41 | 99.36 |
| | ±0.33 | ±0.50 | ±0.30 | ±0.20 | ±0.08 | ±0.21 | ±0.28 | ±0.22 | ±0.10 | ±0.08 | ±0.19 | ±0.06 |
| LSUN(R) | 75.78 | 85.46 | 82.44 | 78.46 | 99.37 | 78.03 | 84.09 | 88.31 | 99.30 | 99.31 | 89.99 | 99.55 |
| | ±0.26 | ±0.32 | ±0.37 | ±0.28 | ±0.05 | ±0.25 | ±0.29 | ±0.28 | ±0.05 | ±0.03 | ±0.23 | ±0.03 |
| LSUN(C) | 75.45 | 82.60 | 88.19 | 75.37 | 92.25 | 78.93 | 82.94 | 88.22 | 98.54 | 95.74 | 93.13 | 97.76 |
| | ±0.53 | ±0.37 | ±0.30 | ±0.48 | ±0.25 | ±0.42 | ±0.41 | ±0.31 | ±0.14 | ±0.22 | ±0.22 | ±0.14 |
| CIFAR-10 | 77.98 | 70.01 | 71.08 | 77.86 | 68.98 | 77.74 | 78.10 | 73.32 | 75.09 | 71.13 | 80.65 | 75.83 |
| | ±0.34 | ±0.38 | ±0.50 | ±0.29 | ±0.33 | ±0.28 | ±0.48 | ±0.39 | ±0.49 | ±0.27 | ±0.30 | ±0.45 |
| ImageNet(R) | 76.83 | 87.43 | 83.56 | 79.99 | 98.90 | 79.70 | 83.74 | 87.89 | 99.04 | 98.74 | 88.19 | 98.89 |
| | ±0.42 | ±0.27 | ±0.39 | ±0.15 | ±0.07 | ±0.13 | ±0.36 | ±0.23 | ±0.08 | ±0.06 | ±0.15 | ±0.06 |
| ImageNet(C) | 79.08 | 84.67 | 85.08 | 81.07 | 97.79 | 81.33 | 85.32 | 88.18 | 99.16 | 97.99 | 89.04 | 98.16 |
| | ±0.23 | ±0.23 | ±0.21 | ±0.44 | ±0.10 | ±0.44 | ±0.15 | ±0.13 | ±0.06 | ±0.08 | ±0.21 | ±0.08 |
| Gaussian | 45.29 | 84.02 | 96.15 | 45.56 | 99.99 | 51.65 | 86.72 | 95.83 | 100.0 | 99.98 | 99.28 | 100.0 |
| | ±0.21 | ±0.15 | ±0.10 | ±0.22 | ±0.00 | ±0.22 | ±0.15 | ±0.06 | ±0.00 | ±0.00 | ±0.01 | ±0.00 |
| Places365 | 74.82 | 73.64 | 85.77 | 72.96 | 88.22 | 77.81 | 83.51 | 84.71 | 95.93 | 92.63 | 92.18 | 95.67 |
| | ±0.61 | ±0.53 | ±0.44 | ±0.44 | ±0.28 | ±0.34 | ±0.45 | ±0.32 | ±0.16 | ±0.24 | ±0.11 | ±0.14 |
| Textures | 76.92 | 77.17 | 86.04 | 77.73 | 91.96 | 80.73 | 85.68 | 86.13 | 96.49 | 94.34 | 91.63 | 96.08 |
| | ±0.24 | ±0.20 | ±0.18 | ±0.19 | ±0.28 | ±0.21 | ±0.25 | ±0.14 | ±0.19 | ±0.19 | ±0.18 | ±0.19 |
| iNaturalist | 82.00 | 82.43 | 87.25 | 82.39 | 88.82 | 84.35 | 89.25 | 89.25 | 94.40 | 89.12 | 87.36 | 88.22 |
| | ±0.40 | ±0.49 | ±0.29 | ±0.35 | ±0.34 | ±0.31 | ±0.26 | ±0.34 | ±0.13 | ±0.23 | ±0.24 | ±0.29 |
| SVHN | 79.12 | 93.42 | 94.06 | 79.43 | 95.94 | 82.72 | 90.78 | 94.92 | 99.67 | 96.88 | 93.37 | 97.54 |
| | ±0.19 | ±0.63 | ±0.34 | ±0.38 | ±0.13 | ±0.30 | ±0.19 | ±0.44 | ±0.05 | ±0.19 | ±0.09 | ±0.15 |
| **average** | 74.47 | 82.41 | 85.65 | 75.36 | 92.83 | 77.39 | 84.86 | 87.70 | 96.05 | 94.08 | 90.38 | 95.19 |

(ID row label at left: CIFAR-100 (ResNet34))

Table 5: Performance comparison (in terms of AUROC) of various methods in CIFAR-100 (ID dataset) under WideResNet. The best and second-best models without accessing external outlier datasets are coloured red and blue, respectively.

| OOD dataset | w/o outlier exposure (FPR95% ↓) | | | | | | | | | w/ outlier exposure (FPR95% ↓) | | |
| | Msp | Odin | Mah | Energy | Gram | Energy+ReAct | pNML | Odin+pNML | Ours | Oecc | Energy | Oecc+pNML |
|---|---|---|---|---|---|---|---|---|---|---|---|---|
| iSUN | 69.62 | 83.64 | 91.98 | 79.48 | 98.87 | 71.98 | 78.11 | 78.11 | 99.41 | 98.49 | 89.26 | 99.25 |
| | ±0.49 | ±0.21 | ±0.15 | ±0.41 | ±0.06 | ±0.36 | ±0.50 | ±0.11 | ±0.05 | ±0.10 | ±0.25 | ±0.06 |
| LSUN(R) | 69.22 | 83.40 | 93.24 | 79.39 | 99.34 | 72.40 | 78.02 | 78.02 | 99.68 | 98.83 | 89.89 | 99.48 |
| | ±0.37 | ±0.30 | ±0.24 | ±0.28 | ±0.05 | ±0.39 | ±0.34 | ±0.15 | ±0.03 | ±0.07 | ±0.22 | ±0.04 |
| LSUN(C) | 77.07 | 86.63 | 88.38 | 83.47 | 90.96 | 85.93 | 57.43 | 80.17 | 98.78 | 96.16 | 95.05 | 98.56 |
| | ±0.40 | ±0.30 | ±0.32 | ±0.33 | ±0.34 | ±0.20 | ±0.42 | ±0.17 | ±0.05 | ±0.26 | ±0.15 | ±0.15 |
| CIFAR-10 | 75.96 | 68.89 | 65.82 | 79.31 | 64.53 | 65.81 | 43.12 | 46.92 | 69.68 | 68.87 | 78.35 | 62.98 |
| | ±0.27 | ±0.46 | ±0.34 | ±0.39 | ±0.71 | ±0.54 | ±0.60 | ±0.41 | ±0.55 | ±0.43 | ±0.16 | ±0.52 |
| ImageNet(R) | 68.77 | 82.23 | 92.89 | 78.12 | 98.61 | 70.79 | 77.61 | 77.61 | 99.11 | 97.78 | 84.90 | 98.57 |
| | ±0.44 | ±0.36 | ±0.16 | ±0.38 | ±0.05 | ±0.41 | ±0.28 | ±0.15 | ±0.06 | ±0.09 | ±0.39 | ±0.08 |
| ImageNet(C) | 72.59 | 82.80 | 89.87 | 81.62 | 97.17 | 75.36 | 68.97 | 70.35 | 99.10 | 97.24 | 87.63 | 98.43 |
| | ±0.39 | ±0.32 | ±0.28 | ±0.34 | ±0.12 | ±0.49 | ±0.59 | ±0.13 | ±0.06 | ±0.08 | ±0.41 | ±0.08 |
| Gaussian | 75.99 | 99.23 | 100.0 | 91.47 | 100.0 | 91.31 | 99.99 | 100.0 | 99.96 | 99.99 | 99.91 | 100.0 |
| | ±0.20 | ±0.15 | ±0.00 | ±0.12 | ±0.00 | ±0.16 | ±0.00 | ±0.00 | ±0.00 | ±0.00 | ±0.00 | ±0.00 |
| Places365 | 74.64 | 78.21 | 85.97 | 77.17 | 86.78 | 79.23 | 64.27 | 80.44 | 97.65 | 94.93 | 93.87 | 97.94 |
| | ±0.39 | ±0.36 | ±0.33 | ±0.55 | ±0.26 | ±0.35 | ±0.51 | ±0.29 | ±0.10 | ±0.17 | ±0.17 | ±0.11 |
| Textures | 74.63 | 79.10 | 91.24 | 77.89 | 91.25 | 83.28 | 70.01 | 81.17 | 97.24 | 96.32 | 92.89 | 98.51 |
| | ±0.33 | ±0.36 | ±0.21 | ±0.43 | ±0.23 | ±0.32 | ±0.79 | ±0.19 | ±0.11 | ±0.14 | ±0.22 | ±0.10 |
| iNaturalist | 78.90 | 83.13 | 85.71 | 83.38 | 87.02 | 80.36 | 62.25 | 71.58 | 96.78 | 91.97 | 88.23 | 95.32 |
| | ±0.27 | ±0.34 | ±0.31 | ±0.40 | ±0.58 | ±0.23 | ±0.56 | ±0.27 | ±0.09 | ±0.32 | ±0.28 | ±0.27 |
| SVHN | 78.61 | 91.86 | 93.22 | 73.95 | 96.05 | 87.39 | 78.73 | 80.42 | 99.63 | 97.93 | 93.72 | 99.08 |
| | ±0.40 | ±0.26 | ±0.18 | ±0.36 | ±0.15 | ±0.29 | ±0.52 | ±0.08 | ±0.05 | ±0.10 | ±0.14 | ±0.05 |
| **average** | 74.18 | 83.56 | 88.94 | 80.48 | 91.87 | 78.53 | 70.77 | 76.80 | 96.09 | 94.41 | 90.34 | 95.28 |

(CIFAR-100 (WideResNet))

Table 6: Performance comparison (in terms of AUROC) of various methods in CIFAR-100 (ID dataset) under DenseNet. The best and second-best models without accessing external outlier datasets are coloured red and blue, respectively.

| OOD dataset | w/o outlier exposure (FPR95% ↓) | | | | | | | | | w/ outlier exposure (FPR95% ↓) | | |
| | Msp | Odin | Mah | Energy | Gram | Energy+ReAct | pNML | Odin+pNML | Ours | Oecc | Energy | Oecc+pNML |
|---|---|---|---|---|---|---|---|---|---|---|---|---|
| iSUN | 70.31 | 84.84 | 96.42 | 77.20 | 99.07 | 93.20 | 96.46 | 96.77 | 98.86 | 99.04 | 82.00 | 99.49 |
| | ±0.24 | ±0.48 | ±0.09 | ±0.39 | ±0.04 | ±0.12 | ±0.09 | ±0.52 | ±0.06 | ±0.06 | ±0.26 | ±0.04 |
| LSUN(R) | 70.84 | 85.68 | 97.15 | 79.49 | 99.36 | 94.37 | 96.78 | 97.12 | 99.03 | 99.30 | 83.17 | 99.59 |
| | ±0.59 | ±0.25 | ±0.16 | ±0.28 | ±0.04 | ±0.21 | ±0.16 | ±0.22 | ±0.06 | ±0.05 | ±0.23 | ±0.03 |
| LSUN(C) | 79.99 | 91.16 | 73.12 | 90.48 | 91.55 | 90.91 | 92.84 | 92.84 | 98.32 | 93.86 | 94.18 | 95.92 |
| | ±0.32 | ±0.42 | ±0.46 | ±0.19 | ±0.31 | ±0.20 | ±0.16 | ±0.20 | ±0.07 | ±0.28 | ±0.18 | ±0.24 |
| CIFAR-10 | 76.61 | 73.56 | 50.48 | 76.75 | 65.24 | 75.64 | 49.87 | 50.43 | 61.65 | 63.02 | 75.61 | 59.80 |
| | ±0.65 | ±0.58 | ±0.56 | ±0.42 | ±0.73 | ±0.43 | ±0.57 | ±0.53 | ±0.61 | ±0.52 | ±0.37 | ±0.52 |
| ImageNet(R) | 71.15 | 85.25 | 96.22 | 78.51 | 98.94 | 93.71 | 97.26 | 97.52 | 99.35 | 98.89 | 81.19 | 99.42 |
| | ±0.50 | ±0.54 | ±0.20 | ±0.44 | ±0.02 | ±0.17 | ±0.12 | ±0.36 | ±0.03 | ±0.04 | ±0.38 | ±0.03 |
| ImageNet(C) | 76.32 | 88.88 | 90.99 | 85.02 | 97.64 | 94.42 | 95.63 | 95.88 | 99.19 | 98.18 | 86.71 | 98.97 |
| | ±0.39 | ±0.43 | ±0.32 | ±0.35 | ±0.09 | ±0.21 | ±0.12 | ±0.45 | ±0.05 | ±0.08 | ±0.30 | ±0.06 |
| Gaussian | 30.65 | 50.50 | 100.0 | 15.34 | 100.0 | 61.56 | 100.0 | 100.0 | 99.99 | 99.99 | 65.24 | 100.0 |
| | ±0.08 | ±0.01 | ±0.00 | ±0.06 | ±0.00 | ±0.13 | ±0.00 | ±0.00 | ±0.00 | ±0.00 | ±0.17 | ±0.00 |
| Places365 | 79.51 | 89.30 | 66.06 | 89.49 | 86.91 | 89.64 | 92.96 | 92.96 | 97.24 | 88.58 | 92.58 | 91.74 |
| | ±0.36 | ±0.31 | ±0.54 | ±0.20 | ±0.23 | ±0.28 | ±0.20 | ±0.65 | ±0.10 | ±0.23 | ±0.18 | ±0.18 |
| Textures | 77.92 | 87.49 | 81.30 | 86.53 | 90.40 | 90.15 | 95.19 | 95.19 | 96.71 | 91.44 | 90.74 | 94.05 |
| | ±0.42 | ±0.27 | ±0.40 | ±0.34 | ±0.21 | ±0.28 | ±0.16 | ±0.46 | ±0.10 | ±0.26 | ±0.26 | ±0.23 |
| iNaturalist | 82.62 | 90.93 | 69.00 | 91.19 | 85.87 | 91.72 | 89.82 | 89.89 | 96.00 | 86.63 | 88.31 | 88.04 |
| | ±0.33 | ±0.49 | ±0.53 | ±0.14 | ±0.50 | ±0.14 | ±0.34 | ±0.42 | ±0.07 | ±0.45 | ±0.17 | ±0.44 |
| SVHN | 82.12 | 92.45 | 87.56 | 87.79 | 97.20 | 91.93 | 96.32 | 96.32 | 99.18 | 96.80 | 89.69 | 97.28 |
| | ±0.24 | ±0.17 | ±0.48 | ±0.21 | ±0.11 | ±0.20 | ±0.17 | ±0.42 | ±0.13 | ±0.17 | ±0.17 | ±0.14 |
| **average** | 72.55 | 83.64 | 82.57 | 77.98 | 92.02 | 87.93 | 91.19 | 91.36 | 95.05 | 92.34 | 84.49 | 93.12 |

(CIFAR-100 (DenseNet))

Table 7: Performance comparison (in terms of AUPR) of various methods in CIFAR-100 (ID dataset) under ResNet34. The best and second-best models without accessing external outlier datasets are coloured red and blue, respectively.

| OOD dataset | w/o outlier exposure (FPR95% ↓) | | | | | | | | | w/ outlier exposure (FPR95% ↓) | | |
|---|---|---|---|---|---|---|---|---|---|---|---|---|
| | Msp/Odin/Mah/Energy/Gram/ Energy+ReAct/pNML/Odin+pNML/Ours | | | | | | | | | Oecc/Energy/ Oecc+pNML | | |
| iSUN | 94.11 | 96.21 | 95.72 | 94.49 | 99.74 | 94.70 | 96.24 | 97.24 | 99.78 | 99.77 | 97.68 | 99.86 |
| | ±0.13 | ±0.21 | ±0.08 | ±0.07 | ±0.03 | ±0.08 | ±0.09 | ±0.06 | ±0.03 | ±0.04 | ±0.06 | ±0.02 |
| LSUN(R) | 93.93 | 96.13 | 95.77 | 94.49 | 99.86 | 94.43 | 96.42 | 97.37 | 99.85 | 99.85 | 97.87 | 99.90 |
| | ±0.12 | ±0.14 | ±0.09 | ±0.10 | ±0.01 | ±0.08 | ±0.10 | ±0.08 | ±0.01 | ±0.01 | ±0.06 | ±0.01 |
| LSUN(C) | 93.76 | 94.82 | 97.00 | 93.52 | 97.91 | 95.12 | 95.87 | 97.08 | 99.69 | 98.88 | 98.52 | 99.44 |
| | ±0.20 | ±0.13 | ±0.10 | ±0.17 | ±0.10 | ±0.12 | ±0.15 | ±0.08 | ±0.02 | ±0.08 | ±0.06 | ±0.05 |
| CIFAR-10 | 94.50 | 91.23 | 91.31 | 94.51 | 90.59 | 94.37 | 94.02 | 91.88 | 93.47 | 91.05 | 95.57 | 92.80 |
| | ±0.09 | ±0.15 | ±0.24 | ±0.12 | ±0.13 | ±0.10 | ±0.22 | ±0.28 | ±0.19 | ±0.14 | ±0.09 | ±0.24 |
| ImageNet(R) | 94.48 | 96.92 | 96.04 | 95.15 | 99.74 | 95.10 | 96.28 | 97.19 | 99.79 | 99.72 | 97.49 | 99.75 |
| | ±0.13 | ±0.09 | ±0.12 | ±0.04 | ±0.02 | ±0.05 | ±0.10 | ±0.08 | ±0.02 | ±0.02 | ±0.05 | ±0.02 |
| ImageNet(C) | 94.91 | 95.82 | 96.39 | 95.27 | 99.44 | 95.48 | 96.65 | 97.29 | 99.82 | 99.54 | 97.67 | 99.57 |
| | ±0.10 | ±0.12 | ±0.07 | ±0.12 | ±0.03 | ±0.12 | ±0.05 | ±0.04 | ±0.01 | ±0.02 | ±0.05 | ±0.02 |
| Gaussian | 86.72 | 96.87 | 99.28 | 86.66 | 100.0 | 88.86 | 97.43 | 99.23 | 100.0 | 100.0 | 99.87 | 100.0 |
| | ±0.07 | ±0.04 | ±0.02 | ±0.09 | ±0.00 | ±0.08 | ±0.03 | ±0.01 | ±0.00 | ±0.00 | ±0.00 | ±0.00 |
| Places365 | 93.01 | 92.40 | 96.63 | 92.24 | 96.56 | 94.74 | 95.98 | 96.33 | 99.09 | 97.97 | 98.26 | 98.85 |
| | ±0.27 | ±0.27 | ±0.14 | ±0.21 | ±0.11 | ±0.13 | ±0.16 | ±0.13 | ±0.04 | ±0.08 | ±0.04 | ±0.04 |
| Textures | 93.51 | 93.43 | 96.65 | 93.85 | 97.74 | 95.43 | 96.49 | 96.59 | 99.21 | 98.49 | 98.15 | 98.94 |
| | ±0.12 | ±0.15 | ±0.06 | ±0.14 | ±0.11 | ±0.09 | ±0.09 | ±0.11 | ±0.04 | ±0.08 | ±0.06 | ±0.07 |
| iNaturalist | 95.43 | 95.41 | 97.11 | 95.43 | 96.80 | 96.40 | 97.66 | 97.66 | 98.78 | 97.02 | 97.29 | 96.73 |
| | ±0.15 | ±0.18 | ±0.09 | ±0.14 | ±0.16 | ±0.10 | ±0.07 | ±0.09 | ±0.03 | ±0.09 | ±0.07 | ±0.11 |
| SVHN | 94.84 | 98.58 | 98.70 | 94.94 | 98.95 | 96.13 | 98.01 | 88.88 | 99.92 | 99.21 | 98.64 | 99.40 |
| | ±0.09 | ±0.26 | ±0.08 | ±0.18 | ±0.04 | ±0.11 | ±0.05 | ±0.13 | ±0.01 | ±0.07 | ±0.02 | ±0.05 |
| **average** | 93.56 | 95.26 | 96.42 | 93.69 | **97.94** | 94.61 | 96.46 | 96.98 | **99.04** | 98.32 | 97.91 | 98.66 |

*ID column label: CIFAR-100 (ResNet34)*

Table 8: Performance comparison (in terms of AUPR) of various methods in CIFAR-100 (ID dataset) under WideResNet. The best and second-best models without accessing external outlier datasets are coloured red and blue, respectively.

| OOD dataset | w/o outlier exposure (FPR95% ↓) | | | | | | | | | w/ outlier exposure (FPR95% ↓) | | |
|---|---|---|---|---|---|---|---|---|---|---|---|---|
| | Msp/Odin/Mah/Energy/Gram/ Energy+ReAct/pNML/Odin+pNML/Ours | | | | | | | | | Oecc/Energy/ Oecc+pNML | | |
| iSUN | 92.05 | 95.93 | 98.11 | 95.07 | 99.71 | 92.87 | 94.26 | 94.26 | 99.87 | 99.63 | 97.65 | 99.82 |
| | ±0.19 | ±0.07 | ±0.07 | ±0.14 | ±0.02 | ±0.12 | ±0.17 | ±0.02 | ±0.01 | ±0.04 | ±0.06 | ±0.02 |
| LSUN(R) | 91.82 | 95.82 | 98.51 | 95.07 | 99.84 | 92.95 | 94.38 | 94.38 | 99.93 | 99.74 | 97.80 | 99.89 |
| | ±0.11 | ±0.08 | ±0.06 | ±0.10 | ±0.02 | ±0.12 | ±0.13 | ±0.04 | ±0.01 | ±0.02 | ±0.05 | ±0.01 |
| LSUN(C) | 94.48 | 96.68 | 97.08 | 96.14 | 97.45 | 96.60 | 85.54 | 94.25 | 99.74 | 98.88 | 98.94 | 99.64 |
| | ±0.12 | ±0.09 | ±0.11 | ±0.09 | ±0.11 | ±0.09 | ±0.11 | ±0.09 | ±0.01 | ±0.11 | ±0.04 | ±0.05 |
| CIFAR-10 | 94.24 | 91.45 | 89.85 | 94.93 | 88.52 | 90.13 | 80.97 | 82.24 | 91.10 | 90.63 | 95.00 | 87.98 |
| | ±0.08 | ±0.19 | ±0.16 | ±0.13 | ±0.29 | ±0.27 | ±0.21 | ±0.19 | ±0.21 | ±0.22 | ±0.07 | ±0.23 |
| ImageNet(R) | 91.50 | 95.49 | 98.39 | 94.39 | 99.63 | 92.41 | 93.91 | 93.91 | 99.80 | 99.49 | 96.46 | 99.68 |
| | ±0.19 | ±0.11 | ±0.05 | ±0.16 | ±0.01 | ±0.15 | ±0.11 | ±0.04 | ±0.02 | ±0.03 | ±0.13 | ±0.02 |
| ImageNet(C) | 92.84 | 95.70 | 97.68 | 95.44 | 99.27 | 93.74 | 90.58 | 90.94 | 99.80 | 99.35 | 97.10 | 99.63 |
| | ±0.14 | ±0.11 | ±0.09 | ±0.13 | ±0.04 | ±0.15 | ±0.22 | ±0.03 | ±0.02 | ±0.03 | ±0.11 | ±0.02 |
| Gaussian | 95.10 | 99.86 | 100.0 | 98.37 | 100.0 | 98.20 | 100.0 | 100.0 | 99.99 | 100.0 | 99.98 | 100.0 |
| | ±0.04 | ±0.06 | ±0.00 | ±0.03 | ±0.00 | ±0.04 | ±0.00 | ±0.00 | ±0.00 | ±0.00 | ±0.00 | ±0.00 |
| Places365 | 93.61 | 94.47 | 96.50 | 94.45 | 96.04 | 94.85 | 88.75 | 94.40 | 99.49 | 98.53 | 98.70 | 99.46 |
| | ±0.14 | ±0.10 | ±0.11 | ±0.15 | ±0.11 | ±0.12 | ±0.24 | ±0.08 | ±0.03 | ±0.08 | ±0.04 | ±0.04 |
| Textures | 93.25 | 94.46 | 97.90 | 94.32 | 97.50 | 96.00 | 91.02 | 94.50 | 99.37 | 99.08 | 98.42 | 99.65 |
| | ±0.16 | ±0.16 | ±0.08 | ±0.13 | ±0.10 | ±0.11 | ±0.11 | ±0.29 | ±0.03 | ±0.04 | ±0.06 | ±0.03 |
| iNaturalist | 94.98 | 95.96 | 96.67 | 96.05 | 96.19 | 95.26 | 88.07 | 91.44 | 99.32 | 97.89 | 97.45 | 98.81 |
| | ±0.09 | ±0.09 | ±0.10 | ±0.15 | ±0.21 | ±0.08 | ±0.29 | ±0.08 | ±0.02 | ±0.12 | ±0.07 | ±0.08 |
| SVHN | 95.08 | 98.22 | 98.47 | 93.59 | 98.96 | 97.38 | 94.50 | 94.16 | 99.91 | 99.45 | 98.68 | 99.77 |
| | ±0.10 | ±0.08 | ±0.05 | ±0.10 | ±0.06 | ±0.08 | ±0.16 | ±0.02 | ±0.01 | ±0.03 | ±0.04 | ±0.01 |
| **average** | 93.54 | 95.82 | 97.20 | 95.26 | **97.56** | 94.58 | 91.09 | 93.13 | **98.94** | 98.42 | 97.83 | 98.58 |

*ID column label: CIFAR-100 (WideResNet)*

Table 9: Performance comparison (in terms of AUPR) of various methods in CIFAR-100 (ID dataset) under DenseNet. The best and second-best models without accessing external outlier datasets are coloured red and blue, respectively.

| OOD dataset | w/o outlier exposure (FPR95% ↓) | | | | | | | | | w/ outlier exposure (FPR95% ↓) | | |
|---|---|---|---|---|---|---|---|---|---|---|---|---|
| | Msp/Odin/Mah/Energy/Gram/ Energy+ReAct/pNML/Odin+pNML/Ours | | | | | | | | | Oecc/Energy/ Oecc+pNML | | |
| iSUN | 92.41 | 96.42 | 99.17 | 94.44 | 99.77 | 98.56 | 99.23 | 99.31 | 99.77 | 99.77 | 95.94 | 99.89 |
| | ±0.08 | ±0.16 | ±0.03 | ±0.12 | ±0.01 | ±0.03 | ±0.02 | ±0.22 | ±0.01 | ±0.03 | ±0.06 | ±0.02 |
| LSUN(R) | 92.63 | 96.58 | 99.39 | 95.08 | 99.83 | 98.84 | 99.31 | 99.39 | 99.80 | 99.84 | 96.22 | 99.92 |
| | ±0.16 | ±0.07 | ±0.04 | ±0.09 | ±0.01 | ±0.05 | ±0.04 | ±0.11 | ±0.01 | ±0.01 | ±0.07 | ±0.01 |
| LSUN(C) | 95.22 | 97.96 | 92.36 | 97.86 | 97.63 | 97.95 | 98.38 | 98.38 | 99.64 | 98.26 | 98.77 | 98.87 |
| | ±0.10 | ±0.13 | ±0.15 | ±0.06 | ±0.09 | ±0.05 | ±0.04 | ±0.33 | ±0.02 | ±0.12 | ±0.04 | ±0.08 |
| CIFAR-10 | 94.36 | 92.93 | 85.25 | 93.87 | 88.96 | 93.78 | 84.22 | 84.41 | 86.56 | 87.87 | 94.21 | 86.64 |
| | ±0.19 | ±0.20 | ±0.23 | ±0.14 | ±0.29 | ±0.13 | ±0.33 | ±0.21 | ±0.33 | ±0.26 | ±0.10 | ±0.26 |
| ImageNet(R) | 92.59 | 96.39 | 99.09 | 94.57 | 99.72 | 98.71 | 99.41 | 99.47 | 99.87 | 99.74 | 95.64 | 99.88 |
| | ±0.17 | ±0.16 | ±0.07 | ±0.14 | ±0.01 | ±0.05 | ±0.03 | ±0.15 | ±0.01 | ±0.01 | ±0.10 | ±0.01 |
| ImageNet(C) | 94.06 | 97.34 | 97.70 | 96.26 | 99.41 | 98.85 | 99.05 | 99.11 | 99.84 | 99.57 | 96.97 | 99.77 |
| | ±0.12 | ±0.14 | ±0.11 | ±0.13 | ±0.03 | ±0.05 | ±0.03 | ±0.21 | ±0.01 | ±0.03 | ±0.09 | ±0.02 |
| Gaussian | 81.88 | 88.83 | 100.0 | 74.17 | 100.0 | 91.93 | 100.0 | 100.0 | 100.0 | 100.0 | 92.71 | 100.0 |
| | ±0.03 | ±0.00 | ±0.00 | ±0.05 | ±0.00 | ±0.04 | ±0.00 | ±0.00 | ±0.00 | ±0.00 | ±0.05 | ±0.00 |
| Places365 | 95.22 | 97.62 | 89.55 | 97.74 | 96.08 | 97.73 | 98.46 | 98.46 | 99.42 | 96.58 | 98.47 | 97.66 |
| | ±0.09 | ±0.11 | ±0.20 | ±0.05 | ±0.09 | ±0.07 | ±0.05 | ±0.33 | ±0.02 | ±±0.09 | ±0.04 | ±0.08 |
| Textures | 94.38 | 96.94 | 94.72 | 96.55 | 97.14 | 97.68 | 98.97 | 98.97 | 99.17 | 97.48 | 97.91 | 98.31 |
| | ±0.14 | ±0.12 | ±0.20 | ±0.13 | ±0.09 | ±0.08 | ±0.04 | ±0.20 | ±0.04 | ±±0.09 | ±0.08 | ±0.07 |
| iNaturalist | 96.06 | 98.08 | 90.76 | 98.10 | 95.72 | 98.16 | 97.64 | 97.66 | 99.16 | 96.04 | 97.48 | 96.52 |
| | ±0.10 | ±0.16 | ±0.25 | ±0.04 | ±0.17 | ±0.03 | ±0.10 | ±0.23 | ±0.02 | ±±0.17 | ±0.04 | ±0.16 |
| SVHN | 96.02 | 98.40 | 96.89 | 97.39 | 99.16 | 98.34 | 99.24 | 99.24 | 99.79 | 99.04 | 97.94 | 99.24 |
| | ±0.05 | ±0.05 | ±0.14 | ±0.05 | ±0.04 | ±0.04 | ±0.04 | ±0.17 | ±0.04 | ±±0.07 | ±0.04 | ±0.06 |
| **average** | 93.17 | 96.14 | 94.99 | 94.18 | 97.58 | 97.32 | 97.63 | 97.67 | 98.46 | 97.65 | 96.57 | 97.88 |

*(Row label on left margin: CIFAR-100 (DenseNet))*

## A.6 FPR95%, AUROC and AUPR for CIFAR-10 experiment

Tables 10, 11, and 12 below demonstrate FPR95%, AUROC, and AUPR values, respectively, for different models in the CIFAR-10 experiments.

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

| | OOD dataset | w/o outlier OOD (AUROC ↑) Msp/Odin/Mah/Energy/Gram/Energy+ReAct/pNML/Odin+pNML/Ours | w/ outlier OOD (AUROC ↑) Oecc/Energy/Oecc+pNML |
|---|---|---|---|
| CIFAR-10 (ResNet34) | iSUN | 90.74 / 93.65 / 96.93 / 92.60 / 99.84 / 92.71 / 96.37 / 97.53 / 99.84 | 99.92 / 98.78 / 99.94 |
| | LSUN(R) | 91.51 / 94.52 / 97.19 / 93.15 / 99.90 / 93.25 / 96.83 / 97.98 / 99.82 | 99.94 / 98.92 / 99.95 |
| | LSUN(C) | 91.98 / 93.80 / 94.38 / 94.00 / 97.73 / 93.99 / 95.41 / 95.57 / 99.56 | 99.08 / 98.55 / 99.48 |
| | CIFAR-100 | 87.01 / 87.32 / 88.34 / 87.11 / 79.80 / 87.14 / 89.63 / 88.98 / 91.05 | 87.24 / 93.30 / 94.19 |
| | ImageNet(R) | 90.77 / 93.62 / 95.86 / 92.23 / 99.69 / 92.44 / 95.21 / 96.36 / 99.66 | 99.86 / 97.77 / 99.90 |
| | ImageNet(C) | 91.27 / 93.15 / 95.44 / 92.79 / 99.40 / 92.92 / 95.29 / 96.04 / 99.74 | 99.71 / 98.09 / 99.85 |
| | Gaussian | 97.48 / 99.99 / 99.97 / 98.22 / 100.0 / 98.21 / 99.98 / 100.0 / 100.0 | 100.0 / 99.32 / 100.0 |
| | Places365 | 91.08 / 92.64 / 93.86 / 92.75 / 97.68 / 92.90 / 95.09 / 95.09 / 98.96 | 98.98 / 98.56 / 99.53 |
| | Textures | 89.04 / 90.05 / 94.12 / 90.42 / 97.55 / 90.54 / 94.43 / 94.43 / 98.86 | 99.17 / 98.39 / 99.59 |
| | iNaturalist | 90.86 / 91.73 / 93.67 / 91.72 / 94.52 / 91.88 / 94.29 / 94.29 / 98.94 | 97.08 / 97.62 / 99.18 |
| | SVHN | 89.89 / 95.92 / 94.11 / 91.28 / 99.43 / 91.21 / 94.85 / 97.96 / 99.92 | 99.74 / 98.93 / 99.85 |
| | **average** | 91.06 / 93.31 / 94.90 / 92.39 / **96.87** / 92.47 / 95.22 / 95.84 / **98.76** | 98.25 / 98.02 / 99.22 |
| CIFAR-10 (WRN) | iSUN | 91.56 / 95.43 / 93.69 / 92.58 / 99.74 / 91.29 / 96.36 / 96.61 / 99.91 | 99.91 / 99.37 / 99.99 |
| | LSUN(R) | 92.02 / 95.94 / 94.28 / 94.42 / 99.85 / 93.26 / 96.53 / 96.98 / 99.88 | 99.92 / 99.38 / 99.99 |
| | LSUN(C) | 92.25 / 94.98 / 94.25 / 95.65 / 96.92 / 93.80 / 96.78 / 96.78 / 99.62 | 98.80 / 99.37 / 99.79 |
| | CIFAR-100 | 87.34 / 84.78 / 82.81 / 87.32 / 76.14 / 86.36 / 83.52 / 83.52 / 91.08 | 82.18 / 93.01 / 89.19 |
| | ImageNet(R) | 89.46 / 92.28 / 92.97 / 88.43 / 99.62 / 87.21 / 95.17 / 95.58 / 99.78 | 99.82 / 98.33 / 99.97 |
| | ImageNet(C) | 90.81 / 92.79 / 93.09 / 91.76 / 99.08 / 90.08 / 94.95 / 95.00 / 99.80 | 99.48 / 98.59 / 99.88 |
| | Gaussian | 81.48 / 96.39 / 100.0 / 81.86 / 100.0 / 92.37 / 100.0 / 100.0 / 99.99 | 100.0 / 99.74 / 100.0 |
| | Places365 | 90.32 / 92.76 / 94.16 / 92.05 / 97.08 / 88.30 / 96.40 / 96.40 / 99.01 | 98.96 / 99.43 / 99.78 |
| | Textures | 89.55 / 90.59 / 96.49 / 88.76 / 97.20 / 85.63 / 96.74 / 96.74 / 98.71 | 98.74 / 99.14 / 99.68 |
| | iNaturalist | 89.48 / 89.96 / 92.84 / 88.82 / 93.55 / 86.53 / 93.08 / 93.08 / 98.67 | 95.77 / 98.33 / 98.50 |
| | SVHN | 89.52 / 94.21 / 97.15 / 91.27 / 99.29 / 87.75 / 98.86 / 98.86 / 99.86 | 99.58 / 99.61 / 99.97 |
| | **average** | 89.44 / 92.74 / 93.79 / 90.27 / **96.22** / 89.33 / 95.31 / 95.41 / **98.76** | 97.56 / 98.57 / 98.79 |
| CIFAR-10 (DenseNet) | iSUN | 94.64 / 98.89 / 96.15 / 98.04 / 99.80 / 97.66 / 98.69 / 98.87 / 99.94 | 99.86 / 99.61 / 99.98 |
| | LSUN(R) | 95.59 / 99.29 / 96.95 / 98.37 / 99.90 / 98.13 / 98.97 / 99.24 / 99.95 | 99.91 / 99.62 / 99.98 |
| | LSUN(C) | 93.12 / 95.85 / 83.05 / 95.73 / 97.45 / 95.76 / 96.36 / 96.36 / 99.71 | 98.92 / 99.51 / 99.91 |
| | CIFAR-100 | 89.29 / 90.28 / 67.94 / 91.27 / 72.78 / 91.36 / 84.48 / 84.48 / 90.37 | 77.44 / 94.04 / 84.35 |
| | ImageNet(R) | 93.97 / 98.39 / 95.77 / 97.42 / 99.70 / 96.69 / 98.66 / 98.84 / 99.94 | 99.82 / 99.23 / 99.96 |
| | ImageNet(C) | 93.81 / 97.60 / 91.78 / 97.10 / 99.34 / 96.58 / 97.56 / 97.69 / 99.90 | 99.49 / 99.16 / 99.79 |
| | Gaussian | 97.64 / 100.0 / 100.0 / 98.84 / 100.0 / 99.74 / 100.0 / 100.0 / 100.0 | 99.99 / 97.50 / 100.0 |
| | Places365 | 91.91 / 94.57 / 82.80 / 94.48 / 97.32 / 94.58 / 95.23 / 95.23 / 98.90 | 98.65 / 99.47 / 99.80 |
| | Textures | 91.42 / 92.89 / 90.77 / 93.15 / 97.46 / 94.40 / 96.03 / 96.03 / 98.90 | 98.41 / 99.27 / 99.69 |
| | iNaturalist | 91.16 / 92.32 / 86.38 / 92.47 / 93.75 / 92.77 / 93.34 / 93.55 / 98.47 | 95.32 / 98.34 / 98.43 |
| | SVHN | 89.90 / 94.13 / 95.40 / 91.01 / 98.94 / 94.73 / 98.33 / 98.63 / 99.93 | 99.46 / 98.77 / 99.92 |
| | **average** | 92.95 / 95.84 / 89.73 / 95.26 / 96.04 / 95.67 / 96.15 / **96.27** / **98.73** | 97.02 / 98.59 / 98.35 |

Table 12: Performance comparison (in terms of AUPR) of various methods in CIFAR-10 (ID dataset) under three commonly-used networks. The best and second-best models without accessing external outlier datasets are coloured red and blue, respectively.

| | OOD dataset | w/o outlier OOD (AUPR ↑) Msp/Odin/Mah/Energy/Gram/ Energy+ReAct/pNML/Odin+pNML/Ours | w/ outlier OOD (AUPR ↑) Oecc/Energy/ Oecc+pNML |
|---|---|---|---|
| CIFAR-10 (ResNet34) | iSUN | 98.01 / 98.54 / 99.37 / 98.32 / 99.96 / 98.35 / 99.26 / 99.48 / 99.97 | 99.98 / 99.76 / 99.99 |
| | LSUN(R) | 98.21 / 98.75 / 99.43 / 98.45 / 99.98 / 98.47 / 99.37 / 99.58 / 99.96 | 99.99 / 99.79 / 99.99 |
| | LSUN(C) | 98.32 / 98.64 / 98.87 / 98.69 / 99.41 / 98.68 / 99.07 / 99.02 / 99.91 | 99.79 / 99.69 / 99.89 |
| | CIFAR-100 | 96.96 / 96.87 / 97.29 / 96.79 / 93.87 / 96.81 / 97.65 / 97.42 / 97.86 | 96.05 / 97.97 / 98.15 |
| | ImageNet(R) | 98.03 / 98.55 / 99.14 / 98.24 / 99.92 / 98.30 / 99.01 / 99.22 / 99.92 | 99.97 / 99.47 / 99.98 |
| | ImageNet(C) | 98.17 / 98.47 / 99.07 / 98.38 / 99.85 / 98.41 / 99.05 / 99.16 / 99.95 | 99.94 / 99.57 / 99.97 |
| | Gaussian | 99.54 / 100.0 / 99.99 / 99.69 / 100.0 / 99.68 / 100.0 / 100.0 / 100.0 | 100.0 / 99.88 / 100.0 |
| | Places365 | 98.08 / 98.33 / 98.75 / 98.37 / 99.40 / 98.42 / 98.98 / 98.98 / 99.78 | 99.77 / 99.71 / 99.90 |
| | Textures | 97.58 / 97.67 / 98.82 / 97.73 / 99.36 / 97.78 / 98.86 / 98.86 / 99.76 | 99.81 / 99.64 / 99.91 |
| | iNaturalist | 98.04 / 98.13 / 98.72 / 98.04 / 98.56 / 98.10 / 98.83 / 98.83 / 99.79 | 99.27 / 99.45 / 99.82 |
| | SVHN | 97.96 / 99.13 / 98.83 / 98.17 / 99.86 / 98.16 / 98.99 / 99.54 / 99.98 | 99.94 / 99.79 / 99.97 |
| | **average** | 98.08 / 98.46 / 98.93 / 98.26 / **99.11** / 98.29 / 99.01 / 99.10 / **99.72** | 99.50 / 99.52 / 99.78 |
| CIFAR-10 (WRN) | iSUN | 98.24 / 98.95 / 98.65 / 98.28 / 99.94 / 98.04 / 99.22 / 99.26 / 99.98 | 99.98 / 99.87 / 100.0 |
| | LSUN(R) | 98.31 / 99.05 / 98.82 / 98.76 / 99.96 / 98.55 / 99.26 / 99.36 / 99.98 | 99.98 / 99.88 / 100.0 |
| | LSUN(C) | 98.26 / 98.72 / 98.75 / 98.90 / 99.18 / 98.50 / 99.27 / 99.27 / 99.92 | 99.68 / 99.87 / 99.92 |
| | CIFAR-100 | 96.86 / 95.50 / 95.85 / 96.62 / 92.88 / 96.50 / 95.78 / 95.78 / 97.96 | 94.38 / 98.14 / 96.19 |
| | ImageNet(R) | 97.70 / 98.09 / 98.47 / 96.99 / 99.90 / 96.87 / 98.94 / 99.02 / 99.95 | 99.95 / 99.63 / 99.99 |
| | ImageNet(C) | 98.03 / 98.29 / 98.50 / 98.05 / 99.77 / 97.69 / 98.86 / 98.86 / 99.96 | 99.86 / 99.68 / 99.96 |
| | Gaussian | 96.49 / 99.33 / 100.0 / 96.02 / 100.0 / 98.61 / 100.0 / 100.0 / 100.0 | 100.0 / 99.96 / 100.0 |
| | Places365 | 97.73 / 98.14 / 98.71 / 98.05 / 99.26 / 96.91 / 99.17 / 99.17 / 99.78 | 99.72 / 99.88 / 99.92 |
| | Textures | 97.46 / 97.38 / 99.25 / 97.01 / 99.27 / 96.22 / 99.26 / 99.26 / 99.69 | 99.64 / 99.80 / 99.88 |
| | iNaturalist | 97.47 / 97.34 / 98.43 / 97.04 / 98.30 / 96.60 / 98.32 / 98.32 / 99.71 | 98.71 / 99.60 / 99.43 |
| | SVHN | 97.57 / 98.70 / 99.37 / 97.84 / 99.82 / 97.07 / 99.75 / 99.75 / 99.97 | 99.90 / 99.92 / 99.99 |
| | **average** | 97.65 / 98.14 / 98.62 / 97.60 / **98.93** / 97.41 / 98.89 / 98.91 / **99.72** | 99.25 / 99.66 / 99.57 |
| CIFAR-10 (DenseNet) | iSUN | 98.94 / 99.77 / 99.06 / 99.61 / 99.95 / 99.53 / 99.72 / 99.76 / 99.99 | 99.97 / 99.92 / 100.0 |
| | LSUN(R) | 99.14 / 99.86 / 99.29 / 99.68 / 99.98 / 99.63 / 99.78 / 99.84 / 99.99 | 99.98 / 99.93 / 100.0 |
| | LSUN(C) | 98.63 / 99.13 / 95.36 / 99.13 / 99.33 / 99.14 / 99.21 / 99.21 / 99.94 | 99.73 / 99.90 / 99.98 |
| | CIFAR-100 | 97.61 / 97.70 / 90.81 / 98.00 / 91.65 / 98.02 / 96.22 / 96.22 / 97.72 | 92.84 / 98.60 / 94.66 |
| | ImageNet(R) | 98.78 / 99.65 / 98.94 / 99.47 / 99.92 / 99.30 / 99.72 / 99.75 / 99.99 | 99.96 / 99.85 / 99.99 |
| | ImageNet(C) | 98.77 / 99.50 / 97.87 / 99.41 / 99.84 / 99.30 / 99.48 / 99.50 / 99.98 | 99.85 / 99.83 / 99.93 |
| | Gaussian | 99.56 / 100.0 / 100.0 / 99.79 / 100.0 / 99.95 / 100.0 / 100.0 / 100.0 | 100.0 / 99.57 / 100.0 |
| | Places365 | 98.35 / 98.87 / 95.31 / 98.86 / 99.31 / 98.87 / 98.97 / 98.97 / 99.77 | 99.65 / 99.89 / 99.94 |
| | Textures | 98.16 / 98.23 / 97.62 / 98.30 / 99.34 / 98.77 / 99.15 / 99.15 / 99.76 | 99.57 / 99.85 / 99.90 |
| | iNaturalist | 98.17 / 98.37 / 96.44 / 98.38 / 98.34 / 98.42 / 98.52 / 98.57 / 99.68 | 98.68 / 99.65 / 99.49 |
| | SVHN | 97.90 / 98.63 / 98.88 / 98.04 / 99.71 / 98.93 / 99.67 / 99.73 / 99.98 | 99.86 / 99.75 / 99.98 |
| | **average** | 98.55 / 99.06 / 97.23 / 98.97 / 98.85 / 99.08 / 99.13 / **99.15** / **99.71** | 99.10 / 99.70 / 99.44 |

# A.7 FPR95%, AUROC and AUPR for ImageNet experiment

Table 13: Performance comparison of various methods in ImageNet (ID dataset). The best and second-best models are coloured red and blue, respectively.

| | OOD dataset | Msp | Odin | Mah | Energy | Gram | Energy +ReAct | pNML | Odin +pNML | Ours |
|---|---|---|---|---|---|---|---|---|---|---|
| ImageNet(MobileNetV2) FPR95% → | iSUN | 42.41 | 0.83 | 97.71 | 30.79 | 16.69 | 8.92 | 56.67 | 8.77 | 0.26 |
| | LSUN(R) | 42.02 | 0.46 | 98.26 | 33.12 | 18.42 | 10.82 | 59.90 | 7.46 | 0.40 |
| | LSUN(C) | 26.11 | 0.03 | 99.55 | 11.38 | 11.07 | 5.76 | 15.33 | 4.09 | 0.00 |
| | CIFAR-10 | 64.21 | 3.49 | 93.88 | 65.83 | 4.00 | 47.21 | 58.51 | 4.23 | 2.23 |
| | CIFAR-100 | 58.98 | 4.71 | 88.60 | 55.30 | 11.43 | 38.33 | 39.90 | 4.20 | 3.07 |
| | Gaussian | 0.00 | 0.00 | 99.98 | 0.00 | 0.00 | 0.00 | 0.00 | 0.00 | 0.00 |
| | Places365 | 76.85 | 60.10 | 99.28 | 66.27 | 83.26 | 60.76 | 93.42 | 93.42 | 60.07 |
| | Textures | 70.90 | 49.95 | 68.00 | 54.50 | 45.22 | 40.50 | 14.63 | 14.63 | 18.33 |
| | iNaturalist | 59.82 | 56.53 | 99.37 | 55.32 | 70.55 | 44.73 | 83.58 | 83.58 | 49.07 |
| | SVHN | 8.49 | 0.04 | 98.48 | 2.90 | 0.66 | 1.54 | 0.08 | 0.08 | 0.00 |
| | **average** | 44.98 | 17.61 | 94.31 | 37.54 | 26.13 | 25.86 | 42.20 | 22.05 | **13.34** |
| ImageNet(MobileNetV2) AUORC ↑ | iSUN | 91.92 | 99.83 | 66.60 | 95.32 | 95.96 | 98.36 | 92.03 | 98.20 | 99.64 |
| | LSUN(R) | 92.00 | 99.87 | 68.24 | 95.01 | 95.46 | 98.08 | 91.58 | 98.38 | 99.56 |
| | LSUN(C) | 95.03 | 99.96 | 27.61 | 98.01 | 97.28 | 98.96 | 97.53 | 98.97 | 99.95 |
| | CIFAR-10 | 83.56 | 99.22 | 70.51 | 86.36 | 98.94 | 90.60 | 91.36 | 98.97 | 99.45 |
| | CIFAR-100 | 86.14 | 99.06 | 66.80 | 88.72 | 97.33 | 92.51 | 94.21 | 98.99 | 99.34 |
| | Gaussian | 99.73 | 99.99 | 73.59 | 99.97 | 100.0 | 99.98 | 100.0 | 100.0 | 99.98 |
| | Places365 | 78.14 | 84.81 | 25.84 | 83.21 | 61.15 | 84.32 | 57.10 | 57.10 | 86.55 |
| | Textures | 78.95 | 85.79 | 74.06 | 86.59 | 85.18 | 91.03 | 96.93 | 96.93 | 96.01 |
| | iNaturalist | 86.72 | 88.36 | 28.85 | 90.34 | 72.39 | 92.55 | 66.73 | 66.73 | 91.48 |
| | SVHN | 98.41 | 99.99 | 33.12 | 99.35 | 99.79 | 99.68 | 99.72 | 99.72 | 99.99 |
| | **average** | 89.06 | 95.69 | 53.52 | 92.29 | 90.35 | 94.61 | 88.72 | 91.40 | **97.20** |
| ImageNet(MobileNetV2) AUPR ↑ | iSUN | 98.43 | 99.97 | 92.75 | 99.18 | 98.99 | 99.70 | 98.59 | 99.68 | 99.94 |
| | LSUN(R) | 98.28 | 99.97 | 92.54 | 99.02 | 98.72 | 99.61 | 98.36 | 99.68 | 99.92 |
| | LSUN(C) | 98.90 | 99.99 | 73.37 | 99.60 | 99.23 | 99.79 | 99.52 | 99.80 | 99.99 |
| | CIFAR-10 | 96.05 | 99.84 | 92.36 | 96.98 | 99.74 | 97.91 | 98.29 | 99.79 | 99.89 |
| | CIFAR-100 | 96.78 | 99.80 | 90.03 | 97.48 | 99.27 | 98.33 | 98.87 | 99.79 | 99.86 |
| | Gaussian | 99.95 | 100.0 | 94.63 | 100.0 | 100.0 | 100.0 | 100.0 | 100.0 | 100.0 |
| | Places365 | 94.39 | 96.12 | 71.86 | 95.80 | 87.66 | 96.04 | 87.36 | 87.36 | 96.77 |
| | Textures | 96.67 | 97.66 | 95.06 | 97.86 | 97.26 | 98.72 | 99.60 | 99.60 | 99.45 |
| | iNaturalist | 96.96 | 97.35 | 73.90 | 97.96 | 91.17 | 98.43 | 90.15 | 90.15 | 98.22 |
| | SVHN | 99.18 | 99.99 | 75.78 | 99.87 | 99.86 | 99.93 | 99.88 | 99.88 | 100.0 |
| | **average** | 97.56 | 99.07 | 85.23 | 98.38 | 97.19 | 98.85 | 97.06 | 97.57 | **99.40** |

## A.8 Detailed results for ablation studies

Table 14 shows the performances of the proposed method under more hyperparameter choices, and some detailed results are provided in Table 15. Table 16 shows the detailed results across different OOD datasets to verify the effectiveness of the proposed two regularization terms.

Table 14: Ablation studies on CIFAR-100 (ID dataset) with WideResNet demonstrating the proposed method's robustness to hyperparameters. The result is the average value across all OOD datasets.

| ablation studies | parameters | mean AUROC (↑) | mean AUPR (↑) | mean FPR95% (↓) |
|---|---|---|---|---|
| batch size | 128 | 96.14 | 98.94 | 13.43 |
| | 192 | 96.08 | 98.93 | 13.63 |
| | 256 | 96.09 | 98.94 | 13.35 |
| | 384 | 96.07 | 98.95 | 13.61 |
| | 512 | 96.08 | 98.93 | 13.63 |
| regularization weight $\gamma$ | 0.005 | 96.02 | 98.94 | 14.08 |
| | 0.075 | 96.16 | 98.95 | 13.29 |
| | 0.01 | 96.09 | 98.94 | 13.35 |
| | 0.015 | 96.12 | 98.92 | 13.08 |
| | 0.02 | 96.15 | 98.91 | 12.92 |
| constant $r$ | 5 | 96.02 | 98.94 | 14.03 |
| | 7.5 | 96.08 | 98.95 | 13.62 |
| | 10 | 96.09 | 98.94 | 13.35 |
| | 15 | 96.12 | 98.92 | 13.08 |
| | 20 | 96.12 | 98.91 | 12.87 |

Table 15: Robustness of the proposed model to hyperparameters evaluated on CIFAR-100 experiment using WideResNet. DCR: density consistency regularization. CDR: contrastive distribution regularization. RW: regularization weight.

| | FPR95% | | | AUROC | | | AUPR | | |
|---|---|---|---|---|---|---|---|---|---|
| **batch size** | 128 | 256 | 512 | 128 | 256 | 512 | 128 | 256 | 512 |
| iSUN | 1.81±0.31 | 2.34±0.32 | 2.74±0.40 | 99.52±0.06 | 99.41±0.05 | 99.35±0.07 | 99.90±0.01 | 99.87±0.01 | 99.86±0.02 |
| LSUN(R) | 0.87±0.18 | 1.08±0.21 | 1.24±0.20 | 99.72±0.03 | 99.68±0.03 | 99.66±0.03 | 99.94±0.01 | 99.93±0.01 | 99.93±0.22 |
| LSUN(C) | 5.38±0.27 | 5.67±0.40 | 5.75±0.30 | 98.81±0.05 | 98.78±0.05 | 98.78±0.05 | 99.75±0.01 | 99.74±0.01 | 99.74±0.01 |
| CIFAR-10 | 89.52±0.60 | 88.49±0.62 | 88.56±0.50 | 69.79±0.50 | 69.68±0.55 | 69.88±0.53 | 90.97±0.20 | 91.10±0.21 | 91.09±0.01 |
| ImageNet(R) | 2.50±0.20 | 3.34±0.24 | 3.73±0.35 | 99.30±0.04 | 99.11±0.06 | 99.02±0.07 | 99.84±0.01 | 99.80±0.02 | 99.77±0.02 |
| ImageNet(C) | 2.54±0.23 | 3.48±0.32 | 3.74±0.28 | 99.29±0.06 | 99.10±0.06 | 99.07±0.06 | 99.84±0.02 | 99.80±0.02 | 99.79±0.02 |
| Gaussian | 0.00±0.00 | 0.00±0.00 | 0.00±0.00 | 99.96±0.00 | 99.96±0.00 | 99.95±0.00 | 99.99±0.00 | 99.99±0.00 | 99.99±0.00 |
| Places365 | 11.72±0.66 | 11.27±0.61 | 11.58±0.72 | 97.60±0.12 | 97.65±0.10 | 97.69±0.11 | 99.47±0.03 | 99.49±0.03 | 99.50±0.03 |
| Textures | 12.78±0.72 | 12.40±0.70 | 12.93±0.49 | 97.31±0.11 | 97.24±0.11 | 97.13±0.11 | 99.39±0.03 | 99.37±0.03 | 99.34±0.03 |
| iNaturalist | 19.08±0.69 | 17.10±0.57 | 18.14±0.62 | 96.59±0.08 | 96.78±0.09 | 96.71±0.09 | 99.28±0.02 | 99.32±0.02 | 99.31±0.02 |
| SVHN | 1.53±0.25 | 1.68±0.31 | 1.56±0.22 | 99.70±0.04 | 99.63±0.05 | 99.66±0.05 | 99.93±0.01 | 99.91±0.01 | 99.92±0.01 |
| average | 13.43 | 13.35 | 13.63 | 96.14 | 96.09 | 96.08 | 98.94 | 98.94 | 98.93 |
| **RW** | 0.005 | 0.01 | 0.02 | 0.005 | 0.01 | 0.02 | 0.005 | 0.01 | 0.02 |
| iSUN | 2.86±0.27 | 2.34±0.32 | 1.38±0.24 | 99.36±0.05 | 99.41±0.05 | 99.58±0.05 | 99.86±0.01 | 99.87±0.01 | 99.91±0.01 |
| LSUN(R) | 1.33±0.23 | 1.08±0.21 | 0.70±0.16 | 99.67±0.04 | 99.68±0.03 | 99.76±0.03 | 99.93±0.01 | 99.93±0.01 | 99.95±0.01 |
| LSUN(C) | 6.84±0.38 | 5.67±0.40 | 4.88±0.27 | 98.68±0.05 | 98.78±0.05 | 98.88±0.05 | 99.73±0.01 | 99.74±0.01 | 99.76±0.01 |
| CIFAR-10 | 89.38±0.37 | 88.49±0.62 | 88.08±0.50 | 69.20±0.54 | 69.68±0.55 | 69.35±0.59 | 91.11±0.20 | 91.10±0.21 | 90.59±0.24 |
| ImageNet(R) | 3.58±0.26 | 3.34±0.24 | 1.88±0.20 | 99.20±0.06 | 99.11±0.06 | 99.42±0.05 | 99.82±0.02 | 99.80±0.02 | 99.87±0.02 |
| ImageNet(C) | 4.09±0.29 | 3.48±0.32 | 2.28±0.21 | 99.14±0.03 | 99.10±0.06 | 99.40±0.03 | 99.82±0.01 | 99.80±0.02 | 99.87±0.01 |
| Gaussian | 0.00±0.00 | 0.00±0.00 | 0.00±0.00 | 100.0±0.00 | 99.96±0.00 | 99.92±0.00 | 100.0±0.00 | 99.99±0.00 | 99.99±0.00 |
| Places365 | 13.86±0.63 | 11.27±0.61 | 10.40±0.56 | 97.42±0.07 | 97.65±0.10 | 97.75±0.10 | 99.45±0.02 | 99.49±0.03 | 99.50±0.03 |
| Textures | 13.42±0.67 | 12.40±0.70 | 12.08±0.53 | 97.12±0.07 | 97.24±0.11 | 97.38±0.11 | 99.34±0.02 | 99.37±0.03 | 99.40±0.03 |
| iNaturalist | 17.14±0.56 | 17.10±0.57 | 19.10±0.65 | 96.94±0.08 | 96.78±0.09 | 96.47±0.09 | 99.37±0.02 | 99.32±0.02 | 99.25±0.02 |
| SVHN | 2.34±0.21 | 1.68±0.31 | 1.33±0.22 | 99.50±0.06 | 99.63±0.05 | 99.71±0.04 | 99.88±0.02 | 99.91±0.01 | 99.93±0.01 |
| average | 14.08 | 13.35 | 12.92 | 96.02 | 96.09 | 96.15 | 98.94 | 98.94 | 98.91 |
| **constant $r$** | 0.005 | 0.01 | 0.02 | 0.005 | 0.01 | 0.02 | 0.005 | 0.01 | 0.02 |
| iSUN | 2.84±0.28 | 2.34±0.32 | 1.48±0.21 | 99.36±0.05 | 99.41±0.05 | 99.56±0.05 | 99.86±0.01 | 99.87±0.01 | 99.91±0.01 |
| LSUN(R) | 1.34±0.21 | 1.08±0.21 | 0.68±0.15 | 99.66±0.04 | 99.68±0.03 | 99.77±0.03 | 99.93±0.01 | 99.93±0.01 | 99.95±0.01 |
| LSUN(C) | 6.78±0.39 | 5.67±0.40 | 4.98±0.40 | 98.68±0.05 | 98.78±0.05 | 98.85±0.06 | 99.73±0.01 | 99.74±0.01 | 99.75±0.02 |
| CIFAR-10 | 89.38±0.40 | 88.49±0.62 | 88.17±0.52 | 69.21±0.54 | 69.68±0.55 | 69.32±0.57 | 91.13±0.20 | 91.10±0.21 | 90.63±0.23 |
| ImageNet(R) | 3.53±0.27 | 3.34±0.24 | 2.26±0.25 | 99.20±0.06 | 99.11±0.06 | 99.36±0.06 | 99.83±0.02 | 99.80±0.02 | 99.85±0.02 |
| ImageNet(C) | 4.06±0.25 | 3.48±0.32 | 2.48±0.25 | 99.14±0.03 | 99.10±0.06 | 99.26±0.05 | 99.82±0.01 | 99.80±0.02 | 99.83±0.02 |
| Gaussian | 0.00±0.00 | 0.00±0.00 | 0.00±0.00 | 100.0±0.00 | 99.96±0.00 | 99.92±0.00 | 100.0±0.00 | 99.99±0.00 | 99.99±0.00 |
| Places365 | 13.80±0.62 | 11.27±0.61 | 10.29±0.64 | 97.42±0.07 | 97.65±0.10 | 97.70±0.10 | 99.45±0.02 | 99.49±0.03 | 99.49±0.03 |
| Textures | 13.29±0.68 | 12.40±0.70 | 11.80±0.60 | 97.12±0.07 | 97.24±0.11 | 97.35±0.14 | 99.34±0.02 | 99.37±0.03 | 99.39±0.04 |
| iNaturalist | 17.00±0.56 | 17.10±0.57 | 18.12±0.66 | 96.94±0.08 | 96.78±0.09 | 96.56±0.09 | 99.37±0.02 | 99.32±0.02 | 99.26±0.02 |
| SVHN | 2.34±0.22 | 1.68±0.31 | 1.30±0.22 | 99.50±0.06 | 99.63±0.05 | 99.70±0.04 | 99.88±0.02 | 99.91±0.01 | 99.93±0.01 |
| average | 14.03 | 13.35 | 12.87 | 96.02 | 96.09 | 96.12 | 98.94 | 98.94 | 98.91 |

Table 16: Contribution of the proposed regularization terms to OOD detection performance evaluated on CIFAR-100 experiment using WideResNet. DCR: density consistency regularization. CDR: contrastive distribution regularization.

| | FPR95% | | | AUROC | | | AUPR | | |
|---|---|---|---|---|---|---|---|---|---|
| DCR | - | - | ✓ | - | - | ✓ | - | - | ✓ |
| CDR | - | ✓ | ✓ | - | ✓ | ✓ | - | ✓ | ✓ |
| iSUN | 80.69±1.01 | 50.56±0.80 | 2.34±0.32 | 79.19±0.19 | 89.94±0.18 | 99.41±0.05 | 94.98±0.08 | 97.79±0.05 | 99.87±0.01 |
| LSUN(R) | 78.87±0.73 | 47.46±0.43 | 1.08±0.21 | 79.18±0.28 | 90.85±0.20 | 99.68±0.03 | 94.95±0.10 | 98.02±0.05 | 99.93±0.01 |
| LSUN(C) | 72.50±0.88 | 24.96±0.66 | 5.67±0.40 | 83.32±0.33 | 95.19±0.10 | 98.78±0.05 | 96.11±0.09 | 98.95±0.02 | 99.74±0.01 |
| CIFAR-10 | 79.79±1.46 | 82.74±1.12 | 88.49±0.62 | 77.25±0.51 | 73.73±0.49 | 69.68±0.55 | 94.26±0.16 | 93.08±0.16 | 91.10±0.21 |
| ImageNet(R) | 79.36±0.59 | 51.81±0.94 | 3.34±0.24 | 77.64±0.38 | 88.34±0.36 | 99.11±0.06 | 94.32±0.16 | 97.34±0.11 | 99.80±0.02 |
| ImageNet(C) | 72.65±0.74 | 45.15±1.11 | 3.48±0.32 | 81.20±0.34 | 90.64±0.32 | 99.10±0.06 | 95.35±0.12 | 97.92±0.09 | 99.80±0.02 |
| Gaussian | 66.77±0.96 | 0.02±0.03 | 0.00±0.00 | 91.42±0.11 | 99.96±0.01 | 99.96±0.00 | 98.36±0.02 | 99.99±0.00 | 99.99±0.00 |
| Places365 | 81.08±0.98 | 39.18±1.03 | 11.27±0.61 | 76.92±0.55 | 92.28±0.23 | 97.65±0.10 | 94.26±0.15 | 98.30±0.06 | 99.49±0.03 |
| Textures | 78.60±0.88 | 37.33±1.02 | 12.40±0.70 | 77.73±0.43 | 92.41±0.25 | 97.24±0.11 | 94.23±0.13 | 98.31±0.06 | 99.37±0.03 |
| iNaturalist | 71.80±0.84 | 43.39±0.86 | 17.10±0.57 | 82.93±0.40 | 91.76±0.19 | 96.78±0.09 | 95.86±0.15 | 98.20±0.06 | 99.32±0.02 |
| SVHN | 85.33±0.63 | 34.94±0.86 | 1.68±0.31 | 73.82±0.36 | 94.18±0.17 | 99.63±0.05 | 93.56±0.10 | 98.77±0.05 | 99.91±0.01 |
| average | 77.04 | 41.59 | 13.35 | 80.05 | 90.84 | 96.09 | 95.11 | 97.88 | 98.94 |

## A.9 Overlapping object between ID and OOD datasets in ImageNet experiment

In the ImageNet experiment, we observed that there are numerous semantically overlapping objects (SOO) between ImageNet (ID dataset) and iNaturalist, Places365 or Textures, although we have used the conceptually disjoint subsets of Places365 and iNaturalist provided by previous study [3] for model evaluation, in which researchers had removed images whose category conceptually related to ImageNet dataset from iNaturalist and Places365. However, categorical label irrelevance does not imply object discrepancy or distribution separability. For example, in the Places365 dataset, images of the field wild category contain lions, dragonflies, snow leopards and other objects that appear in the ImageNet dataset. The forest road and underwater (ocean deep) images in Places365 include trolleybus and turtles seen in ImageNet. In the Textures dataset, images of bubbly, cobwebbed, frilly, etc., contain objects such as bubbles, spider webs, head cabbages, etc., annotated in ImegeNet. The iNaturalist dataset also includes many labeled objects in ImageNet, such as lycaenid butterflies, sulfur butterflies, daisies, bees, etc. Fig.1, 2, and 3 show a few typical examples. Such a large number of overlapping objects appears to be the reason why our method performs relatively poorly on Places365, Textures and iNaturalist datasets in the ImageNet experiment.

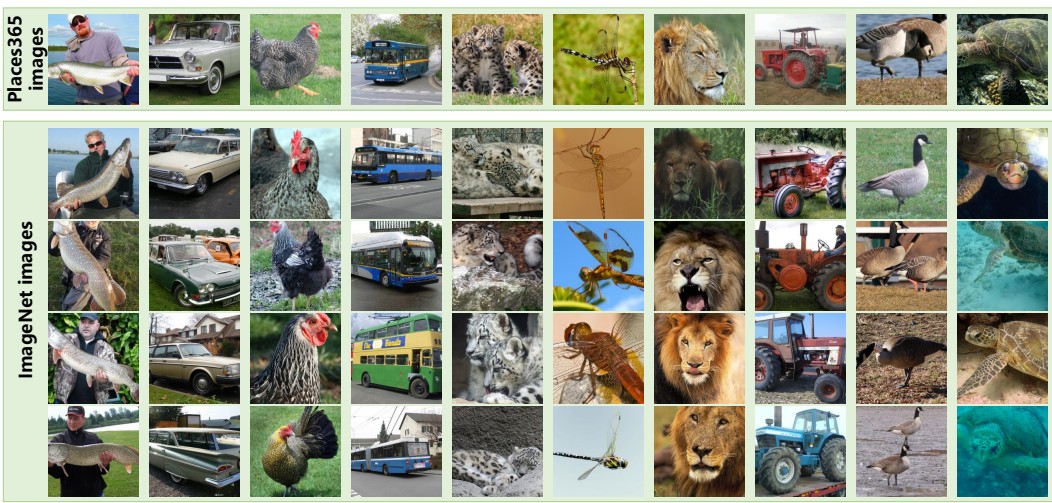

Figure 1: Examples of semantically overlapping objects between ImageNet and Places365 datasets

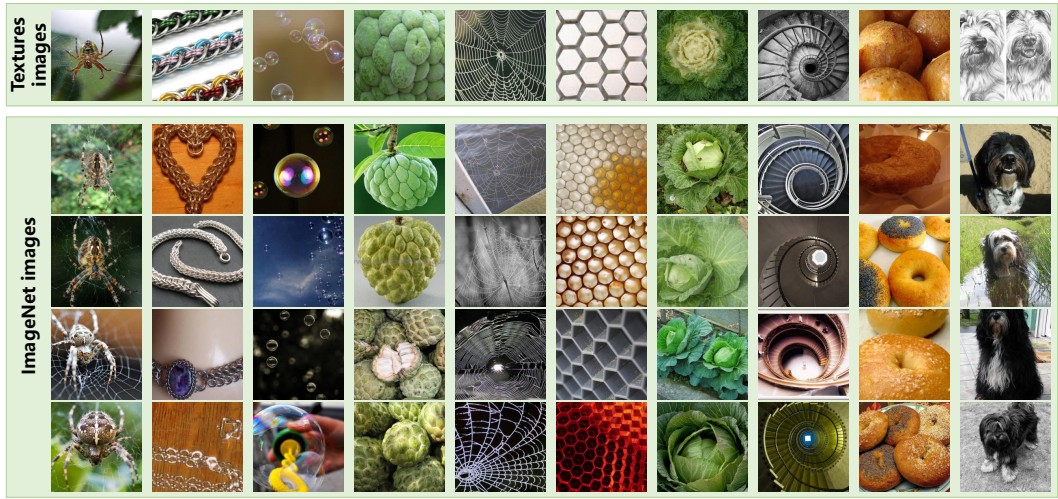

Figure 2: Examples of semantically overlapping objects between ImageNet and Textures datasets

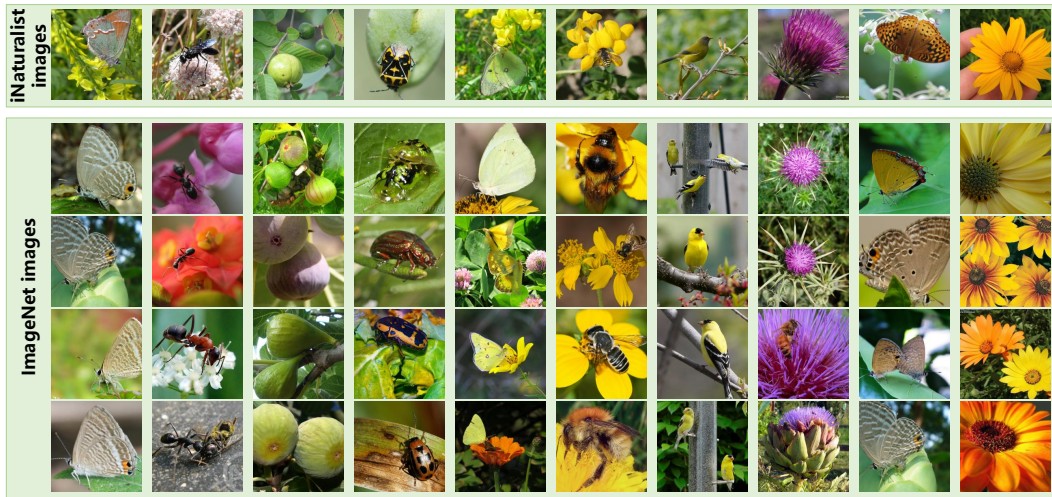

Figure 3: Examples of semantically overlapping objects between ImageNet and iNaturalist datasets

Furthermore, we automatically extracted potential SOO images with relatively higher density values $p(x)$ from three test OOD dataset, followed by manual confirmation and identification of SOO samples, of which a subset of the SOO images (∼2000 images) was finally selected. Fig.4 shows the distribution of log-likelihood values $\log(p(x))$ of the SOO samples and the remaining testing samples from OOD datasets. It can be observed that the SOO images yield a relatively lower OOD score (higher density value), which also validates the effectiveness of the proposed method.

## A.10 Computation efficiency comparison to memory-intensive approaches

The inference time and memory requirements for comparing our method with two memory-intensive approaches, i.e., Gram [4] and pNML [5] are listed below in Table 17 (evaluating 5000 images in our ImageNet experiment, PyTorch framework with Nvidia A100 GPU). It worth mentioning that another evaluated method OECC [6] in our manuscript had similar memory requirement as the Gram method since they combined Gram features for OOD detection.

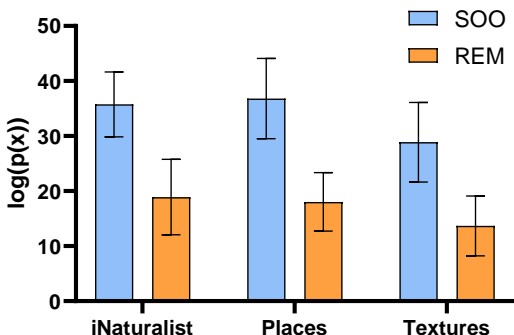

Figure 4: Log-likelihood value (inverse OOD score) distributions $\log(p(x))$ of images with identified semantically overlapping objects (SOO) and remaining images (REM) in the three test OOD datasets.

Table 17: Computation cost comparison

|                     | Ours     | pNML     | Gram     |
| ------------------- | -------- | -------- | -------- |
| GPU Memory (MB)     | 4289     | 7949     | 25598    |
| Inference time (ms) | 53506.12 | 49374.42 | 74044.12 |