# OpenReview forum: "Density-driven Regularization for Out-of-distribution Detection"
_NeurIPS.cc/2022/Conference — NeurIPS 2022 Accept_

### Official Review · Reviewer_gXcr · 2022-07-11

**Rating:** 4
**Confidence:** 5
**Soundness:** 2 fair
**Presentation:** 2 fair
**Contribution:** 2 fair

**Summary:**

The paper proposes to improve OOD detection with energy score [1] by incorporating a density-based regularizer during training that promotes better estimate of P(Y). In particular, the regularizer is derived based on the rejection criteria of a hypothesis test to ensure the consistency of Monte Carlo mean of P(y) (estimated per batch) and the empirical mean in the training set (n_y/N).




[1] Liu et al., Energy-based out-of-distribution detection, NIPS 2020.

**Questions:**

It would be great if the authors can provide further clarifications on the above Weaknesses points and the following two points:

- It is unclear why ImageNet(R)  and ImageNet(C) can be used as OOD sets for CIFAR (ID). If OOD is defined based on the label, ImageNet contains categories with overlapping semantics with CIFAR and therefore may not serve as valid OOD test sets.

- The reported results for energy score seem much worse than the numbers in the original paper [1] under the same architecture (WideResNet). Are there any potential reasons for the discrepancy?


[1] Liu et al., Energy-based out-of-distribution detection, NIPS 2020.

**Limitations:**

The paper included discussions on limitations.

**Strengths And Weaknesses:**

Strengths
- The motivation that existing OOD detection scores suffer from poor density estimation is clear and the task is important.
- The proposed method based on law of large numbers and central limit theorem is simple.

Weaknesses
- The rationale behind the proposed method is unclear to me. Currently the works promotes Monte Carlo estimate of P(Y=y) to be closer to n_y/N in the training set. However, during test time, this assumption may not hold: OOD detection concerns with label shift and it is unclear why one wants to overfit the density of label distribution in the training set.
- While both Lemma 1 and Lemma 2 hold when N goes to infinity,  the batch size used in practice is 256, where Monte Carlo estimate can result in large variance and unreliable estimation. Therefore, there exists significant gaps between the proposed theoretical insights and practical implementation.
- While y is in low dimensional space, obtaining P(Y) requires integrating over x, which are high dimensional vectors and therefore the effects of dimensionality on the accuracy of estimation is non-negligible.

---

> ### Author Response · Authors · 2022-08-02
> **Response to Reviewer (part 1)**
>
> **Comment #1**:  The rationale behind the proposed method is unclear to me. Currently the works promotes Monte Carlo estimate of P(Y=y) to be closer to n_y/N in the training set. However, during test time, this assumption may not hold: OOD detection concerns with label shift and it is unclear why one wants to overfit the density of label distribution in the training set.
>
> **RE**: We would like to emphasize that our detection of OOD is directly achieved by the estimation of $p(x)$. Our estimation of $p(y)$ is to calibrate $p(x)$, and it is NOT used in the testing phase. So we don’t need this assumption about label distribution during the testing phase. Additionally, we want to explain that we are not overfitting $p(Y=y)$ to $n_y/N$, but strictly prove the deviation range (Lemmas 1 and 2 in manuscript) between them and derive an adaptive regularization term based on asymptotic statistical testing to ensure the theoretical consistency between them.
>
> To help the reviewer better understand our method, here we explain our DCR regularization more intuitively (similar explanation see lines 76-90 in manuscript). It is known that supervised classification models generally fit $p(y|x)$, not $p(x)$. Therefore directly estimating $p(x)$ for OOD detection from high-level features from pre-trained networks can be biased. To tackle this problem, we consider label distribution $Y$ is a one-dimensional discrete random variable whose empirical density can be reliably estimated. From the theoretical point of view, we pointed out that analytical density $p(y)= \int p(x, y)dx= \int p(x)p(y |x)dx$ should be consistent with the observed empirical batch-wise density of $Y$. With this inherent property, we developed our adaptive density regularization (DCR) to enforce the consistency between the empirical and analytical density of $Y$. Therefore, this constraint implicitly helps learn both $p(x)$ and $p(y|x)$. Our experimental results also verified this DCR regularization can provide a better $p(x)$ estimate for OOD detection.
>
> **Comment #2**:  While both Lemma 1 and Lemma 2 hold when N goes to infinity, the batch size used in practice is 256, where Monte Carlo estimate can result in large variance and unreliable estimation. Therefore, there exists significant gaps between the proposed theoretical insights and practical implementation.
>
> **RE**: We have considered the influence of this factor in the manuscript by analyzing the model performances with different $N$ values in the ablation study. The results showed that the model performance is excellent for different $N$ values compared to traditional approaches, and performance variation is relatively small when setting $N$ in a certain typical range, as compared to performance variation across different models in Table 1.
>
> **Comment #3**:  While y is in low dimensional space, obtaining P(Y) requires integrating over x, which are high dimensional vectors and therefore the effects of dimensionality on the accuracy of estimation is non-negligible.
>
> **RE**: In Eq. (7) in the manuscript, we have derived the formula stating that computing the integral over $X$ can be approximated by the Monte Carlo method (the large law of numbers) in the form of the averaged exponential of logits, rather than calculating high-dimensional integration over $X$ directly. So the high-dimensional integration over $X$ is not required both in theory and practice.

---

> ### Author Response · Authors · 2022-08-02
> **Response to Reviewer (part 2)**
>
> **Comment #4**:  It is unclear why ImageNet(R) and ImageNet(C) can be used as OOD sets for CIFAR (ID). If OOD is defined based on the label, ImageNet contains categories with overlapping semantics with CIFAR and therefore may not serve as valid OOD test sets.
>
> **RE**: Thanks for your question. We follow this experimental setting because many related works [1-5] were conducted on these benchmarks, i.e., considering ImageNet(C) and ImageNet(R) as the OOD datasets for the CIFAR dataset. As suggested, we have carefully checked the labels between CIFAR and TinyImageNet, from which ImageNet(R) and ImageNet(C) are generated. Few class labels are overlapping, i.e., 2% and 11% of TinyImageNet class labels have text overlap with CIFAR10 and CIFAR-100 class labels, respectively. We further check the image instances and find they are from different domains. From this perspective, they could be served as OOD dataset even if they share class labels as done in previous studies. Moreover, we will take a more strict step to update the result by excluding the images with overlapping text labels to CIFAR labels from TinyImageNet.
>
> [1] Liang S, Li Y, Srikant R. Enhancing the reliability of out-of-distribution image detection in neural networks, ICLR, 2018.
>
> [2] Sastry C S, Oore S. Detecting out-of-distribution examples with gram matrices, ICML, 2020.
>
> [3] Bibas K, et al. Single Layer Predictive Normalized Maximum Likelihood for Out-of-Distribution Detection, NeurIPS, 2021.
>
> [4] Lee K, et al. A simple unified framework for detecting out-of-distribution samples and adversarial attacks, NeurIPS, 2018.
>
> [5] Papadopoulos A A, et al. Outlier exposure with confidence control for out-of-distribution detection[J]. Neurocomputing, 2021, 441: 138-150.
>
> **Comment #5**:  The reported results for energy score seem much worse than the numbers in the original paper [1] under the same architecture (WideResNet). Are there any potential reasons for the discrepancy?
>
> **RE**: Please note that we have two versions of the Energy model (for both Table 1 in the manuscript and Tables 1-5 in supplementary material), one w/o outlier exposure and another one w/ outlier exposure (please notice the third column in the right part of the tables). Our outlier exposure version, not the w/o exposure version, corresponds to the fine-tuned model of the original Energy paper [1]. For the fine-tuned version, the researchers used an additional dataset with millions of images as the outlier exposure dataset to fine-tune the energy model. Our reproduced Energy model w/o outlier exposure corresponds to the no fine-tune version in the original Energy paper [1]. Both reproduced Energy models (w/ and w/o outlier exposure) agree very well with the original published results [1].
>
> [1] Liu W, Wang X, Owens J, et al. Energy-based out-of-distribution detection, NeurIPS, 2020.

---

> ### Author Response · Authors · 2022-08-08
> **To Reviewer gXcr: would you please kindly reconsider your rating based on our response.**
>
> Thanks very much for your comments. Regarding your questions, we have provided detailed point-to-point responses, which we believe can address your major concerns and clarify the issues raised. We wonder whether you would kindly reconsider your rating based on our responses to your questions. Your consideration is highly appreciated. Many thanks!
>
> Authors of the Paper6858

---

> ### Comment · Reviewer_gXcr · 2022-08-09
> **Thanks for the responses. I'm still concerned on the validity and soundness.**
>
> I sincerely appreciate the authors responses. My questions on the dataset validity and results on energy score are solved.  However, after carefully reading the manuscript and the authors' responses again,  I'm still not convinced on the validity and soundness of the proposed method.
>
> 1. One key to the paper is to calibrate the density estimate of y to make the model output after softmax closer to the empirical estimate of y in the training set ($\hat p(y) = n_y / N$), where $N = \sum_{y=1}^L n_y$, and $L$ is the total number of in-distribution classes (Lemma 1). One fundamental issue is that in the test set, label shift can occur and the density estimate of y can be severely impacted. For example, in the training set, there are 10 balanced ID classes, while in the test set, 1) some in-distribution (ID) classes are imbalanced; 2) even if the ID classes are balanced (this assumption is not made explicitly in the manuscript), the existence of OOD samples will decrease the density of ID.
>
> 2. The authors argue that "OOD score should be constructed from P(X) rather than P (Y |X )" (L149-150), but the equation (4) for the density of x is **based on the closed-world** assumption. As the task is OOD detection, one should instead consider the open world setting where $y \in$ { 1,2,... L, L+1, L+2, ...,} instead of simply using $L$.  In other words, $k(x)=\int g(x, Y) d \pi_{Y} \neq \frac{1}{Z} \sum_{1 \leq y \leq L} \exp \left(h^{y}(x)\right)$ which is actually $P(x | x\text{ is ID})$ [1]
>
> 3. Neither variance nor error bars are reported for the experiments (e.g. Table 1), which makes it hard to justify that the reported results are indeed statistically significant, stable, and reproducible.
>
>
> Given the above fundamental concerns, I cannot justify the acceptance of the paper given the current state.
>
>
>
>  [1] Hsu et al., Generalized ODIN: Detecting Out-of-distribution Image without Learning from Out-of-distribution Data, CVPR 2020.

---

> > ### Public Comment · ~xiaoyuan_Guan1 · 2022-12-10
> > **a little question for theoretical logic**
> >
> > the author claims the restraint (DCR) would benefit for Eq (3) to be satisfied, but in this paper, the lemma 1 and lemma 2 have in common to assume Eq 3 has been satisfied. In another words, the first contribution can not be clear to have positive impact on Eq 3 to be satisfied, it is a nessesary condiction.

---

> > > ### Public Comment · ~Wenjian_Huang1 · 2022-12-10
> > > **Response**
> > >
> > > Thanks for your interest in our work. The motivation of lemma1 and lemma2 is to derive some necessary conditions of Eq.(3), i.e., when Eq.(3) is assumed to hold, so we can turn the necessary conditions into regularization term to better ensure the establishment of the preassumed condition Eq.(3). Hope this explanation clears up your doubts.

---

### Official Review · Reviewer_fnQU · 2022-07-11

**Rating:** 7
**Confidence:** 5
**Soundness:** 3 good
**Presentation:** 3 good
**Contribution:** 3 good

**Summary:**

This paper highlights the drawbacks of current OOD detection methods that most of them try to model ID p(x) using the features that are intially trained to fit p(y|x).
To solve the distribution misalignment, a novel density-consistency regularizaiton is proposed.
A contrastive criterion is also used to flatten p(x) so that OOD samples can locate in low-density area to be identified easily.
The proposed method is evaluated upon both OE and non-OE settings, and the performance exceeds the previous methods in a large margin.

**Questions:**

- How is Mah implemented? Only use the last layer feature to compute mah dist or use features from multiple layers and use validation set to learn optimal weight combination of different feature layers, as the original paper suggest?

**Limitations:**

The authors do not include the discussion on the limitation of the method.
We hope the authors can provide more analysis on its weakness in discusssion.

**Strengths And Weaknesses:**

Pro:
- The paper is well-motivated, and the methods can address the problem intuitively.
- The method rely on proven assumptions with theoretical proof.
- The paper obtains significant great results across various experiments with fixed hyperparameters.


Con:
- The results in Table 1 are super charming. However, for CIFAR-100 experiments, I am more interested in the results when CIFAR-10 is OOD, as CIFAR-10 and CIFAR-100 are generally looks alike, but they are from different label distribution. This result usually clearly show how the method capture clues to distinguish ID/OOD.
In CV, OOD samples are usually considered as semantic anomalies to ID datasets. However, a common shortcoming of current OOD detection evaluation protocols use different datasets as OOD, so that some methods can use low-level superficial shortcut to make good results (e.g., a simple low/high-resolution image classifier might do extra well on some OOD datasets). But actually, we hope the model uses semantic difference to distinguish ID/OOD. I hope the proposed method gets the better result not by superficial shortcut. The easiest way to proof is to show (ID: CIFAR-10, OOD: CIFAR-100) and (ID: CIFAR-100, OOD: CIFAR-10) cases.
- The paper points out the semantically overlapping objects between ImageNet and iNaturalist. In some other works such as `ViM: Out-Of-Distribution with Virtual-logit Matching`, `https://github.com/hendrycks/natural-adv-examples`, `Scaling Out-of-Distribution Detection for Real-World Settings`, some cleaner OOD datasets for ImageNet are provided. Authors are suggested to do experiments on these clean ood data. Or to support the reason that the proposed method does not do well on Texture / iNaturalist due to the label noise, the author shall show the appropriated sampled semantically-overlapping objects get high confidence.
- It would be great if the authors provide more visualization or deeper analysis on what DCR and CDR impact (maybe on the feature space) to help readers better understand the method.

---

> ### Author Response · Authors · 2022-08-02
> **Response to Reviewer (part 1)**
>
> **Comment #1**:  The results in Table 1 are super charming. However, for CIFAR-100 experiments, I am more interested in the results when CIFAR-10 is OOD, as CIFAR-10 and CIFAR-100 are generally looks alike, but they are from different label distribution. This result usually clearly show how the method capture clues to distinguish ID/OOD. In CV, OOD samples are usually considered as semantic anomalies to ID datasets. However, a common shortcoming of current OOD detection evaluation protocols use different datasets as OOD, so that some methods can use low-level superficial shortcut to make good results (e.g., a simple low/high-resolution image classifier might do extra well on some OOD datasets). But actually, we hope the model uses semantic difference to distinguish ID/OOD. I hope the proposed method gets the better result not by superficial shortcut. The easiest way to proof is to show (ID: CIFAR-10, OOD: CIFAR-100) and (ID: CIFAR-100, OOD: CIFAR-10) cases.
>
> **RE**: Thanks for your suggestion. This is indeed a good question. We were not aware of this setting before. To the best of our knowledge, none of the compared OOD methods conducted this experiment and reported the results in this setting. We followed existing settings for a fair comparison. However, as suggested, we have now conducted experiments with this specific setting. The results are attached below.
>
> | ID: CIFAR-10; OOD: CIFAR-100; (WideResNet, w/o outlier exposure) | MSP   | Odin  | Mah   | Energy | Gram  | Energy+ReAct | pNML  | Odin+pNML | Ours  |
> |------------------------------------------------------------------|-------|-------|-------|--------|-------|--------------|-------|-----------|-------|
> | AUROC(↑)                                                         | 87.34 | 84.78 | 84.38 | 87.32  | 76.14 | 86.36        | 83.52 | 83.52     | 91.29 |
> | AUPR(↑)                                                          | 96.86 | 95.50 | 96.27 | 96.62  | 92.88 | 96.50        | 95.78 | 95.78     | 98.04 |
> | FPR95(↓)                                                         | 63.50 | 54.05 | 67.85 | 50.20  | 70.68 | 54.00        | 62.00 | 62.00     | 44.95 |
>
>
> | ID: CIFAR-100; OOD: CIFAR-10; (WideResNet, w/o outlier exposure) | MSP   | Odin  | Mah   | Energy | Gram  | Energy+ReAct | pNML  | Odin+pNML | Ours  |
> |------------------------------------------------------------------|-------|-------|-------|--------|-------|--------------|-------|-----------|-------|
> | AUROC(↑)                                                         | 75.96 | 68.89 | 68.84 | 79.31  | 64.53 | 65.81        | 43.12 | 46.92     | 73.53 |
> | AUPR(↑)                                                          | 94.24 | 91.45 | 91.58 | 94.93  | 88.52 | 90.13        | 80.97 | 82.24     | 92.74 |
> | FPR95(↓)                                                         | 81.85 | 85.90 | 93.00 | 77.85  | 90.41 | 87.90        | 98.45 | 97.60     | 87.49 |
>
> It can be seen in the CIFIAR-10 experiment (as ID dataset) that we have achieved the best performance. However, in the CIFAR-100 experiment, all the methods perform poorly in terms of PFR95 in this setting, and most published papers in recent years still perform worse than the baseline approach (MSP). The result indicates that this setting is particularly challenging compared to other settings. Although this might be due to the fact that the two datasets look alike, the result is a bit surprising, and we have not figured out the specific reason behind this due to time limitations. But we will surely investigate this setting in our future work. Finally, It is worth mentioning that our approach still achieves much higher averaged performance when adding CIFAR-10 and CIFAR-100 as the OOD dataset for CIFAR-100 and CIFAR-10 experiments, respectively. The updated averaged FPR95 (over all the OOD datasets) of our method are 5.61 and 12.87 for the CIFAR-10 experiment and CIFAR-100 experiment, respectively. Comparatively, the second best approach (Gram) obtain averaged FPR95 of 13.43 and 27.66 for CIFAR-10 and CIFAR-100 experiments, respectively. We will report this result in our final paper.

---

> > ### Comment · Reviewer_fnQU · 2022-08-07
> > **The author address my concern and the paper should be accepted**
> >
> > The experiments on the difficult CIFAR-10/100 looks good to me, which strongly shows that the method does not use the superficial appearance shortcut for OOD detection but focus on the semantics. As the author suggests, CIFAR-100 is still difficult and shall be explored.
> > We hope the author could try to emphasize why the density-based methods can well capture the semantic difference between ID/OOD in their camera-ready version, so that the community can know deeper about the cool method.

---

> ### Author Response · Authors · 2022-08-02
> **Response to Reviewer (part 2)**
>
> **Comment #2**:  The paper points out the semantically overlapping objects between ImageNet and iNaturalist. In some other works such as ViM: Out-Of-Distribution with Virtual-logit Matching, https://github.com/hendrycks/natural-adv-examples, Scaling Out-of-Distribution Detection for Real-World Settings, some cleaner OOD datasets for ImageNet are provided. Authors are suggested to do experiments on these clean ood data. Or to support the reason that the proposed method does not do well on Texture / iNaturalist due to the label noise, the author shall show the appropriated sampled semantically-overlapping objects get high confidence.
>
> **RE**: Thanks for your comment. We have indeed found that the images in OOD datasets (iNaturalist/Texture) that have semantically overlapping objects with the ID dataset (ImageNet) have higher ID confidence (lower OOD score) than other images without overlapping objects. Specifically, the OOD images from Fig.1 to Fig.3 in our supplementary material, which have overlapping objects with ID datasets but different text labels, all exhibit higher ID confidence (low OOD score) compared to other OOD images. In fact, these image samples with overlapping objects were automatically selected from the OOD dataset according to the criteria of low OOD scores. We will provide more statistical details regarding this issue in our revised paper.
>
> **Comment #3**:  It would be great if the authors provide more visualization or deeper analysis on what DCR and CDR impact (maybe on the feature space) to help readers better understand the method.
>
> **RE**: Thanks for your suggestion. We will provide visual examples concerning the distribution of high-level features, e.g., classification logit, to demonstrate our approach more intuitively in our final paper.
>
> **Comment #4**:  How is Mah implemented? Only use the last layer feature to compute mah dist or use features from multiple layers and use validation set to learn optimal weight combination of different feature layers, as the original paper suggest?
>
> **RE**: We adopted the implementation of the Mah model (w/o logistic regression part) from the Energy paper [1], which used the last layer feature to compute the Mah distance for a fair comparison. We follow this setting because our method does not require any additional outlier or OOD data to learn the OOD detector. Please note for the original Mah code [2], OOD data are directly USED in training logistic regression model for utilizing features from multiple layers (which is essentially a two-class classification problem, ID v.s. OOD). This setting is clearly out of our interest as in our setting (w/o outlier exposure), we do not want the model to see any examples from the OOD dataset.
>
> [1] Liu W, Wang X, Owens J, et al. Energy-based out-of-distribution detection, NeurIPS, 2020.
>
> [2] Lee K, Lee K, Lee H, et al. A simple unified framework for detecting out-of-distribution samples and adversarial attacks, NeurIPS, 2018.
>
> **Comment #5**:  The authors do not include the discussion on the limitation of the method. We hope the authors can provide more analysis on its weakness in discussion.
>
> **RE**: Thanks for your suggestion. We add the following limitation and will include it in our final manuscript: Since our approach requires fine-tuning, this can lead to the problem that there exists a discrepancy in classification probabilities between the original network and the fine-tuned network. This problem can be avoided by using the original and the fine-tuned networks for the classification and OOD detection tasks, respectively. However, it can double the inference time and computation load.

---

### Official Review · Reviewer_9BXS · 2022-07-12

**Rating:** 5
**Confidence:** 3
**Soundness:** 2 fair
**Presentation:** 2 fair
**Contribution:** 2 fair

**Summary:**

A method is presented to fine-tune pretrained networks by introducing regularization terms such that the marginal density $p(y)$ estimated from logits matches the empirical density $n_y/N$. The claim is that this method allows the sum of logit-exponentials $\log(\sum_y \exp(h_y(x)))$ to be used as a score for OOD detection. Effectiveness on various datasets is examined with good results.

**Questions:**

1) In lines 243-245, the OOD score is specified as $\log(\sum_y \exp(h_y(x)))$. Since log is a monotonic function, we can focus instead on the term inside the logarithm i.e. $\sum_y \exp(h_y(x))$, which is basically the sum (or average) of logit exponentials (which is the same as the denominator of the softmax function). Can the authors motivate this intuitively? How and why is it better than using the softmax score, itself? Specifically, how does introducing the regularization terms during training impact the characteristics of the logit such that this term $\log(\sum_y \exp(h_y(x)))$ can serve as an effective OOD score?

2) A related question: it is stated that "existing detectors relying on discriminative probability suffer from the overconfident posterior estimate for OOD data." This is true, particularly for Softmax based OOD scores. This is closely related to the problem of uncertainty calibration and plenty of research has been done on softmax calibration. Can the authors comment if their OOD score, which is also closely derived from logit-exponentials, is better calibrated?

3) Can the authors please clarify how the numbers reported in Table 1 were obtained? Assuming FPR95% means FPR when TPR=95%, then the benchmarks would seem to be performing remarkably poorly, getting false positive rates of well over 70-80% ? If so, these do not seem to agree with what is widely reported in literature for these methods. For example, TNR rates of well over 80% are reported in reference [9] (Mahalanobis), which would imply FPRs of under 20%. The authors should clarify any differences in settings that were used to obtain their results on benchmarks.

4) The distribution deviating operator $\mathcal{S}$ comprises fairly standard transformations such as rotations, crops, etc. Typically these are used to generate positive or similar samples in contrastive learning, but here they are being used to synthetically generate out-of-distribution samples. Wouldn't this result in such inputs (from the in-distribution dataset that have been rotated, cropped, etc.) being classified as OOD?

5) It is claimed that the computational algorithm that is developed, "can be used for any pretrained classification network." Hence, the proposed approach is akin to fine-tuning it's parameters with additional regularization terms. Can the authors comment if they tried training a network from scratch (ie with randomly initialized weights), with their combined loss function?

**Limitations:**

No concerns with potential negative societal impact.

**Strengths And Weaknesses:**



The exposition is heavy on mathematical proofs at the cost of conceptual clarity and readability. A lot of emphasis is placed on various proofs, but the intuition and concept do not come across as clearly. For instance, in a paper on OOD detection, what is used as the OOD score should be clearly emphasized. However, here the fact that $\log(\sum_y \exp(h_y(x)))$ is used as the eventual OOD score is mentioned only fleetingly, and is easy to miss on the first reading.
Some recommendations for improvement:
1) I feel that the training algorithm should be included in the main manuscript instead of the appendix since it clarifies how the various regularization terms are calculated and gives a clearer picture of the overall flow. Some of the details of the lemmas and proofs can be moved to the Appendix.
2) Mathematical notation should be simplified and made consistent to improve readability. For instance, $y$ is used both as the dependent variable in Eq. (2) and also as the index of summation in the denominator. The term $\Upsilon^y_n$ in Eq. (9) seems to be a scalar. However, in Eqs. (12) and (13), the matrix transposition operator is applied to it. Can the authors please clarify how to interpret $\Upsilon^y_n$? Also, notation should be explained where it is introduced. For instance $\mathbb{S}$ is introduced in Eq. (16), but is explained almost a page later on line 274.

The paper is well motivated, though, and the results seem good, but there is a question about benchmarks (explained in the next question before).

---

> ### Author Response · Authors · 2022-08-02
> **Response to Reviewer (part 1)**
>
> **Comment #1**: I feel that the training algorithm should be included in the main manuscript instead of the appendix since it clarifies how the various regularization terms are calculated and gives a clearer picture of the overall flow. Some of the details of the lemmas and proofs can be moved to the Appendix.
>
> **RE**: Due to the page limit, we have already moved all the proof details to the appendix, and only the conclusions of the lemmas are given in the manuscript. As suggested, we will consider moving the pseudo-code into the manuscript in the revision phase.
>
> **Comment #2**: Mathematical notation should be simplified and made consistent to improve readability. For instance, $y$ is used both as the dependent variable in Eq. (2) and also as the index of summation in the denominator. The term $\Upsilon^y_n$ in Eq. (9) seems to be a scalar. However, in Eqs. (12) and (13), the matrix transposition operator is applied to it. Can the authors please clarify how to interpret $\Upsilon^y_n$? Also, notation should be explained where it is introduced. For instance $\mathbb{S}$ is introduced in Eq. (16), but is explained almost a page later on line 274.
>
> **RE**: For Eq.(2),  we used the unified symbol $y$ because both $y$ in the numerator and denominator represent the same meaning, i.e., category indicator. As suggested, we will change $y$ into $y’$ in the denominator to avoid misunderstanding. For Eq.(12) and (13), we use this idiomatic multidimensional expression since it is the general form for transforming Gaussian distribution to chi-square distribution. However, as mentioned, our $\Upsilon^y_n$ is a scalar, so we will remove the transposition operator to avoid misunderstanding as you suggested. For Eq.(16), since symbol $\mathcal s$ had already been defined as a distribution-deviating operator, we considered the notation $\mathcal s \in \mathbb{S}$ implicitly implies that $\mathbb{S}$ is a distribution-deviating operator set. However, as suggested, we will provide an explanation for $\mathbb{S}$ there.

---

> ### Author Response · Authors · 2022-08-02
> **Response to Reviewer (part 2)**
>
> **Comment #3**: In lines 243-245, the OOD score is specified as $\log(\sum_y \exp(h_y(x)))$. Since log is a monotonic function, we can focus instead on the term inside the logarithm i.e. $\sum_y \exp(h_y(x))$, which is basically the sum (or average) of logit exponentials (which is the same as the denominator of the softmax function). Can the authors motivate this intuitively? How and why is it better than using the softmax score, itself? Specifically, how does introducing the regularization terms during training impact the characteristics of the logit such that this term $\log(\sum_y \exp(h_y(x)))$ can serve as an effective OOD score?
>
> **RE**: From the energy-based perspective [1,2,3], $\exp(h_y(y))$ can be considered as $Z \cdot p(y,x)$, where $Z$ is a normalization factor to transform it as the density of a probability measure. Therefore, the energy score $\sum_y \exp(h_y(x))$ can be considered as the unnormalized value for $p(x)$, which can be used to approximate sample density. Comparatively, softmax score $p(y|x)$ represents the posterior probability. Since deep classification neural networks are generally trained to fit $p(y|x)$ by the supervised loss, it can produce over-confidence/over-fitting $p(y|x)$ by the powerful fitting ability even for samples with smaller density value $p(x)$. Thus theoretically, $\sum_y \exp(h_y(x))$ and softmax score are essentially different, where one can be considered as sample density, and the other is the posterior probability prone to over-fitting issue, even though it seems that the difference between them is the summation symbol. Experimentally speaking, typical softmax score-based and energy score-based OOD detectors are Msp [4] and Energy [3], respectively. Specifically, Msp used $\max_y p(y|x)$ and the Energy used $-T \log (\sum_y \exp(h_y(x)/T) )$ to detect OOD samples ($T$ is a hyperparameter). It can be observed from our study (Table 1 in manuscript and Tables 1-6 in supplementary material) and the results in previous study [3] that the energy score-based detector can perform better than the softmax-score-based detector. The reviewer is referred to our manuscript (lines 149-156) and [3] for a detailed explanation.
>
> *It should be pointed out that although we have emphasized that $\log(\sum_y \exp(h_y(x)))$ can be more suitable for OOD detection than softmax score, it is NOT our contribution in this research. In our study, what we want to illustrate is that directly using the energy score as the OOD detector can be biased because the classification model generally only supervises $p(y|x)$, not $p(x)$.* In this study, since label $Y$ is a one-dimensional discrete random variable whose empirical density can be reliably estimated, from the theoretical point of view, we pointed out that $p(y)= \int p(x, y)dx= \int p(x)p(y |x)dx$ should be consistent with the observed empirical batch-wise density of $Y$. With this inherent property, we develop our adaptive density regularization constraints based on asymptotic statistical theories. This constraint implicitly learns both $p(x)$ and $p(y|x)$. Therefore, our regularization term can provide a more accurate estimate of $p(x)$ (see lines 76–90 in the manuscript). Our experimental results also verified the effectiveness of our approaches that the proposed method achieved significantly better performance than traditional energy approaches [3] (see Table 1 in manuscript and Tables 1-6 in supplementary material). Our contribution lies in we reveal that directly estimating density p(x) using high-level network features can be biased, and the proposed regularization (constraining on both $p(x)$ and $p(y|x))$ can significantly improve sample density estimation accuracy for OOD detection. We expect the above explanation can well address your concerns.
>
> [1] Grathwohl W, Wang K C, Jacobsen J H, et al. Your classifier is secretly an energy based model and you should treat it like one, ICLR, 2020.
>
> [2] LeCun Y, Chopra S, Hadsell R, et al. A tutorial on energy-based learning. In Predicting structured data, MIT Press, 2006.
>
> [3] Liu W, Wang X, Owens J, et al. Energy-based out-of-distribution detection, NeurIPS, 2020.
>
> [4] Hendrycks D, Gimpel K. A baseline for detecting misclassified and out-of-distribution examples in neural networks, ICLR, 2017.

---

> > ### Comment · Reviewer_9BXS · 2022-08-10
> > **Thank you for your response**
> >
> > Thank you authors for your detailed response, especially clarifying some of the theoretical aspects which I had stated in my initial review. From a practical perspective, however, what is used as a proxy for $p(x)$ is indeed the sum of logits and it would seem from your argument that this sum of logits, when regularized by the proposed method, is better calibrated than other OOD scores.
> >
> > I still continue to have concerns about the baselines looking worse than reported in literature, however. Granted that ODIN and Mahalanobis based approaches do use OOD data to tune hyperparameters to report their best results. However, Mahalanobis also reported using adversarial samples in lieu of OOD and the results were still much better that what are being reported here. Similarly, ODIN  used a different OOD dataset (not the one being tested on) to tune their hyperparameters and reported the relative insensitivity of their method to the choice of this third dataset.
> >
> > Nonetheless, I am updating my rating.

---

> ### Author Response · Authors · 2022-08-02
> **Response to Reviewer (part 3)**
>
> **Comment #4**: A related question: it is stated that "existing detectors relying on discriminative probability suffer from the overconfident posterior estimate for OOD data." This is true, particularly for Softmax based OOD scores. This is closely related to the problem of uncertainty calibration and plenty of research has been done on softmax calibration. Can the authors comment if their OOD score, which is also closely derived from logit-exponentials, is better calibrated?
>
> **RE**: We are fully aware of this issue that there are lots of research efforts on uncertainty calibration, e.g., the Platt scaling in SVM for unconstrained output and temperature scaling for constrained softmax score. Some confidence/uncertainty-calibrating tricks for softmax [1,2] indeed had been used in previous OOD detection approaches, e.g., Odin [3], Energy [4], React [5], and Mah [6]. We would like to emphasize that our approach differs from these empirical approaches by providing a probabilistic theoretical framework based on exploring inherent density properties (consistency and contrastive properties) and finally derives analytical regularization term Eq.(17) with rigorous proofs.
> Moreover, we have also indeed compared our method with these traditional approaches, such as Odin [3], Energy [4], React [5], Mah [6], and Energy+React (see Table 1 and more Tables 1-6 in supplementary material), in which calibration tricks have been used. The result shows that our proposed method generally achieved significantly better performances, demonstrating our OOD score is better calibrated.
>
> [1] Minderer M, Djolonga J, Romijnders R, et al. Revisiting the calibration of modern neural networks. NeurIPS, 2021.
>
> [2] Guo C, Pleiss G, Sun Y, et al. On calibration of modern neural networks, ICML, 2017.
>
> [3] Liang S, Li Y, Srikant R. Enhancing the reliability of out-of-distribution image detection in neural networks, ICLR, 2018.
>
> [4] Liu W, Wang X, Owens J, et al. Energy-based out-of-distribution detection, NeurIPS, 2020.
>
> [5] Sun Y, Guo C, Li Y. React: Out-of-distribution detection with rectified activations, NeurIPS, 2021.
>
> [6] Lee K, Lee K, Lee H, et al. A simple unified framework for detecting out-of-distribution samples and adversarial attacks, NeurIPS, 2018.
>
> **Comment #5**: Can the authors please clarify how the numbers reported in Table 1 were obtained? Assuming FPR95% means FPR when TPR=95%, then the benchmarks would seem to be performing remarkably poorly, getting false positive rates of well over 70-80% ? If so, these do not seem to agree with what is widely reported in literature for these methods. For example, TNR rates of well over 80% are reported in reference [9] (Mahalanobis), which would imply FPRs of under 20%. The authors should clarify any differences in settings that were used to obtain their results on benchmarks.
>
> **RE**: Yes, FPR95% means FPR when TPR=95% (see lines 280-281 in manuscript). This metric has been widely used in previous OOD detection works. As suggested, we will explain this metric more specifically in our revised paper.
> Regarding your benchmark difference issue, please note that in the original Mah (Mahalanobis) code [1]  directly USED OOD data to tune many hyperparameters in input preprocessing and logistic regression of feature ensemble (which is essentially a two-class classification problem, ID v.s. OOD). This setting is clearly out of our interest as in our setting (w/o outlier or OOD exposure), we do not want the model to see any examples from the OOD dataset. Therefore we adopted another implementation of the Mah model (w/o logistic regression part) from [2] for a fair comparison.
>
> [1] Lee K, Lee K, Lee H, et al. A simple unified framework for detecting out-of-distribution samples and adversarial attacks, NeurIPS, 2018.
>
> [2] Liu W, Wang X, Owens J, et al. Energy-based out-of-distribution detection, NeurIPS, 2020.

---

> ### Author Response · Authors · 2022-08-02
> **Response to Reviewer (part 4)**
>
> **Comment #6**: The distribution deviating operator $\mathcal{S}$ comprises fairly standard transformations such as rotations, crops, etc. Typically these are used to generate positive or similar samples in contrastive learning, but here they are being used to synthetically generate out-of-distribution samples. Wouldn't this result in such inputs (from the in-distribution dataset that have been rotated, cropped, etc.) being classified as OOD?
>
> **RE**: We are fully aware of this issue. Therefore we empirically set image transformation with a large distortion range as the distribution-deviating operator to avoid producing too similar images compared to In-distribution (ID) dataset as the distribution-deviated samples. Our experimental results have also verified that this approach helps improve OOD detection performance. Some related work also showed that simple image transformations/augmentations, e.g., rotation, can be used to produce distribution deviated data, e.g. [1,2], and “hard” augmentations, e.g., rotation, were known to be harmful to the standard contrastive learning [3]. We would like to highlight that one of our main contributions is proposing a specific regularization term, from a theoretical point of view, to calibrate density value by using distribution deviated samples. Surely, our theoretical approach can be used to explore more useful transformations. But currently, we think extensive analysis of different diverse transformations is out of the score of this paper.
>
> [1] Koh P W, Sagawa S, Marklund H, et al. Wilds: A benchmark of in-the-wild distribution shifts, ICML, 2021.
>
> [2] Tack J, Mo S, Jeong J, et al. Csi: Novelty detection via contrastive learning on distributionally shifted instances. NeurIPS, 2020.
>
> [3] Chen T, Kornblith S, Norouzi M, et al. A simple framework for contrastive learning of visual representations, ICML, 2020.
>
> **Comment #7**:  It is claimed that the computational algorithm that is developed, "can be used for any pretrained classification network." Hence, the proposed approach is akin to fine-tuning it's parameters with additional regularization terms. Can the authors comment if they tried training a network from scratch (ie with randomly initialized weights), with their combined loss function?
>
> **RE**: We want to explain that our focus is not on whether to perform fine-tuning. For supervised classification tasks, our goal is to build a reliable OOD detector for a trained classifier. To the best of our knowledge, most related OOD detectors were trained by fine-turning or using the features from a well pre-trained classification network [1,2,3,4,5,6,7] rather than training from scratch. We follow this general setting because it will allow us to focus more on OOD detection rather than diverting our effort on how to train a multiclass classifier. More importantly, it will enable researchers to more fairly compare different approaches with the fixed network architectures and standard pre-trained model weights.
>
> [1] Liang S, Li Y, Srikant R. Enhancing the reliability of out-of-distribution image detection in neural networks, ICLR, 2018.
>
> [2] Hendrycks D, Gimpel K. A baseline for detecting misclassified and out-of-distribution examples in neural networks, ICLR, 2017.
>
> [3] Lee K, Lee K, Lee H, et al. A simple unified framework for detecting out-of-distribution samples and adversarial attacks, NeurIPS, 2018.
>
> [4] Liu W, Wang X, Owens J, et al. Energy-based out-of-distribution detection, NeurIPS, 2020.
>
> [5] Sastry C S, Oore S. Detecting out-of-distribution examples with gram matrices, ICML, 2020.
>
> [6] Sun Y, Guo C, Li Y. React: Out-of-distribution detection with rectified activations, NeurIPS, 2021.
>
> [7] Bibas K, Feder M, Hassner T. Single Layer Predictive Normalized Maximum Likelihood for Out-of-Distribution Detection, NeurIPS, 2021.

---

> ### Author Response · Authors · 2022-08-08
> **To Reviewer 9BXS: would you please kindly reconsider your rating based on our response.**
>
> Thanks very much for your comments. Regarding your questions, we have provided detailed point-to-point responses, which we believe can address your major concerns and clarify the issues raised. We wonder whether you would kindly reconsider your rating based on our responses to your questions. Your consideration is highly appreciated. Many thanks!
>
> Authors of the Paper6858

---

### Official Review · Reviewer_QcD7 · 2022-07-26

**Rating:** 7
**Confidence:** 5
**Soundness:** 4 excellent
**Presentation:** 3 good
**Contribution:** 3 good

**Summary:**

The authors tackle the issue of out-of-distribution detection for deep learning classifiers by proposing two regularization terms that can be added to a loss-function to fine-tune a pretrained network in order to ensure a calibrated probability output of P(X) that can be used to detect OOD samples.

Existing approaches rely on either using a function of the logits P(Y|X) or directly estimating P(X) based on high-level features of the deep learning classifier. However, the logits of a deep network can be overconfident (mis-calibrated) [1] and are often not linearly correlated with P(X) for OOD samples [2]. Furthermore, density estimation on a biased low-dimensional projection of X might not produce an unbiased estimate of P(X).

Instead, the authors propose to compute P(X) based on the softmax logits (P(Y|X)) after first (re)-calibrating the joint distribution P(X,Y) (g(x,y) in Eq. 3). Towards this goal, they derive a density-consistency regularization (DCR) term  based on the asymptotic testing of the consistency of P(Y) (based on logits) and its empirical density function, in a batch-wise fashion.
This regularization term effectively (although asymptotically) re-calibrates v(Y) and thus, according to the author’s derivations, re-calibrates the joint distribution h(X,Y). Since a discriminative model already optimizes for the accuracy of P(Y|X), calibrating P(X,Y) = P(Y|X) P(X) should ensure a calibrated P(X) which can be used as a reliable OOD score.

As a second contribution, this paper introduces a contrastive distribution regularization (CDR) term that incentivizes a high likelihood ratio between augmented samples (assumed to be distribution-deviating) and in-distribution samples.


[1] Matthias Hein, Maksym Andriushchenko, and Julian Bitterwolf. Why relu networks yield high-confidence predictions far away from the training data and how to mitigate the problem. In 2019 IEEE/CVF Conference on Computer Vision and Pattern Recognition (CVPR), pages 41–50, 2019

[2] Weitang Liu, Xiaoyun Wang, John Owens, and Yixuan Li. Energy-based out-of-distribution detection. In Advances in Neural Information Processing Systems, pages 21464–21475, 2020.


**Questions:**

1. It is unclear to me why the OOD performance degrades with a larger batch size (Table 3) as it should ensure a tighter CLT bound thus a more accurate analytical approximation in the density-consistency regularization (DCR) term. Do the authors have any intuition regarding this?
2. To establish the utility of density-consistency regularization, could the authors compare and report the CDFs of the joint (X,Y) or marginals between the networks fine-tuned by the baselines and that using the DCR loss?
3. The aggregation of the results across different metrics for the ablation study somewhat defies the purpose of disentangling which features matter for which datasets, especially when there is a high variance of difficulty across the datasets (20% FPR vs. ~0). Would it be possible to de-aggregate these results in the appendix?
4. This approach does indeed seem simple and, based on the reported results, effective. However, could the authors provide some information on the inference time/memory requirements of this approach (as compared to some memory-intensive approaches such as Gram, for example) ?
5. Could you report the performance of adding CDR alone in Table 2? While the paper is anchored on the novelty of the density consistency regularization term, it seems that the most empirical gain might be from the contrastive distribution regularization term.
6. What do you mean by "q is a detached tensor dynamically determined to lienarly map the exponent..." shouldn't this just be a constant for numerical stability rather than being an adaptive tensor?

**Limitations:**

1. The performance on ImageNet is not as impressive as that on CIFAR-10/100 which brings up the question of whether this approaches scales with the number of classes or size of dataset. Furthermore, larger networks that are becoming more commonplace for ImageNet-derived tasks might be better calibrated [3]. It would be interesting to verify that this approach can be as effective on these modern architectures.
2. While the authors claim robustness to hyper parameter choices, the FPR metric does seem to change significantly (on the average benchmark) with different choices of “r” and batch size. It would be interesting to see if that's on a subset of the dataset or a common theme.


**Strengths And Weaknesses:**

Strengths:
* The proposed approach is simple and the derivation is straightforward.
* The idea of converting the asymptotic consistency test into a regularization term for fine-tuning seems novel to me. The idea of calibrating P(Y) to ensure the calibration of P(X) also seems new.
* The authors present an extensive empirical study in which their approach outperforms most state of the art benchmarks.

Weaknesses:
* The writing could be clearer. For example, the authors claim to propose a new approach for estimating P(X) when they are actually re-calibrating the logits-based estimation of P(X). “Calibration” was not once mentioned in the paper even though it is equivalent to the assumption of Eq. 3 (that the learned model is faithful to the joint distribution).
* The authors emphasize on the novelty of the density-consistency regularization term. However, from the empirical study as well as previous literature, it is highly likely that the contrastive-loss term is the main driver of the improved performance. Ablation studies do not disentangle the contribution of this term from that of the density-consistency one (Table 2 is lacking a row for enabling CDR alone).
* It is difficult to assess the significance of these contributions due to the lack of a detailed (rather than aggregated) ablation study and the use of smaller networks than commonly deployed for the ImageNet results (which I believe are more interesting than those of CIFAR-10/100).

[3] Minderer, Matthias, Josip Djolonga, Rob Romijnders, Frances Hubis, Xiaohua Zhai, Neil Houlsby, Dustin Tran, and Mario Lucic. "Revisiting the calibration of modern neural networks." Advances in Neural Information Processing Systems 34 (2021): 15682-15694.

---

> ### Author Response · Authors · 2022-08-02
> **Response to Reviewer (part 1)**
>
> **Comment #1**:  The writing could be clearer. For example, the authors claim to propose a new approach for estimating P(X) when they are actually re-calibrating the logics-based. “Calibration” was not once mentioned in the paper even though it is equivalent to the assumption of Eq. 3 (that the learned model is faithful to the joint distribution).
>
> **RE**: Thanks for your suggestion. We have indeed used “calibrate” to state the role of key formula Eq.(14) in line 225 and also used it in the conclusion. However, as suggested, we will rephrase some other sentences using “calibration” in the manuscript.
>
> **Comment #2**:  The authors emphasize on the novelty of the density-consistency regularization term. However, from the empirical study as well as previous literature, it is highly likely that the contrastive-loss term is the main driver of the improved performance. Ablation studies do not disentangle the contribution of this term from that of the density-consistency one (Table 2 is lacking a row for enabling CDR alone).
>
> **RE**: Thanks for your suggestion. We would like to emphasize our contrastive distribution regularization (CDR) is also one of our novel contributions. Our CDR novelly provides a theoretical framework utilizing the distribution deviating operator for improving the OOD detector. It naturally fits and strengthens the density regularization by using Eq.(16). One key aspect of such a novel design is contrasting densities of deviated samples can be used to boost the performance of the OOD detector. More importantly, our proposed analytical CDR can be used to explore more useful distribution deviators for OOD detection research.
>
> However, as suggested, we also conduct an ablation study to disentangle the contribution between contrastive distribution regularization (CDR) and density consistency regularization (DCR). The updated result for Table 2 is listed below. It can be seen that both DCR and DCR are important to improve detector performance.
>
> | in-dist (model) | DCR | CDR | mean AUROC (↑) | mean AUPR (↑) | mean FPR95% (↓) |
> |:---------------:|:---:|:---:|:--------------:|:-------------:|:---------------:|
> |                 |  -  |  -  |      80.29     |     95.19     |      76.73      |
> |    CIFAR-100    |  √  |  -  |      92.71     |     98.40     |      37.29      |
> |   (WideResNet)  |  -  |  √  |      93.69     |     98.64     |      33.01      |
> |                 |  √  |  √  |      98.85     |     99.75     |       5.41      |
>
> **Comment #3**:  It is difficult to assess the significance of these contributions due to the lack of a detailed (rather than aggregated) ablation study and the use of smaller networks than commonly deployed for the ImageNet results (which I believe are more interesting than those of CIFAR-10/100).
>
> **RE**: In response to comment #2, we have now updated Table 2 for the ablation study to disentangle the contribution of two regularization terms. In the current version, we focus on the theoretical part of our approach and fairly evaluate our method on the most commonly used OOD detection benchmark. To our best knowledge, we have conducted the most extensive evaluation compared to diverse previous studies [1-6]. Model comparison on much larger deep networks is also a critical issue, especially for those computation resource-demanding methods, such as Gram [4] and OECC [7]. We agree that testing on very large-scale networks on ImageNet will be a very interesting point, and we will do this experiment when sufficient computation resources are available. At the moment, we will put this in our future work.
>
> [1] Liang S, Li Y, Srikant R. Enhancing the reliability of out-of-distribution image detection in neural networks, ICLR, 2018.
>
> [2] Lee K, Lee K, Lee H, et al. A simple unified framework for detecting out-of-distribution samples and adversarial attacks, NeurIPS, 2018.
>
> [3] Liu W, Wang X, Owens J, et al. Energy-based out-of-distribution detection, NeurIPS, 2020.
>
> [4] Sastry C S, Oore S. Detecting out-of-distribution examples with gram matrices, ICML, 2020.
>
> [5] Sun Y, Guo C, Li Y. React: Out-of-distribution detection with rectified activations, NeurIPS, 2021.
>
> [6] Bibas K, et al. Single Layer Predictive Normalized Maximum Likelihood for Out-of-Distribution Detection, NeurIPS, 2021.
>
> [7] Papadopoulos A A, et al. Outlier exposure with confidence control for out-of-distribution detection[J]. Neurocomputing, 2021, 441: 138-150.

---

> ### Author Response · Authors · 2022-08-02
> **Response to Reviewer (part 2)**
>
> **Comment #4**:  It is unclear to me why the OOD performance degrades with a larger batch size (Table 3) as it should ensure a tighter CLT bound thus a more accurate analytical approximation in the density-consistency regularization (DCR) term. Do the authors have any intuition regarding this?
>
> **RE**: We would like to highlight that the performance variations between batch sizes are relatively small, with $p$-values$>$0.05 for both FPR95%, AUROC and AUPR metrics (Kruskal-Wallis test, testing the performance variation to batch size for different OOD datasets). This indicates that the performance is relatively robust to the batch size (please also note that the performance variations of different hyperparameters in Table 3 are very small compared to the performance variations of different models in Table 1). We agree it needs further investigation why larger batch sizes did not consistently improve model performance. At the moment, a possible intuitive explanation we can provide is that there exists a trade-off. When the batch size is relatively small, the regularization strength may be greater (having more combinatorial constraints on batch-wise statistics with fixed fine-tuning epochs), and the convergence may not be perfect. When the batch size becomes larger, although the convergence is better guaranteed, the regularization strength may be weakened.
>
> **Comment #5**:  To establish the utility of density-consistency regularization, could the authors compare and report the CDFs of the joint (X,Y) or marginals between the networks fine-tuned by the baselines and that using the DCR loss?
>
> **RE**: It is hard to show the CDF of very high-dimensional random variable $X$, but we indeed have compared the marginal distribution of $Y$ both w/ and w/o using DCR. As expected, DCR does calibrate the marginal distribution of $P(y)$ and make it closer to the empirical distribution. We will add this to our final paper.
>
> **Comment #6**:  The aggregation of the results across different metrics for the ablation study somewhat defies the purpose of disentangling which features matter for which datasets, especially when there is a high variance of difficulty across the datasets (20% FPR vs. ~0). Would it be possible to de-aggregate these results in the appendix?
>
> **RE**: Thanks for your suggestion. We will add the de-aggregated results of the ablation study across different datasets in the appendix. The tables are very long with lots of numbers. For the sake of clarity and readability, we do not insert those number here for now.
>
> **Comment #7**:  This approach does indeed seem simple and, based on the reported results, effective. However, could the authors provide some information on the inference time/memory requirements of this approach (as compared to some memory-intensive approaches such as Gram, for example) ?
>
> **RE**: Thanks for your suggestion. The inference time and memory requirements for comparing our method with Gram [1] and pNML [2] are listed below (evaluating 5000 images in our ImageNet experiment, PyTorch framework with Nvidia A100 GPU). It worth mentioning that another evaluated method OECC [3] in our manuscript had similar memory requirement as the Gram method since they combined Gram features for OOD detection.
>
> |                     | Ours     | pNML     | Gram     |
> |---------------------|----------|----------|----------|
> | GPU Memory (MB)     | 4289     | 7949     | 25598    |
> | Inference time (ms) | 53506.12 | 49374.42 | 74044.12 |
>
> [1] Sastry C S, Oore S. Detecting out-of-distribution examples with gram matrices, ICML, 2020.
>
> [2] Bibas K, et al. Single Layer Predictive Normalized Maximum Likelihood for Out-of-Distribution Detection, NeurIPS, 2021.
>
> [3] Papadopoulos A A, et al. Outlier exposure with confidence control for out-of-distribution detection[J]. Neurocomputing, 2021, 441: 138-150.
>
> **Comment #8**:  Could you report the performance of adding CDR alone in Table 2? While the paper is anchored on the novelty of the density consistency regularization term, it seems that the most empirical gain might be from the contrastive distribution regularization term.
>
> **RE**: Thanks for your suggestion. The results are attached in response to comment #2.
>
> **Comment #9**:  What do you mean by "q is a detached tensor dynamically determined to lienarly map the exponent..." shouldn't this just be a constant for numerical stability rather than being an adaptive tensor?
>
> **RE**: Since the classification logits vary in a relatively large range, to ensure the numerical stability of the contrastive distribution regularization, we constructed the constant $q$ to linearly map the exponent in Eq.(16) from a larger range to a smaller range. In the form of addition, it required $q$ to be adaptively varied with the exponent, but we have detached this tensor to ensure it is not involved in the back-propagation process; therefore, it can be considered as a constant in this sense.

---

> ### Author Response · Authors · 2022-08-02
> **Response to Reviewer (part 3)**
>
> **Comment #10**:  The performance on ImageNet is not as impressive as that on CIFAR-10/100 which brings up the question of whether this approaches scales with the number of classes or size of dataset. Furthermore, larger networks that are becoming more commonplace for ImageNet-derived tasks might be better calibrated [3]. It would be interesting to verify that this approach can be as effective on these modern architectures.
>
> **RE**: Thanks for your suggestion. As in response to comment #3, in the current version, we focus on the theoretical part of our approach and fairly evaluate our method on the most commonly used OOD detection benchmarks. To our best knowledge, we have conducted the most extensive evaluation compared to previous studies (see refs [1-6] in response to comment#3). We acknowledge that assessment and comparison on modern larger deep networks are worth investigating for most OOD detectors, especially for these computation resource-demanding approaches, e.g., Gram and OECC (see refs [4] and [7] in response to comment#3). We agree that it would be an interesting point to test and compare diverse existing approaches on very large-scale networks on ImageNet. We will conduct this experiment when sufficient computing resources are available. For now, we will place it in our future work.
>
> **Comment #11**:  While the authors claim robustness to hyper parameter choices, the FPR metric does seem to change significantly (on the average benchmark) with different choices of “r” and batch size. It would be interesting to see if that's on a subset of the dataset or a common theme.
>
> **RE**: We would like to highlight that the performance variations of our approach with different hyperparameters (both $r$ and batch size) are relatively much smaller compared to the performance variations of different models (see both Table 1 and Table 3). Moreover, the performance of our model both consistently and significantly outperformed that of existing approaches with varying choices of parameters. Therefore, we considered the model performance is quite robust in this sense. In addition, the statistical tests we conducted also reveal that the distribution of model performance in terms of FPR95% (with different OOD datasets in the ablation study) does not show significant changes with hyperparameter variation ($P$ values>0.05 for both $r$ and batch size, Kruskal-Wallis test). Moreover, as suggested, we will also include the de-aggregated result of the ablation study in the appendix.

---

### Meta-Review · Area_Chair_ZAB4 · 2022-08-26

**Recommendation:** Accept
**Confidence:** Less certain

**Metareview:**

This paper develops a method for improving out-of-distribution detection in deep learning based on a novel regularization term.  There was significant variance in review scores with two championing the paper for acceptance and two borderline scores (7, 7, 5, 4) resulting in an aggregated score just above borderline accept.  The reviewers arguing for acceptance found the method novel, the simplicity of the algorithm compelling and the experiments extensive and convincing.  One reviewer was concerned that baseline comparisons provided in the paper seem less strong than reported in other work.  Two reviewers questioned the mathematical derivations and some of the underlying assumptions of the paper.

That two reviewers are arguing for acceptance is a signal that the paper could be a useful contribution and interesting to the community.  Since the experiments seem extensive and seem to demonstrate that the method consistently works well, and given that it is simple to implement, that seems to validate the underlying assumptions and it could provide a useful baseline.  Therefore the recommendation is to accept the paper.  Please make sure to address the remaining reviewer concerns in the final manuscript.

**Award:**

No

---

### Decision · Program_Chairs · 2022-09-14

Accept